# Anti-progestin therapy targets hallmarks of breast cancer risk

Bruno M. Simões[1 ✉], Robert Pedley[1,2,3], Curtis W. McCloskey[4,5,6], Matthew Roberts[1], Austin D. Reed[7], Alecia-Jane Twigger[7], Pirashaanthy Tharmapalan[4], Amanda Caruso[1,8], Sara Cabral[1], Anthony J. Wilby[1], Hannah Harrison[1], Yuxi Zhou[1,2], Alice Greenhalgh[1], Suad A. Alghamdi[1], Martina Forestiero[1,8], Jesica Lopez-Muñoz[1], Jasmin Roche[1], Ren Jie Tuieng[9], Muhammad A. Khan[2], Steven Squires[10], Susan M. Astley[1,10], Elaine F. Harkness[10], Angélica Santiago-Gómez[1], Katherine Spence[1], Jessica Ritchie[11], Susan Pritchard[11], Yit Lim[11], Michael J. Sherratt[9], Sebastiano Andò[8], Anthony Howell[1,11], D. Gareth Evans[11,12], Andrew P. Gilmore[1,2], Walid T. Khaled[7], Rama Khokha[4], Robert B. Clarke[1,14 ✉] & Sacha J. Howell[1,11,13,14 ✉]

Breast cancer is the leading cause of cancer-related death in women worldwide[1]. Here, in the Breast Cancer-Anti-Progestin Prevention Study 1 (BC-APPS1; NCT02408770), we assessed whether progesterone receptor antagonism with ulipristal acetate for 12 weeks reduces surrogate markers of breast cancer risk in 24 premenopausal women. We used multilayered OMICs and live-cell approaches as readouts for molecular features alongside clinical imaging and tissue micromechanics correlates. Ulipristal acetate reduced epithelial proliferation (Ki67) and the proportion, proliferation and colony formation capacity of luminal progenitor cells, the putative cell of origin of aggressive breast cancers[2]. MRI scans showed reduction in fibroglandular volume with treatment, whereas single-cell RNA sequencing, proteomics, histology and atomic force microscopy identified extracellular matrix remodelling with reduced collagen organization and tissue stiffness. Collagen VI was the most significantly downregulated protein after ulipristal acetate treatment, and we uncovered an unanticipated spatial association between collagen VI and SOX9[high] luminal progenitor cell localization, establishing a link between collagen organization and luminal progenitor activity. Culture of primary human breast epithelial cells in a stiff environment increased luminal progenitor activity, which was antagonized by anti-progestin therapy, strengthening this mechanistic link. This study offers a template for biologically informed early-phase therapeutic cancer prevention trials and demonstrates the potential for premenopausal breast cancer prevention with progesterone receptor antagonists through stromal remodelling and luminal progenitor suppression.

Breast cancer is the leading cause of cancer-related mortality in women globally and the most common of any cause of death in UK women aged 35–64 years[1] (https://www.ons.gov.uk/). In both mouse and human mammary glands, progesterone-induced proliferation of stem and progenitor cells results in increased branching and ductal complexity[3,4]. This proliferation is mediated through paracrine signals secreted from progesterone receptor (PR)-positive 'luminal mature' cells that act on PR-negative 'luminal progenitor' cells, the postulated cell of origin for basal (triple-negative) breast cancer[2,4–7]. In premenopausal women, breast epithelial cell proliferation is highest during the progesterone-dominant luteal phase of the menstrual cycle and can be reduced by anti-progestins such as mifepristone[8,9]. Supplementation of progestin, as a contraceptive or hormone replacement therapy, increases breast cancer incidence[10–12] and stimulates epithelial proliferation and hyperplasia in preclinical models[13]. Conversely, inhibiting PR or its downstream pathways in mouse models results in a

[1]Manchester Breast Centre, Division of Cancer Sciences, School of Medical Sciences, Faculty of Biology, Medicine and Health, University of Manchester, Manchester, UK. [2]Manchester Cell-Matrix Centre, University of Manchester, Manchester, UK. [3]Medicines Discovery Catapult, Alderley Park, Cheshire, UK. [4]Princess Margaret Cancer Centre, University Health Network, Toronto, Ontario, Canada. [5]Department of Computational and Systems Biology, University of Pittsburgh, Pittsburgh, PA, USA. [6]UPMC Hillman Cancer Center, University of Pittsburgh School of Medicine, Pittsburgh, PA, USA. [7]Department of Pharmacology, University of Cambridge and Wellcome-MRC Cambridge Stem Cell Institute, Cambridge, UK. [8]Department of Pharmacy, Health and Nutritional Sciences, Health Center, University of Calabria, Cosenza, Italy. [9]Division of Cell Matrix Biology & Regenerative Medicine, School of Biological Sciences, Faculty of Biology, Medicine and Health, University of Manchester, Manchester, UK. [10]Division of Informatics, Imaging & Data Sciences, School of Health Sciences, Faculty of Biology, Medicine and Health, University of Manchester, Manchester, UK. [11]Nightingale and Prevent Breast Cancer Centre, Manchester University NHS Foundation Trust, Wythenshawe Hospital, Manchester, UK. [12]Manchester Centre for Genomic Medicine, Division of Evolution, Infection and Genomic Health, University of Manchester and St Mary's Hospital, Manchester, UK. [13]Department of Medical Oncology, The Christie NHS Foundation Trust, Manchester, UK. [14]These authors jointly supervised this work: Robert B. Clarke, Sacha J. Howell. ✉e-mail: bruno.simoes@manchester.ac.uk; robert.clarke@manchester.ac.uk; sacha.howell@nhs.net

substantial reduction in mammary carcinogenesis through suppression of mammary luminal progenitor and stem cell activity[7,14–17], with clinical window studies also showing reduced proliferation in normal and cancerous breast tissue[9,18–20].

One of the challenges of primary prevention studies is identifying clinically relevant surrogate indicators of risk reduction. Mammographic density is one of the strongest risk factors for breast cancer[21] and is a reliable clinical measure across a range of methods, including automated volumetric analysis[22]. Magnetic resonance imaging (MRI) measurements of fibroglandular volume (FGV) correlate well with automated volumetric mammographic density, and FGV is greater in the luteal than in the follicular phase of the menstrual cycle[23,24]. Mammographic density also declines through menopause and increases in post-menopausal women using progestin-containing hormone replacement therapy[25,26]. Mammographically dense areas contain increased epithelial and fibroblast cell numbers as well as collagen[27,28]. Breast stroma has a role in cancer initiation and progression by regulating epithelial cell proliferation[29]; in rodent models, stromal crosslinked fibrillar collagen increases the incidence of invasive tumour formation[30,31]. Periductal tissue stiffness positively correlates with increased collagen fibril alignment in human breast tissue with high mammographic density[32]. The question that we set out to address was how anti-progestin therapy might prevent luminal progenitor cells undergoing oncogenic transformation through both direct effects on the epithelium and indirect effects on the microenvironment structure, composition and stiffness that could potentially be appreciated radiologically.

Here we report findings from the BC-APPS1 study (NCT02408770) that demonstrate the profound effects of 12 weeks of ulipristal acetate (UA) therapy on normal breast composition in 24 premenopausal women at increased risk of breast cancer. We conducted multi-OMICs analyses on paired vacuum-assisted breast biopsy (VAB) tissues, before and after treatment, alongside critical clinical correlates such as mammographic density or FGV. Our comprehensive analyses of the primary tissues at cellular, molecular and functional levels have exposed powerful dependencies of the extracellular matrix (ECM) and breast epithelial progenitor fractions on hormone-dependent stromal triggers. This work demonstrates that critical components of the mammary progenitor cell niche and mammographic density determinants can be altered with anti-progestins. Together, targeting PR signalling may be a valuable strategy in preventing aggressive breast cancers in premenopausal women at increased risk.

## Anti-progestin prevention study participants

Between 29 March 2016 and 11 March 2019, 32 women with an increased risk of breast cancer due to their family histories consented to the BC-APPS1 study. Six failed screening owing to inability to time the luteal phase of the menstrual cycle (P4 of less than 15 nmol l⁻¹). Of the 26 eligible participants who received UA therapy, two underwent baseline investigations but subsequently withdrew from the study before the second VAB: one participant owing to anxiety related to a small biopsy-associated haematoma and one participant owing to drug-induced anxiety. Therefore, 26 participants were included in toxicity analyses, and 24 with paired VAB samples were included in molecular analyses of response to UA therapy. Downstream OMICs analyses of VAB samples were applied to selected samples depending primarily on the tissue availability for the technology utilized. Baseline VAB was timed to the luteal phase of the menstrual cycle owing to the profound effect of cycling ovarian hormones on breast biology and epithelial dynamics. The trial schema in Fig. 1a outlines our systematic multi-tiered workflow of OMICs analyses. Participant demographics are presented in detail in Supplementary Table 1. In summary, the 24 participants with paired samples had a median age of 39 years (range of 34–44 years), median BMI of 26 kg m⁻² (range of 21–42) and a median remaining lifetime breast cancer risk of 25.5% (range of 17–38.3%; Tyrer

Cuzick v7.02). Treatment was generally well tolerated with no grade 3 or 4 adverse events (Supplementary Table 2).

## Anti-progestin treatment reduces luminal progenitor activity

The primary end point of the BC-APPS1 study was epithelial proliferation assessed by Ki67 immunohistochemistry, chosen primarily to power the study statistically, as Ki67 is not a recognized surrogate for breast cancer risk. The study met its primary end point with a significant reduction in proliferation between baseline (8.2%; 95% confidence interval (CI) 5.2–11.2%) and 12-week samples (2.9%; 95% CI 2.1–3.7%; $P < 0.0001$; Fig. 1b). Mean serum progesterone levels reduced with treatment from 36 nmol l⁻¹ (95% CI 29.4–41.6 nmol l⁻¹) at baseline to less than 3 nmol l⁻¹ (95% CI 0.3–4.6 nmol l⁻¹; $P < 0.0001$; Extended Data Fig. 1a), effectively abrogating the luteal phase. Both the epithelial area within each lobule (Fig. 1c) and the average area of acinar structures (Extended Data Fig. 1b) were significantly reduced with UA treatment; however, the mean number of acini per lobule did not change (Extended Data Fig. 1c). Next, flow cytometry analysis showed a significant reduction in the luminal progenitor (CD49f⁺EpCAM⁺) fraction with treatment from 43% (95% CI 35–52%) to 30% (95% CI 21–39%; $P < 0.001$), with no significant changes detected in luminal mature (CD49f⁻EpCAM⁺) or basal (CD49f⁺EpCAM⁻/low) populations (Fig. 1d and Extended Data Fig. 1d). Epithelial colony-forming assays used to enumerate progenitor activity yield three distinct colony phenotypes: myoepithelial/basal, luminal and mixed (where mixed colonies represent bi-lineage differentiation potential)[33]. Anti-progestin treatment reduced the proportion of mixed colonies from 70% (95% CI 60–80%) to 55% (95% CI 44–67%; $P < 0.05$; Fig. 1e and Extended Data Fig. 1e). Mammosphere-forming efficiency (MFE), another measure of luminal progenitor activity, was also reduced by UA (baseline 0.29%; 95% CI 0.19–0.39% versus 12 weeks 0.16%; 95% CI 0.04–0.28%; $P < 0.01$; Fig. 1f). In vitro treatment of baseline cell suspensions with UA and an alternative anti-progestin (onapristone) similarly reduced MFE (Extended Data Fig. 1f,g). SOX9 is a marker of luminal progenitor cells[34], and both the overall percentage of SOX9⁺ (Extended Data Fig. 1h) and proliferating SOX9⁺ cells (dual staining for SOX9 and Ki67) were reduced with UA treatment (SOX9⁺Ki67⁺ at baseline 4.4%; 95% CI 1.6–7.2% versus 12 weeks 1.3%; 95% CI 0.7–1.9%; $P < 0.05$; Fig. 1g). Overall, these data demonstrate that anti-progestin treatment reduces the proportion, proliferation and activity of luminal progenitor cells in the normal breast tissue of women at increased breast cancer risk. Given that luminal progenitors are the putative cell of origin in basal (triple-negative) breast cancers, abrogation of this breast cancer precursor pool is pertinent for targeted breast cancer prevention.

## Luminal mature cells regulate the basal cell/fibroblast matrisome

To evaluate transcriptional changes with treatment, bulk tissue RNA sequencing (RNA-seq) analysis was performed. RNA quality was sub-optimal in at least one sample from each of 14 participants, and data are presented for the paired samples from 10 participants that met quality standards. UA treatment resulted in differential expression of 50 genes (log₂[fold change] (log₂FC) > 1.5, $P < 0.05$; Extended Data Fig. 2a), including two established PR target genes (*TNFSF11* and *CXCL13*) that were significantly downregulated with treatment (Extended Data Fig. 2b). Gene Ontology term analysis of the top 50 differentially expressed genes showed that almost half of these genes (23) were associated with the extracellular space (Extended Data Fig. 2a).

To define the molecular changes in diverse breast cell types after anti-progestin treatment, single-cell RNA-seq (scRNA-seq) profiling of six paired samples was performed (Fig. 2a). Single-cell transcriptomes of 115,875 cells were obtained after quality filtering for gene coverage, read counts and mitochondrial reads (see Methods).

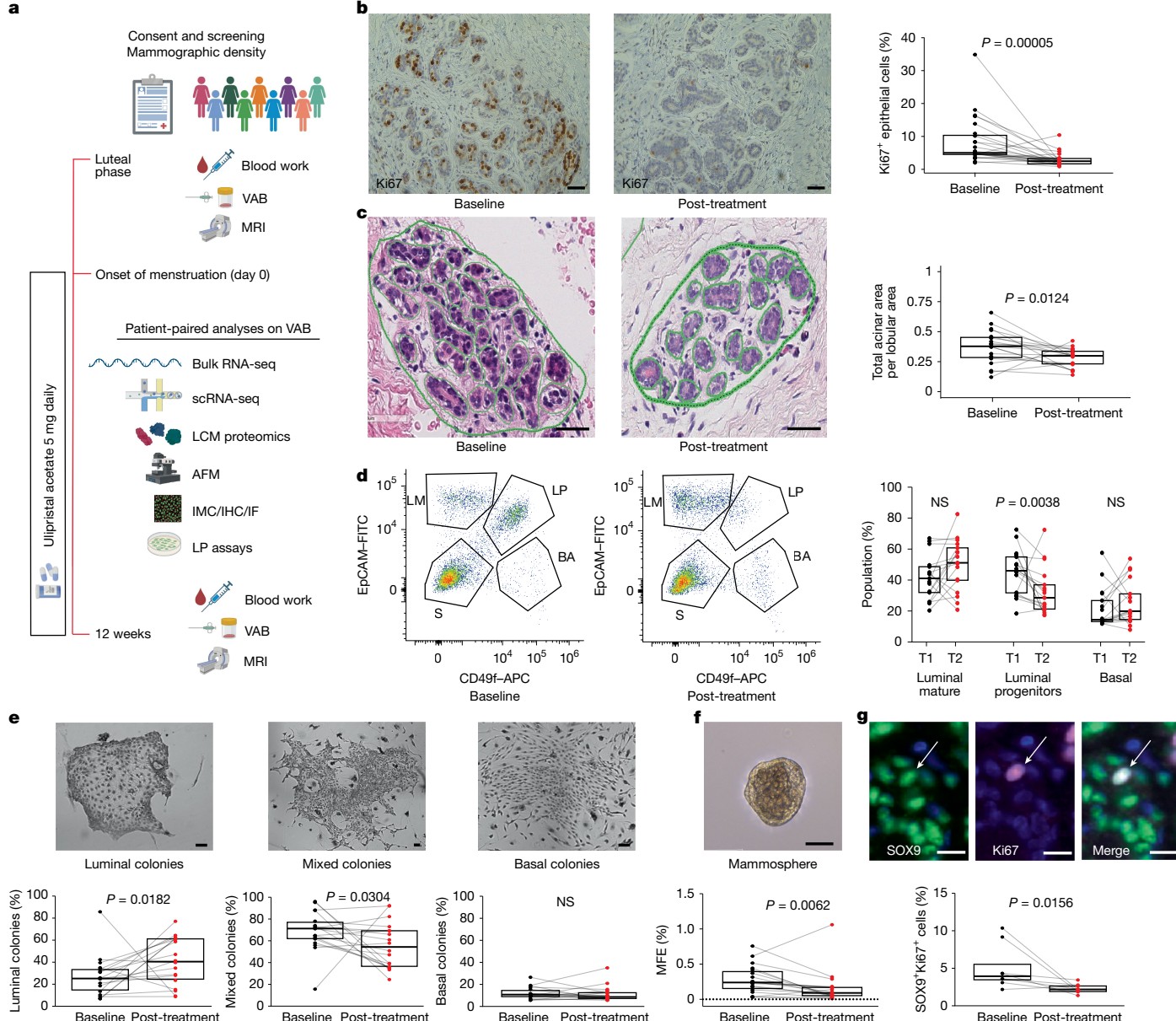

**Fig. 1 | Anti-progestin treatment reduces luminal progenitor activity.**
**a**, Trial schema of the BC-APPS1 study. A VAB was collected in the luteal phase (baseline), and repeated in the opposite breast after 12 weeks of UA (5 mg daily). AFM, atomic force microscopy; IF, immunofluorescence; IHC, immunohistochemistry; IMC, imaging mass cytometry. The trial schema was created using BioRender (https://biorender.com). **b**, Percentage of Ki67-positive cells in 24 paired breast tissue samples before (baseline) and after (post-treatment) 3 months of UA therapy. Representative staining is shown. **c**, Proportion of epithelial area per lobule area before (baseline) and after (post-treatment) 3 months of UA therapy ($n = 19$ tissue pairs). Examples of lobule epithelial areas (green outlines) are shown. **d**, Flow cytometry analysis of luminal mature (LM; CD49f⁻EpCAM⁺), luminal progenitor (LP; CD49f⁺EpCAM⁺), basal (BA; CD49f⁺EpCAM⁻/low) and stromal (S; CD49f⁻EpCAM⁻) cells. The graph

shows the percentage of epithelial populations (LP, LM and BA) in 17 tissue pairs. NS, not significant. **e**, Percentage of luminal, mixed or basal colonies in 18 breast tissue sample pairs before and after UA therapy. Representative examples of clonogenic assay colonies are shown above. **f**, MFE data expressed as a percentage for 19 tissue pairs. Horizontal dotted line, 0. A representative example of a mammosphere is shown above. **g**, Percentage of SOX9 and Ki67 double positive cells in eight tissue pairs quantified by immunofluorescence. The arrow in the representative images above indicates a cell expressing both SOX9 and Ki67. In all plots, boxplot centre lines represent median values and box bounds indicate the 25th and 75th percentiles, with connecting lines between paired data points. $P$ values were calculated with two-sided Wilcoxon matched-pairs signed-rank test (**b**–**g**). Scale bars, 50 μm (**b**,**c**,**e**,**f**) and 10 μm (**g**).

Uniform manifold approximation and projection (UMAP) analysis of the combined 12 samples revealed seven major cell populations (Fig. 2b). Using previously published gene signatures[35], we identified three epithelial (luminal adaptive secretory precursor (LASP), luminal hormone sensing (LHS) and basal-myoepithelial (BMYO)) and four stromal (fibroblasts, endothelial, perivascular and immune) cell types (Fig. 2c). A similar number of cells from baseline (56,014) and post-treatment (59,861) were analysed, and all seven cell populations

were present in each of the 12 samples (Supplementary Table 3). Using differential abundance testing, we did not observe any significant changes in cellular abundance following UA treatment across all seven broad cell populations (Fig. 2d). However, when considering only the total epithelial population, we observed a significant reduction in the proportion of LASPs post-treatment (Fig. 2d). This reduction was seen in five of six paired samples analysed, with no consistent trend for LHS and BMYO populations (Extended Data Fig. 3a). Participant samples

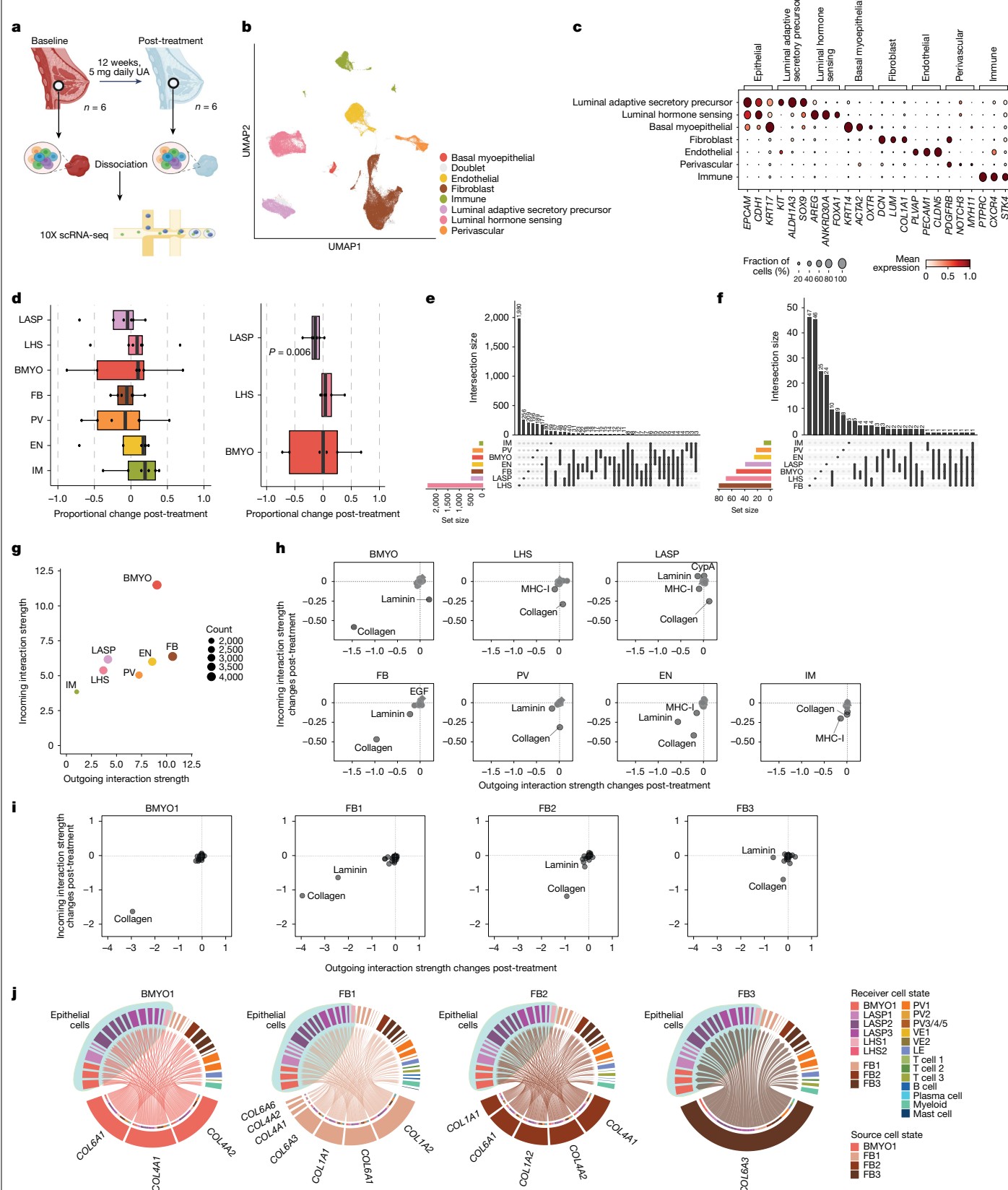

**Fig. 2** | See next page for caption.

with both flow cytometry (Fig. 1d) and scRNA-seq data showed a strong correlation ($r = 0.762$; $P = 0.0055$) between the percentage of luminal progenitor cells detected by flow cytometry and LASP cells identified by scRNA-seq (Extended Data Fig. 3b), indicating that luminal progenitor and LASP cells are largely the same population. Here we use 'luminal progenitor/LASP cells' to refer to luminal progenitor or LASP cells, defined by the specific assays. To explore further granularity within each cell type, we performed Leiden subclustering to match

**Fig. 2 | Transcriptome network analyses reveal that luminal mature cells orchestrate the matrisome landscape of basal and fibroblast cells.**
**a**, Workflow for paired biopsy single-cell transcriptomics from six participants at baseline and 12 weeks after UA treatment. The workflow was created using BioRender (https://biorender.com). **b**, UMAP of breast tissue cells annotated by broad cell type. *n* = 115,875 cells. **c**, Dot plot of broad cell-type marker genes. The columns correspond to key markers (normalized per gene) with brackets detailing the cell type, and the rows correspond to the cell population identified in the dataset. **d**, Proportionality fold change (post-treatment to baseline) across broad cell types (left) and restricted to epithelial cells (right). Positive or negative changes denote enrichment or depletion post-treatment, respectively. Boxplot centre lines represent median values, box bounds indicate the 25th and 75th percentiles, and whiskers extend to the extreme datapoint within 1.5 times the interquartile range (IQR) of the boxplot hinges. Significance was calculated with a two-sided Student's *t*-test adjusted *P* value for multiple comparisons using Benjamini–Hochberg correction. *n* = 6 tissue pairs.

**e**, UpSet plot depicting downregulated genes (less than −0.25 logFC, *P* < 0.05, Memento analysis) post-treatment across broad cell types. The intersection size indicates the number of genes uniquely regulated within a single cell type or shared across multiple cell types. **f**, UpSet plot depicting downregulated genes that encode proteins that act as ligands (less than −0.25 logFC, *P* < 0.05, Memento analysis) post-treatment across broad cell types. **g**, CellChat analysis of incoming–outgoing interaction strength between broad cell types at baseline. The node size represents the number of interactions in each cell type.
**h,i**, Differential L–R pathway signalling changes (post-treatment to baseline) across broad cell types (**h**) or within basal (BMYO1) and fibroblast (FB1–3) cell states (**i**). Negative values represent a decrease in L–R signalling post-treatment.
**j**, Chord diagram of pairwise downregulated collagen gene signalling post-treatment (less than −0.25 logFC, *P* < 0.05) from basal and fibroblast cell states (sender cells) to all breast cell states (receiver cells with 5% or more receptor expression), highlighting the epithelial populations in light blue.

the 'level 2' annotations used in the integrated Human Breast Cell Atlas (iHBCA), the largest integrated breast scRNA-seq dataset[35]. This identified several subclusters within each major cell type in our dataset, consistent across all six paired samples (Extended Data Figs. 4 and 5). iHBCA clusters BMYO2 and LASP4 (ref. 35) were not identified in the BC-APPS1 dataset, and differential abundance testing did not reveal any significant changes in the abundance of individual iHBCA-annotated subclusters following UA treatment (Extended Data Fig. 3c). This included the three LASP subclusters 1, 2 and 3, even when analysis was restricted to epithelial populations, suggesting that LASP subgroup response to UA is variable between individual participants (Extended Data Fig. 3d). Pairwise differential expression analysis for each of the major cell populations in response to UA revealed that the majority of differentially expressed genes were observed in LHS cells (Fig. 2e and Extended Data Fig. 6a), in which pathway analysis showed mainly downregulation of predominantly cell-intrinsic RNA processing pathways (Extended Data Fig. 6b). The known PR target genes *TNFSF11* and *CXCL13* did not meet stringent cell number and expression thresholds in this analysis (see the section 'Memento differential expression analysis' in Methods), but both showed significant per-participant downregulation in LHS cells following UA treatment (Extended Data Fig. 6c). Although other cell types also exhibited significant gene expression changes in response to UA treatment, these were considerably less pronounced than those observed in LHS cells. The list of differentially expressed genes for each of the seven cell populations is provided in Supplementary Table 4, and analyses of the major pathways that are upregulated and downregulated in each population after UA treatment are also included (Extended Data Figs. 6b and 7a). Given that paracrine signalling is known to have a critical role in normal mammary gland development, we next investigated differentially expressed ligands following UA treatment. LHS cells, but also fibroblast and BMYO cells, showed a high number of downregulated ligands (Fig. 2f). The number of upregulated ligands was lower overall, but higher in LHS cells than in the other cell types (Extended Data Fig. 6d). The list of differentially expressed ligands for each of the seven cell populations is provided in Supplementary Table 5.

To investigate how UA treatment affects cell communication networks across broad and granular cell states, Cell Chat[36] was used to model potential ligand–receptor (L–R) interactions between cell populations. After normalizing cell numbers to infer 'per cell' L–R interaction signalling strengths (ISSs) at baseline, BMYO and fibroblast populations had far greater incoming and outgoing ISSs than LASP or LHS populations (Fig. 2g). Annotating L–R pairs by established pathways revealed outgoing collagen ISS to be most markedly affected in BMYO and FB cells post-treatment, suggesting that UA therapy diminishes their role as sources of collagen signals, with a reduction in collagen incoming ISS seen in all seven cell states (Fig. 2h). To further corroborate these findings, gene set enrichment analysis of pairwise differentially

expressed genes in BMYO and fibroblast cells demonstrated robust overrepresentation of ECM-related terms including 'ECM organization', 'degradation of the ECM', 'collagen degradation' and 'assembly of collagen fibrils' (Extended Data Fig. 7a). When we restricted the analysis of pairwise differentially expressed genes to the 'Reactome ECM organization' gene set, fibroblast and BMYO cells exhibited a higher number of downregulated ECM genes than other cell populations, including many genes encoding collagen proteins (Extended Data Fig. 7b and Supplementary Table 6). By contrast, LHS cells displayed a greater number of upregulated ECM regulatory genes, with 4 out of the 13 genes encoding matrix metalloproteinases (*MMP1*, *MMP3*, *MMP10* and *MMP12*), which are known to have key roles in ECM degradation (Extended Data Fig. 8a,b). The complete list of ECM-related differentially expressed genes, both downregulated and upregulated, across the seven cell populations is provided in Supplementary Table 6. These results point to the ECM as a prime target downstream of UA treatment.

To determine whether specific subpopulations of fibroblast cells are driving reduced collagen signalling post-UA treatment, we assessed L–R networks within each cell subcluster. BMYO1 and fibroblast 1 (FB1) cells were confirmed as the primary sender subclusters exhibiting the most pronounced reduction in collagen signalling compared with FB2 and FB3 cell states (Fig. 2i). Analysis of collagen gene expression across all cell subclusters revealed that genes encoding collagen I, collagen IV and collagen VI are the most abundantly expressed in the human breast, with FB1–3 cells being primary producers of collagen I (*COL1A1* and *COL1A2*) and collagen VI (*COL6A1*, *COL6A2* and *COL6A3*), whereas BMYO1 cells primarily express collagen IV (*COL4A1* and *COL4A2*) and collagen VI (*COL6A1* and *COL6A2*; Extended Data Fig. 8c). As the most profound changes were in BMYO1 and FB1 cells, we examined the inferred differential collagen L–R interactions between these cells and all other subclusters. Collagen gene expression was downregulated after UA treatment in FB1 (specifically *COL1A1*, *COL1A2*, *COL4A1*, *COL4A2*, *COL6A1*, *COL6A3* and *COL6A6*), FB2 (specifically *COL1A1*, *COL1A2*, *COL4A1*, *COL4A2* and *COL6A1*), FB3 (*COL6A3*) and BMYO1 cells (specifically *COL4A1*, *COL4A2* and *COL6A1*; Extended Data Figs. 9 and 10d). This reduction in collagen expression potentially affects autocrine and paracrine interactions of numerous cell types given the collagen receptor expression across subclusters, most notably in the epithelial subclusters (Fig. 2j). A list of the collagen L–R interactions shown in Fig. 2j, along with the percentage of cells within each target population expressing collagen receptors, is provided in Supplementary Table 7. We then interrogated whether the observed collagen gene expression changes in FB1 and BMYO1 cells could be mediated by ligands secreted from LHS cells, the PR-expressing targets of UA treatment. We used NicheNet analysis[37] to investigate ligands predicted to be secreted by LHS cells (sender cell) that influence the expression of collagen target genes across FB1–3 and BMYO1 (receiver cell). Among

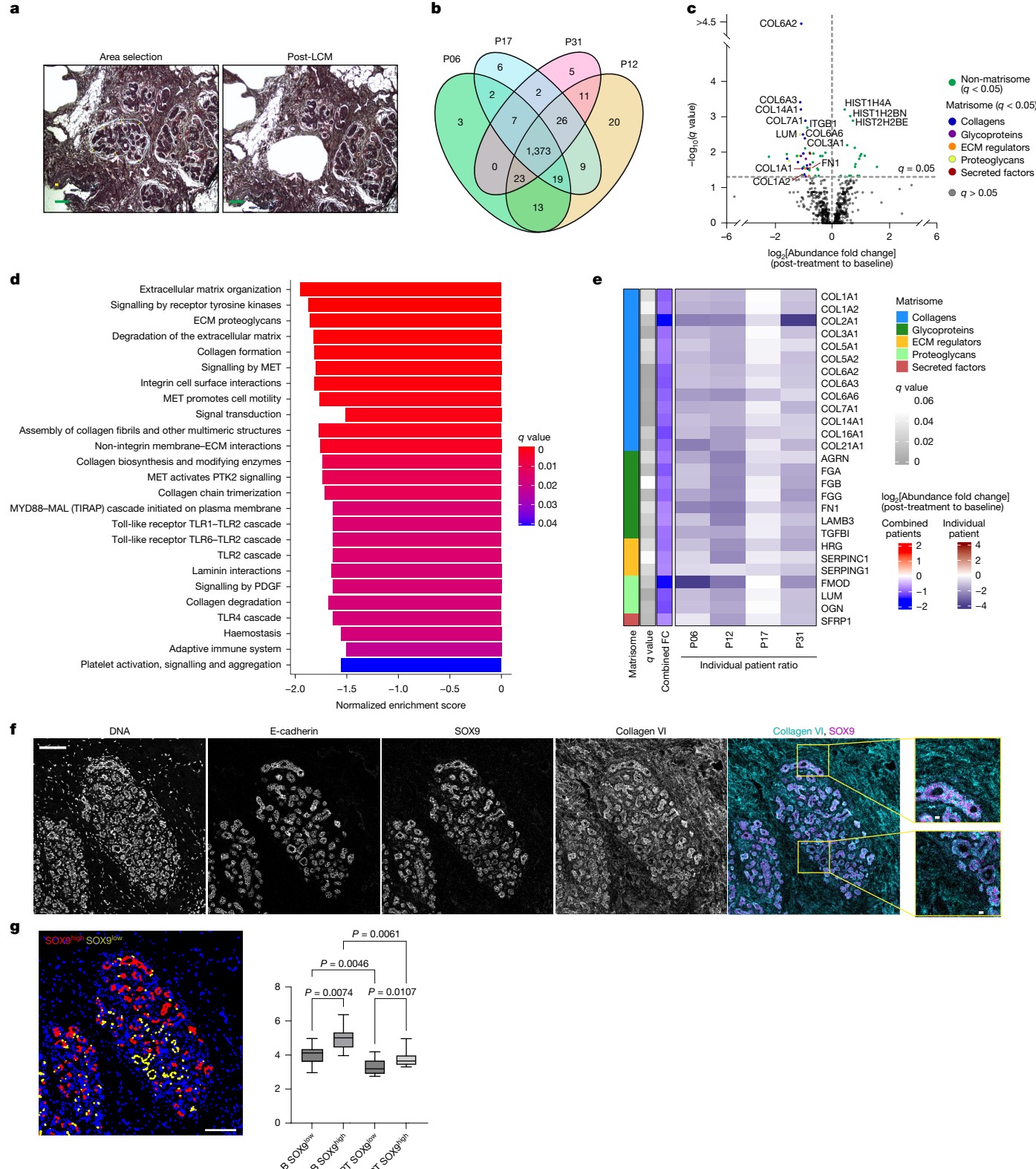

**Fig. 3** | See next page for caption.

the downregulated collagen genes, only *COL1A2* and *COL6A3* could be linked to LHS ligands. Downregulation of *WNT5A* and *RARRES1* in LHS cells were ligands predicted to regulate *COL6A3* expression in FB1 and FB3 cells, whereas *APOD* was predicted to regulate *COL1A2*, specifically in FB1 cells. No LHS ligands could be associated with regulation of collagen IV or other differentially expressed collagens in BMYO1 or FB2 cells,

suggesting that FB1 and FB3 expression of *COL1A2* and *COL6A3* are key targets of UA-driven alterations in LHS paracrine signalling (Extended Data Fig. 10a,b). *WNT5A* is expressed at higher levels in the LHS1 subcluster but was significantly downregulated in both LHS1 and LHS2 after UA treatment, whereas *COL6A3* was similarly expressed across FB1 and FB3 subclusters and was significantly downregulated in both

**Fig. 3 | Anti-progestin treatment remodels the breast matrix. a**, Lobular epithelium and peri-lobular stroma (within 25 μm of the observable edge of the epithelium) were laser capture microdissected from haematoxylin and eosin-stained paired tissue sections before (baseline) and after (post-treatment) UA treatment. A representative example of undissected tissue (left) and tissue after laser ablation (right) is shown. *n* = 4 tissue pairs. Scale bar, 100 μm. **b**, Venn diagram representing the distribution of the total proteins detected (1,519) in the four participants (P06, P12, P17 and P31) used for LCM proteomics. **c**, Volcano plot shows differential protein abundance analysis following UA treatment. Matrisome (structural ECM or ECM-modifying) proteins among the significantly altered proteins are colour coded according to their respective subcategories. **d**, Gene set enrichment analysis of LCM proteomics data using the Reactome Pathways reference set, showing pathways significantly altered by UA treatment. **e**, Heatmap of the 27 matrisome proteins identified as significantly differentially abundant after UA treatment. ECM proteins are grouped by their structural and functional properties. **f**, Imaging mass cytometry was performed on paired tissue sections before (baseline) and after (post-treatment) UA treatment. Representative images show staining with metal-conjugated antibodies to E-cadherin, SOX9 and collagen VI. Nuclei were visualized using a metal-tagged DNA intercalator. The yellow boxes indicate regions corresponding to the zoomed-in inserts. *n* = 8 tissue pairs. Scale bars, 100 μm and 10 μm (insets). **g**, Single-cell neighbourhood analysis of pericellular collagen VI abundance in SOX9[high] and SOX9[low] cell populations across paired BC-APPS1 samples at baseline (B) and post-treatment (PT) timepoints. Tissue images were segmented into single-cell objects, and cells were classified based on expression of specific markers. Analysis was performed on E-cadherin[+] cells classified as either SOX9[high] or SOX9[low]. For each selected cell, collagen VI staining intensity was quantified within a 10-μm radius. Scale bar, 100 μm. Boxplot centre lines represent median values, box bounds indicate the 25th and 75th percentiles, and whiskers denote minimum and maximum values. Statistical analysis was performed using a repeated measure one-way analysis of variance (ANOVA) followed by Sidak's multiple comparisons test. *n* = 8 tissue pairs.

post-treatment (Extended Data Fig. 10c,d). Fibroblasts (FB1–3) express nine receptors for *WNT5A*, which could mediate *WNT5A*-dependent regulation of *COL6A3* (Extended Data Fig. 10e).

Thus, beyond the known paracrine PR signals from luminal mature or LHS cells to luminal progenitor/LASP cells[4,7], we identified LHS-secreted progesterone ligands that are prime candidates for down-regulation of key collagen genes in human fibroblasts and basal cells, potentially shaping the matrisome landscape. Consistent with this, steroid hormones have recently been shown to stimulate ECM-remodelling fibroblasts, probably increasing mammary gland stiffness in mice[38]. Altogether, we identified striking cell–cell communication network alterations with major changes in fibroblast and basal cell matrisome components, probably mediated by a reduction in LHS-secreted ligands in response to 12 weeks of anti-progestin treatment in women at increased risk of breast cancer.

## Anti-progestin treatment remodels the breast matrix

To investigate the effects of UA therapy on breast tissue in proximity to luminal mature cells, we undertook laser capture microdissection (LCM) of breast lobules and peri-lobular stroma of four paired BC-APPS1 samples (Fig. 3a). Proteomic analysis of the tissue revealed the detection of 8,197 unique peptides corresponding to 1,519 proteins. Among these 1,519 proteins, 1,454 (96%) were consistently detected before and after treatment (data not shown) with 1,373 (90%) identified in all four participants (Fig. 3b). We identified 65 proteins regulated by UA treatment with *q* < 0.05 (Fig. 3c). Collagen α2 (VI) chain (COL6A2) and collagen α3 (VI) chain (COL6A3) were the most significantly downregulated proteins after treatment, whereas several histones (for example, histone H4 (HIST1H4A)) were the most significantly upregulated proteins. Gene set enrichment analysis using Reactome Pathway annotations revealed many pathways related to ECM and collagen (for example, 'ECM organization', 'ECM proteoglycans', 'collagen formation' and 'assembly of collagen fibrils') that were downregulated with UA treatment (Fig. 3d), in line with scRNA-seq data. Of the 65 proteins that were differentially abundant after treatment (Extended Data Fig. 11a), 27 (41.5%) were 'matrisome' proteins, comprising thirteen collagens, seven glycoproteins, three proteoglycans, three ECM regulators and one secreted factor (Fig. 3e), consistent with extensive remodelling of the ECM.

To examine the spatial location of luminal progenitor cells in relation to these specific stromal components and their perturbation in response to UA, Hyperion imaging mass cytometry was performed. Metal-conjugated antibodies for collagen I, collagen VI and fibronectin (FN1) were used in combination with markers of epithelial (E-cadherin) and luminal progenitor/LASP cells (SOX9), as well as Ki67. Eight paired BC-APPS1 samples with plentiful lobules were selected. Initial analysis confirmed decreased expression of collagen I, collagen VI and FN1 (Extended Data Fig. 11b) with UA treatment as previously observed by LCM proteomics (Fig. 3e). Single-cell neighbourhood analysis of SOX9[high] and SOX9[low] cells at baseline identified the SOX9[high] cells to be in close proximity to regions of high collagen VI and FN1 but not collagen I expression compared with SOX9[low] cells, a finding that persisted following UA treatment (Fig. 3f,g and Extended Data Fig. 11c). In both baseline and post-treatment conditions, Ki67[+] cells were significantly more prevalent in the SOX9[high] than SOX9[low] populations, confirming their higher proliferative activity, although UA treatment reduced proliferation in both populations (Extended Data Fig. 11d). The widespread staining pattern of collagen VI (a non-fibrillar collagen) is consistent with its expression in both stromal and epithelial cells (Extended Data Fig. 8c). These data identify stromal remodelling as an early event in breast tissue perturbed by anti-progestin treatment, although the persistent spatial association of SOX9[high] cells with collagen VI and FN1 after treatment suggests some continued colocalization despite short-term UA therapy (Extended Data Fig. 11c).

## Anti-progestins reduce stiffness-driven luminal progenitor activity

Increased elastic force (stiffness) between cells expressing oncogenes and their surrounding ECM have been shown to induce signals that promote epithelial transformation[39,40]. Increased matrix stiffness also enhances the enrichment of cancer stem cells and the induction of chemoresistance in patients with breast cancer[41]. Given the robust downregulation of multiple collagens post-treatment (Figs. 2 and 3), we next investigated the effects of a stiff microenvironment on breast tissue. Organoids (3D microstructures) from six women at higher risk of breast cancer were grown for 1 week in collagen-mimetic hydrogels with 'soft' (600–900 Pa) or 'stiff' (1,800–3,000 Pa) conditions (Supplementary Table 8). Expression of the PR target gene *TNFSF11* and luminal progenitor markers *SOX9* and *KIT* were increased in stiff hydrogels, which was confirmed at the protein level for SOX9 and KIT expression, and accompanied by increased MFE after extraction and dissociation of the cells (Fig. 4a–c). Anti-progestin treatment of breast microstructures using UA or onapristone blocked stiffness-induced increases in *SOX9* and *KIT*, as well as MFE; however, onapristone did not reduce MFE under soft conditions (Fig. 4b,c and Extended Data Fig. 12a,b; for gel source data, see Supplementary Fig. 1). Overall, these results establish that anti-progestin treatment attenuates stiffness-induced upregulation of progesterone signalling and progenitor cell activity, and also reduces the basal level of PR activity seen in softer gels in this in vitro system.

The structure and biomechanical properties of the ECM were next examined in the BC-APPS1 samples. Collagen coherency measurements on picrosirius red (PSR)-stained peri-lobular regions from 22

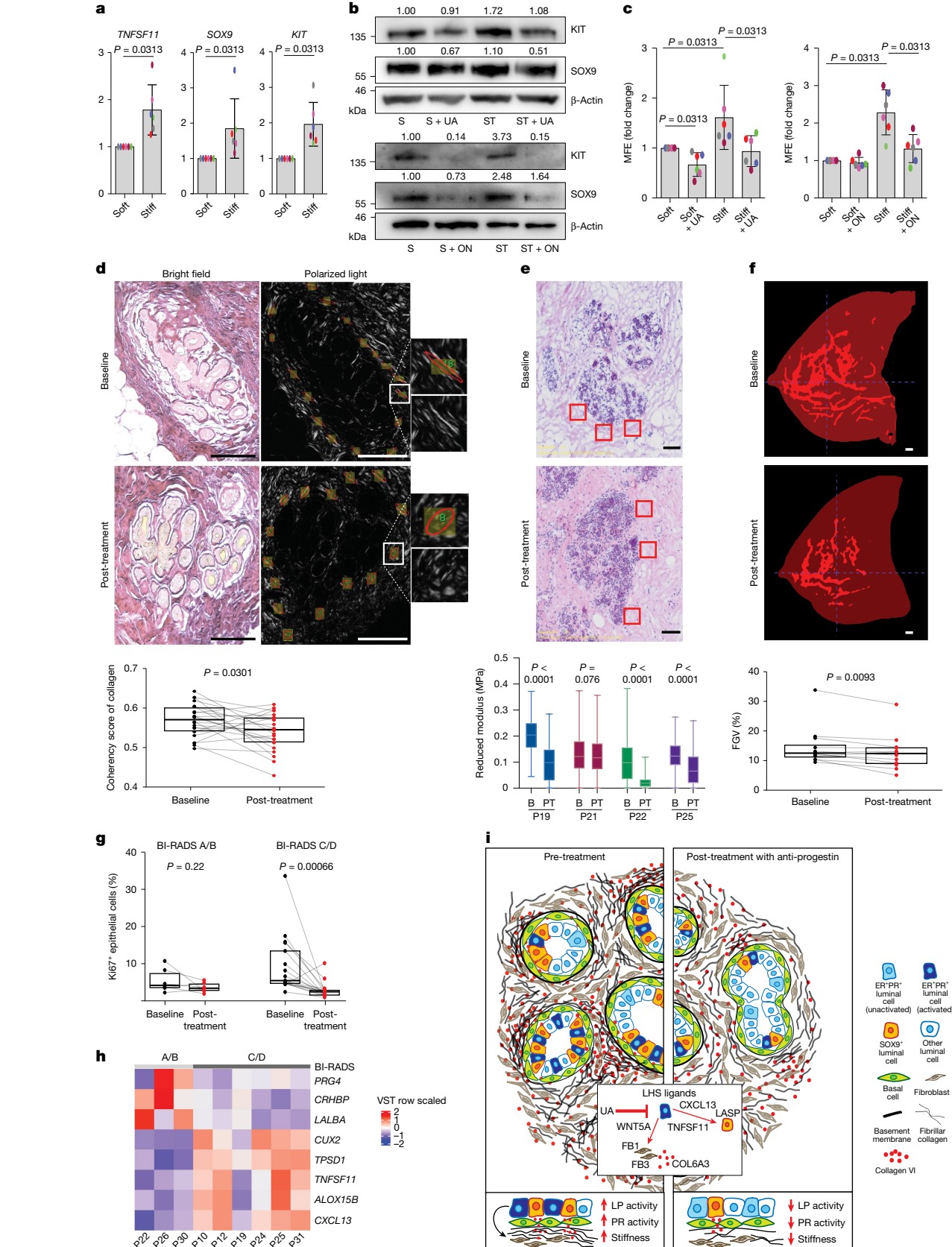

**Fig. 4** | See next page for caption.

**Fig. 4 | Tissue stiffness-amplified progesterone response and luminal progenitor activity are inhibited by anti-progestins. a**, Real-time PCR gene expression of *TNFSF11*, *KIT* and *SOX9* in breast tissue microstructures from women at increased cancer risk, cultured in 'soft' and 'stiff' hydrogels. Data are shown as mean fold change ± s.d., with individual points. $n = 6$ breast samples. **b**, KIT and SOX9 protein in breast microstructures (sample 1989N) cultured in soft (S) and stiff (ST) hydrogels, treated with UA (2 nM) or onapristone (ON; 100 nM). Densitometry normalized to β-actin is shown above the bands. $n = 3$ breast samples. **c**, MFE after culture in soft and stiff hydrogels with UA (2 nM) or ON (100 nM). Data are shown as mean fold change ± s.d., with individual points. $n = 6$ breast samples. **d**, Collagen coherency was assessed in peri-lobular regions (three lobules per sample) with representative PSR-stained sections shown at baseline and post-treatment. The ellipse indicates fibre alignment: examples of aligned (baseline) and non-aligned (post-treatment) collagen are shown in the insets. The graph shows mean collagen coherency for $n = 22$ paired samples. Scale bars, 100 µm. **e**, Reduced modulus of peri-lobular regions at baseline (B) and post-treatment (PT) measured by AFM indentation. At least three 100 µm² regions per sample were measured as shown in the representative images. $n = 4$ tissue pairs. Scale bars, 100 µm. **f**, MRI annotation in ITK-snap: black denotes the background, opaque red indicates fatty tissue, and bright red shows the fibroglandular tissue. The FGV percentage was calculated by dividing the number of fibroglandular pixels by the total number of fibroglandular and fat pixels across slices. $n = 12$ paired MRI scans. Scale bars, 1 cm. **g**, Percentage of Ki67⁺ cells before treatment and post-treatment stratified by mammographic density. Participants were grouped using Volpara density grades to approximate BI-RADS categories (A/B denotes low MD, $n = 6$ tissue pairs; C/D indicates high MD, $n = 17$ tissue pairs). **h**, Heatmap of whole-tissue RNA-seq showing the differentially expressed genes between high MD (BI-RADS C/D; dark grey) and low MD (BI-RADS A/B; light grey) breast tissue at baseline ($n = 9$; FC > 3, $P < 0.05$). VST, variance-stabilizing transformation. **i**, Illustration shows that progesterone paracrine signalling regulates luminal progenitor/LASP (SOX9⁺) cells and fibroblasts, driving ECM remodelling and stiffness. Stiffness amplifies PR signalling, establishing a feedback loop. Anti-progestins disrupt this by inhibiting luminal cell-derived ligands (for example, WNT5A), lowering fibroblast collagen (for example, COL6A3), decreasing stiffness and reducing luminal progenitor/LASP cells. Boxplot centre lines represent median values and box bounds indicate the 25th and 75th percentiles (**d**–**g**), with connecting lines between paired data points (**d**,**f**,**g**) or whiskers denoting minimum and maximum values (**e**). *P* values were calculated with two-sided Wilcoxon matched-pairs test (**a**,**c**,**d**,**f**,**g**) or two-sided Student's *t*-test (**e**).

participants showed a significant decrease in collagen fibre alignment with UA treatment (Fig. 4d). Atomic force microscopy of four paired samples that had at least 10% reduction in collagen organization by PSR showed a consistent decrease in tissue stiffness, with three reaching statistical significance (Fig. 4e). We next investigated the available paired MRI data from 12 participants, revealing a significant reduction in FGV (a surrogate for mammographic density) with 3 months of UA treatment (Fig. 4f).

Recently, it has been established, using mouse models of elevated stiffness, that a stiff ECM increases luminal progenitor and stem cell frequency and tumour initiation by enhancing PR activation[42]. This study also reported an elevated number of luminal progenitor cells in women with high mammographic density, potentially providing mechanistic insights into the known positive association of breast cancer risk and mammographic density. In our study, we did not find significant correlations between baseline percentage of volumetric breast density (%VBD) as a continuous variable and baseline or fold change in any variable examined with UA treatment (data not shown). However, categorization of %VBD into Volpara density grades (1–4) to approximate Breast Imaging-Reporting and Data System (BI-RADS) 4th edition categories (A–D) demonstrated statistically significant reduction in %Ki67 in those with high but not low mammographic density (BIRADS C/D versus A/B; Fig. 4g). A similar pattern was observed in luminal progenitor/LASP cell frequency by flow cytometry, MFE and SOX9⁺ cell percentages (data not shown). In the RNA-seq dataset, gene expression was compared between participants similarly classified with high and low mammographic density. Five genes were strongly upregulated ($P < 0.05$; FC > 3) in women with high mammographic density: *TNFSF11*, *CXCL13*, *CUX2*, *TPSD1* and *ALOX15B* (Fig. 4h and Extended Data Fig. 12c). Increased *TNFSF11* and *CXCL13* gene expression, indicating PR signalling activation, correlated with VBD (Extended Data Fig. 12d). These results support the concept that women with high mammographic density and thus tissue stiffness-driven PR activity may derive greater benefit from anti-progestin therapy that remodels the mammary stroma and reduces the number of cancer-precursor luminal progenitor/LASP cells (Fig. 4i).

## Discussion

Here we have provided evidence that inhibiting progesterone signalling alters hallmarks of breast cancer risk. Progesterone contributes to breast cancer development through paracrine effects on the luminal progenitor/LASP cell fraction, the likely target of oncogenic transformation. Our BC-APPS1 trial demonstrates the short-term safety and efficacy of anti-progestin treatment in women at increased breast cancer risk. UA reduced FGV and epithelial cell density, lowering the proportion, proliferation and activity of the luminal progenitor/LASP population. Our work highlights the potential of FGV on MRI, and possibly mammographic density, as early biomarkers of anti-progestin response and suggests that women with high mammographic density are more likely to benefit from the reduction in luminal progenitor/LASP cell activity.

Mechanistically, PR antagonism remodels the ECM, reducing collagen organization and tissue stiffness, highlighting the importance of stromal–epithelial interactions in both luminal progenitor/LASP cell maintenance and breast density. We showed that the response to PR antagonism is related to baseline mammographic density, which has previously been linked to collagen abundance and organization[32]. Among ECM-related proteins, collagen VI was one of the most downregulated, linking epithelial cells with the ECM, including biophysical connectivity with collagen types I and IV and perlecan[43,44]. Alongside UA-induced effects in collagen, we uncovered a striking spatial association between collagen VI and SOX9^high luminal progenitor/LASP cells. Several luminal progenitor and stem cell markers (for example, CD49f (also known as integrin α6)) function as ECM receptors[45], underscoring the importance of stromal remodelling for luminal progenitor/LASP cellular dynamics.

Progestins are well known to contribute to both the cyclical expansion of the luminal progenitor/LASP cell pool and the development of human breast cancer[11,46]. As luminal progenitor/LASP cells have previously been shown to be more susceptible to DNA damage, treatment with anti-progestins is likely to counteract the mutational burden resulting from recurrent progesterone stimulation of the breast epithelium[47]. We and others have shown that UA and mifepristone treatments decrease epithelial proliferation and DNA methylation signatures of luminal progenitor cells and mitotic age[20,48]. Previous studies have established paracrine PR signalling via RANKL (*TNFSF11*) from the luminal mature cells to luminal progenitor cells, promoting their proliferation[4,6,7]. We revealed the broader impact of paracrine signalling in mediating the effects of progesterone on human breast tissue composition, showing that UA-induced changes in secreted ligands from luminal mature cells downregulated ECM genes in fibroblast and basal cell compartments. The response to UA did not show any correlation with age, breast cancer risk, parity or BMI. Importantly, controlling for menstrual cycle phase in BC-APPS1 overcame some of the hormonal complexity of human breast physiology, uncovering preventive vulnerabilities.

In summary, we have identified progesterone signalling as a key regulator of breast cancer-precursor luminal progenitor/LASP cells and established a complex interplay between anti-progestin treatment, ECM remodelling and luminal progenitor/LASP cell dynamics. Comparable in vitro effects with two anti-progestins, plus pre-existing clinical data on other PR antagonists[9,18], suggest that these may be class effects of PR antagonism. Longer-term studies are required to evaluate safety, particularly hepatotoxicity and effects on other hormone-sensitive tissues such as the endometrium, and to formally test whether PR antagonism reverses mammographic density-associated breast cancer risk. Mammographic density reporting is now mandated in all US states, albeit without recommendations on methods to reduce it. Our therapeutic cancer prevention trial in premenopausal women demonstrates the potential for PR antagonism to reduce mammographic density, tissue stiffness and luminal progenitor activity, which are important hallmarks of breast cancer risk.

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

## Methods

### The BC-APPS1 study

The BC-APPS1 was a single-arm, single-centre phase II study registered under the name 'a pilot prevention study of the effects of the anti-progestin ulipristal acetate (UA) on surrogate markers of breast cancer risk' (EudraCT registration number: 2015-001587-19; registration date: 15 July 2015; Greater Manchester-South, Research Ethics Committee number 15/NW/0478). Eligible women were premenopausal, 25–45 years of age with regular menses and a residual lifetime breast cancer risk of at least 17% (≥1:6) assessed by the Tyrer–Cuzick risk estimation programme (v7.02; https://ems-trials.org/riskevaluator/). All women were recruited from the Family History Risk and Prevention Clinic at the Nightingale Centre, Wythenshawe Hospital, Manchester, UK. Complete eligibility criteria are provided in the protocol (Supplementary Appendix 1). Following informed consent and screening, participants underwent blood testing to confirm serum progesterone levels consistent with the luteal phase of the menstrual cycle (15 nmol l$^{-1}$ or more; Abbott Architect Immunoassay) and then underwent a contrast-enhanced MRI scan of both breasts (Philips Achieva 1.5 T MRI). A baseline VAB was then performed by a consultant radiologist under ultrasound guidance to identify areas of fibroglandular breast tissue. On the onset of menstruation, participants commenced 5 mg oral tablets of commercially available UA, taken once daily for a duration of 12 weeks. During the final week, blood was drawn for progesterone levels, MRI of both breasts was repeated and VAB of the contralateral breast was performed. For each VAB, 10 cores were taken with a 10-G biopsy needle (cores were divided and fixed in formalin for paraffin embedding, snap frozen for RNA extraction and placed in tissue culture medium for subsequent digestion to single-cell suspensions). The primary end point was the change in epithelial cell proliferation measured by the percentage of Ki67 staining before and after treatment. Secondary end points were (1) percentage of luminal, basal and mixed colonies by morphological analysis of the adherent feeder layer assay; (2) percentage of luminal progenitor cells (EPCAM$^+$CD49f$^+$) by flow cytometry analysis; (3) tissue stiffness assessed as the reduced indentation modulus by atomic force microscopy; (4) mean tissue section percentage fibrillar collagen assessed by PSR staining and polarized light microscopy; (5) background parenchymal enhancement assessed by MRI; (6) the side effect profile of UA assessed by CTCAE (v4.03); and (7) the relative change in Ki67 with UA treatment between those with and without known mutation in *BRCA1/2* genes. Exploratory end points are included in the attached protocol and the methods described below. All participants had blood drawn for complete blood count, renal function and liver function tests at baseline and 3 months. Treatment with UA was suspended by the EMA/MHRA in February 2018 due to concerns of liver damage by UA. On reopening of the study after the suspension was lifted, the protocol was amended to include measurement of liver function tests every 4 weeks during treatment and a final check 4 weeks after the end of treatment. Toxicity assessment was undertaken every 4 weeks using CTCAE (v4.03).

### Other human breast tissue procurement

Normal breast tissue samples were collected from women at moderate or high risk undergoing surgical risk-reducing mastectomy at the Manchester University NHS Foundation Trust through the Manchester Cancer Research Centre Biobank. Fully informed consent from all patients was obtained in accordance with local National Research Ethics Service Guidelines, and the collection of demographic and clinical data was granted under the MCRC Biobank Research Tissue Bank Ethics (NHS NW Research Ethics Committee 18/NW/0092).

Frozen primary human female breast tissue was additionally sourced from the Breast Cancer Now Tissue Bank (REC 15/EE/0192), with all procedures conducted in compliance with applicable ethical regulations.

### Breast tissue processing

Normal breast tissue obtained via VAB (mean tissue weight of 1.22 g; 95% CI 1.06–1.37) was manually minced with a scalpel into small fragments (approximately 2 mm$^3$ pieces) and incubated in dissociation medium: phenol red-free DMEM/F12 with HEPES (Gibco) supplemented with 25% BSA Fraction V solution (Gibco), 1 mg ml$^{-1}$ collagenase/hyaluronidase (Stem Cell Technologies) and 5 µg ml$^{-1}$ insulin (Sigma). After overnight digestion at 37 °C with shaking at 100 rpm, the dissociated breast cell suspension was centrifuged at 450$g$ for 5 min at 4 °C. The fat layer was discarded and the epithelial pellet was resuspended in DMEM/F12 medium and centrifuged again. This wash step was repeated until the supernatant became clear. Then, 1 ml of pre-warmed 0.05% trypsin-EDTA was added to the enriched epithelial pellet, pipetting it up and down gently with a P1000 pipette for 2–3 min. Next, 10 ml of cold Hank's balanced salt solution (HBSS; Gibco) supplemented with 10 mM HEPES (Gibco) and 2% FBS (Gibco) was added and the cells were centrifuged at 450$g$ for 5 min at 4 °C. After removing the supernatant, 1 ml of pre-warmed 5 mg ml$^{-1}$ dispase (StemCell Technologies) was added to the sample and pipetted for 1 min to further dissociate cell clumps. Cells were resuspended in HBSS–HEPES–FBS solution, centrifuged and supernatant was discarded. The HBSS–HEPES–FBS solution was then added and cells were sieved using 100-µm and 40-µm filters to yield a single-cell suspension. Cells were counted using a Fuchs Rosenthal haemocytometer (mean cell yield of 1.39 million; 95% CI 0.99–1.79) and plated for experiments. Remaining cells were frozen using Bambanker freezing media (Lymphotec Inc.) until further analysis (for example, flow cytometry and scRNA-seq).

Normal breast tissue obtained via risk-reducing mastectomy was cut into 2–3 mm$^3$ pieces and digested overnight at 37 °C with collagenase IA (C2674, Sigma) and hyaluronidase (H3506, Sigma), both to a final concentration of 1 mg ml$^{-1}$, in phenol red-free DMEM/F-12 medium (Gibco). Following enzyme digestion, the breast tissue was washed three times with medium by centrifuging at 400$g$ for 10 min and discarding the supernatant. The pellet was resuspended in medium and left to sediment three times for 25 min at 4 °C on a flat surface. The collected breast organoids or microstructures from each tissue preparation were frozen in Bambanker freezing medium until experimental use.

### Measurement of progesterone (P4) levels

Serum progesterone levels were determined at baseline and after 3 months of anti-progestin treatment. Serum progesterone concentrations were measured by an NHS-accredited laboratory using the Abbott Architect immunoassay (Abbott Laboratories).

### Ki67 staining

Immunohistochemistry was performed using an automated Ventana medical system (BenchMark Ultra) using the UltraVIEW universal DAB detection kit (760–500, Roche). Slides were de-paraffinized under standardized conditions, blocked by endogenous biotin blocking kit (Ventana) and incubated for 32 min with Confirm Anti-Ki67 (30-9) antibody (Ventana 790–4286, Roche). Sections were counterstained with haematoxylin II and bluing reagent (Ventana 760–2021 and 760–2037), dehydrated and coverslipped. Slides were scanned using a Leica SCN400 slide scanner and visualized using Aperio ImageScope Digital Pathology Slide viewer (Leica Biosystems). A breast pathologist (S.P.) confirmed the percentage of Ki-67-positive epithelial cells by assessing a minimum of 1,000 epithelial cells per sample. Ki67 quantification was independently performed by a researcher fully blinded to participant number and timepoint using the HALO software, with Cohen's kappa score (0.62) confirming substantial agreement between the two assessments.

### Tissue morphometry

To investigate morphological alterations in normal breast tissues, haematoxylin and eosin (H&E) stained sections were scanned using

a Leica SCN400 slide scanner. Digital images were visualized using Aperio ImageScope Digital Pathology Slide viewer (Leica Biosystems) and the lobules and acini tissue compartments were manually annotated. The area of each lobule as well as the area of each acinus within the lobule was measured to calculate the ratio of acinar-to-lobular area. This analysis was performed blind to participant number and timepoint. The average acinar-to-lobular ratio of at least three lobules was calculated at the baseline and post-treatment for each participant.

## Flow cytometry analysis

Single-cell suspensions of digested breast tissue were incubated with primary antibodies for 10 min on ice according to the manufacturer's guidance. For endothelial and haematopoietic lineage depletion, CD31 (13-0319-82, eBioscience) and CD45 (304004, BioLegend) biotin-conjugated antibodies were used, followed by incubation with the secondary APC–Cy7 streptavidin-conjugated antibody (405208, BioLegend). To identify epithelial subpopulations, CD49f–APC (313616, BioLegend) and EPCAM–FITC (10109, StemCell Technologies) antibodies were used. Following each incubation, cells were washed with PBS, and at the end, they were resuspended with flow cytometry buffer containing HBSS (14025, Gibco), 10 mM HEPES (15630-056, Gibco), 2% FBS (Gibco) and 2 mM EDTA (Sigma). DAPI (422801, BioLegend) was added for dead cell exclusion. Data were acquired on a LSR II (BD) flow cytometer and analysed using the BD FACSDiva software.

## Mammosphere colony assay

Breast epithelial cells were plated in six-well plates treated with poly-(2-hydroxyethylmethacrylate) (poly-HEMA; Sigma) at a density of 1,000 cells per $cm^2$ following the protocol previously described[49]. Cells were grown in phenol red-free DMEM/F-12 medium with L-glutamine (Gibco), supplemented with B27 (Gibco) and 20 ng $ml^{-1}$ EGF (Sigma) at 37 °C in 5% $CO_2$. When indicated, cells were treated directly in the mammosphere medium with onapristone (100 nM; supplied by Astrazeneca) or UA (2 nM; Selleckchem). MFE was determined in six different wells per sample on days 10–12 and calculated by dividing the number of mammospheres formed (diameter of 50 µm or more) by the original number of single cells seeded.

## 2D human mammary colony-forming assay

Single epithelial cells were cultured in adherence in Human EpiCult-B media (StemCell Technologies) supplemented with 5% FBS (Gibco), 0.48 µg $ml^{-1}$ hydrocortisone (Sigma) and 2 mM L-glutamine (Gibco). For each participant, sample, cells were plated at 400 cells per $cm^2$ in 60-mm culture dishes. Irradiated NIH 3T3 feeder cells (50,000 cells per ml; 50 Gy) were added to each plate. Three separate culture dishes per condition were plated and incubated at 37 °C in 5% $CO_2$ for 10–12 days. Cells were fixed with acetone:methanol (1:1), air dried, rinsed with distilled water and stained with Giemsa (Sigma) for 2–3 min. Colonies were defined as discrete clusters of 50 or more cells and classified according to established morphological criteria. All three colony types could be observed on the same plate: luminal colonies appeared as tightly packed, cobblestone-like clusters with smooth, well-defined edges; myoepithelial colonies consisted of dispersed, teardrop-shaped, spindle-like cells with visible gaps between them; and mixed colonies displayed features of both, with irregular, non-uniform edges[33].

## Immunofluorescence staining

Four-micrometre-thick formalin-fixed, paraffin-embedded (FFPE) sections of breast tissue were mounted onto Superfrost Plus microscope slides (9951APLUS, Thermo Fisher) and dried in a 60 °C oven for 1 h. Slides were placed in a Leica-Bond-RX Immuno-stainer and stained using BOND Research Detection System (DS9455, Leica). SOX9 (AB5535, Millipore) and Ki67 (M7240, Dako) antibodies were diluted in Bond primary antibody diluent (ARD1001EA, Leica). Opal520 (FP1487001KT, Leica) and Opal650 (FP1496001KT, Leica) fluorophores were used to mark SOX9[+] and Ki67[+] cells, respectively. The Envision+ System HRP-labelled polymer anti-mouse (K4001, Dako; Ki67) or anti-rabbit (K4003, Dako; SOX9) was used to amplify the signal. The following steps were performed automatically for each antibody: 10% peroxidase block for 10 min, primary antibody for 30 min, the Envision+ System for 30 min, Opal fluorophore for 10 min, and Bond ER 1 buffer for 20 min at 95 °C. Spectral DAPI (FP1490, Leica) was used to mark the nuclei and slides were mounted with Prolong Gold (P36930, Thermo Fisher). Slides were scanned with an Aperio VERSA scanner and images were analysed with HALO Image Analysis Software.

## PSR staining and polarized light microscopy

Breast tissue slides were stained with PSR as previously described[32]. PSR images were captured using the Leica MC190 HD Camera on the Leica DM2500 microscope with the ×10 (0.22 aperture) objective (for bright field, exposure of 70 ms and gain of 1; for polarised light, exposure of 250 ms and gain of 2). Polarizer was set at 102.5° based on the lowest background obtained. Quantitative analysis of collagen-associated birefringence as a measure of collagen orientation (coherency) was conducted on 22 PSR-stained lobule pairs, pre-treatment and post-treatment, using the 'Orientation J' plugin[50]. For each lobule image taken, 11–30 regions were selected from the surrounding tissue depending on size. The software was then used in the 'measure' mode as previously described to obtain coherency scores relating to the alignment of collagen fibres in each region[51]. The average coherency scores from at least three lobules were then calculated before and after treatment for each participant.

## MRI

MRI scans taken from baseline and 3-month visits were analysed to investigate the change in FGV following a 3-month course of UA. FGV was defined as the volume of annotated fibroglandular tissue divided by the total volume of the breast. The latter was computed from the sum of the volumes of fibroglandular and fatty tissues. Eighteen individuals were identified with available scans, and 12 were included in the analysis. Individuals were excluded if their scans were truncated, or if one of their baseline or 3-month scans was missing. An automated segmentation method was used to initially annotate each slice of the MRI scans into segments of background, fatty tissue or fibroglandular tissue. The software ITK-SNP[52] was then used to manually correct the annotations. Tissue between and behind the breasts, and above and below the breasts was classed as background. Python scripts were written and used to calculate the number of pixels in each annotated slice. From this, breast density was calculated for each MRI dataset by dividing the number of fibroglandular pixels in all slices by the total number of fibroglandular and fat pixels in all slices. The change in breast density over 3 months was plotted for all participants, and the *P* value was calculated using the Wilcoxon matched-pairs signed-rank test.

## Mammographic density analysis

The percentage dense volume (PDV) was automatically assessed from the raw data files of all four exposures (craniocaudal and mediolateral oblique for both breasts) of mammograms taken within 1 year of study entry (median of 2.5 months; range of 0–11), using Volpara density (v1.5.0; Matakina). Volpara density grades (VDGs) were determined using cut-offs representative of BIRADS 4th edition (VDG1/BIRADS A: 0% ≤ PDV < 4.5%, VDG2/BIRADS B: 4.5% ≤ PDV < 7.5%, VDG3/BIRADS C: 7.5% ≤ PDV < 15.5% and VDG4/BIRADS D: PDV ≥ 15.5%).

## Tissue stiffness by atomic force microscopy

Seven-micrometre-thick sequential cryosections were obtained for each participant sample. Three peri-lobular regions (100 × 100 µm) were identified for each participant sample from a H&E-stained section, and the same region was then located on an unstained sequential slice to be probed. Immediately before the experiment, each

participant sample was allowed to thaw and dry at room temperature for 2 h, followed by five quick washes in deionized water to remove the optimal cutting temperature (OCT) compound. Atomic force microscopy was conducted using the Peakforce Quantitative Nanomechanics mode in fluid on the BioScope Resolve AFM (Bruker), with a cantilever of spring constant 0.14–0.16 N m$^{-1}$ that is attached with a gold spherical probe of 5 μm diameter (CONT-Silicon-SPM-Sensor with colloidal particle, sQUBE). Calibration of probe deflection sensitivity and spring constant were conducted for every participant sample. To obtain the force curve, the cantilever was indented to a depth of 50 nm into the sample at a rate of 6 μm s$^{-1}$, and the reduced elastic modulus of the sample was estimated based on the Hertzian model (spherical), taking 75% and 25% of maximum force on the force curve as fitting boundary. The results of this study were kept as reduced modulus with no assumption made on the Poisson's ratio. A total of 400 force curves were obtained evenly throughout each $100 \times 100$ μm peri-lobular region to give a total of 1,200 force curves per participant sample ($3 \times 400$). The force curves were filtered to remove those that poorly fit the Hertzian model ($r^2 < 0.95$), followed by values that lie outside 2 standard deviations from the remaining population. The final population of force curves were subjected to a $t$-test to determine any significant differences in the reduced modulus between baseline and treated participant samples.

## Tissue bulk RNA-seq analysis

Only good-quality RNA samples (RNA integrity number (RIN) > 8) were considered adequate for library preparation. Indexed PolyA libraries were prepared using 100 ng of total RNA and 15 cycles of amplification in the Agilent Sure Select Strand Specific RNA Library Prep Kit for Illumina Sequencing (G9691B, Agilent). Libraries were quantified by quantitative PCR using a Kapa Library Quantification Kit for Illumina sequencing platforms (KK4873, Kapa Biosystems). Paired-end 75-bp sequencing was carried out by clustering 2.0 pM of the pooled libraries on a NextSeq 500 sequencer (Illumina).

The fastq files were processed with Nextflow (v19.10.0), nf-core/rnaseq (v1.3) and aligned using GRCh38 as reference. Samples with low base quality or genomic DNA contamination were excluded from the analysis. Additional gene identifiers (gene symbols, Entrez ID, gene type) were retrieved from Ensembl BioMart (v101; https://www.ensembl.org/biomart/martview/). Differentially expressed gene analysis was performed with DESeq2 (v1.26.0) with a multifactor design formula that accounted for treatment and participant ID. A heatmap was generated with ComplexHeatmap (v2.16.0) and principal component analysis was performed using variance-stabilizing transformation values from DESeq2, stats (v3.6.0) and SummarisedExperiment (v1.16.1). Gene Ontology terms were downloaded from Ensembl BioMart (https://www.ensembl.org/biomart/martview/) using Ensembl Genes 103 and Human Genes GRCh38.p13, selecting the attributes 'Gene stable ID', 'Gene name' and 'GO term name'. For data access, please refer to 'Data availability' section.

## LCM and mass spectrometry

Five-micrometre-thick FFPE sections from four participants (BAP06, BAP12, BAP17 and BAP31) were mounted onto either glass slides or 'MMI MembraneSlides' (Molecular Machines & Industries), then stained with H&E using a Leica ST5010 autostainer XL. From tissue sections mounted on MMI slides, regions of 'lobular epithelium and peri-lobular stroma' were dissected using a MMI CellCut Laser Microdissection system across multiple sections. Tissue areas of 10 mm$^2$ were collected using MMI transparent isolation caps and pooled to a final volume of 0.05 mm$^3$. Formalin-mediated protein crosslinking was reversed by resuspending the dissected tissue in 50 mM triethyl ammonium bicarbonate (TEAB) containing 5% SDS (w/v) and heating at 95 °C for 20 min, then 60 °C for 2 h. To assist the solubilization of ECM proteins, urea and dithiothreitol was added to the samples to a final concentration of 8 M and 5 mM, respectively. Samples were then sonicated in a LE220-Plus focused ultrasonicator (Covaris) for 10 min. Samples were reduced and alkylated, then proteins were isolated and digested with trypsin using S-Trap spin columns (Protifi) as per the manufacturer's instructions. Peptides were desalted using POROS Oligo R3 beads (Thermo Fisher) and analysed by liquid chromatography–tandem mass spectrometry using an UltiMate 3000 Rapid Separation LC system (RSLC, Dionex Corporation) coupled to a Q Exactive HF mass spectrometer (Thermo Fisher) for 90 min.

## Proteomics data analysis

Raw mass spectra were processed in MaxQuant (v1.6.14.0, available from Max Planck Institute of Biochemistry)[53]. Features were identified using default parameters, then searched against the UniProt human proteome reference database (UP000005640, August 2020). Oxidation of methionine, hydroxylation of proline and acetylation (protein N terminus) were set as variable peptide modification for protein identification, whereas carbamidomethylation of cystine was set as a fixed modification. Peptide quantification was performed using label-free quantification, using only unmodified, unique peptides and with 'match between runs' enabled. Statistical analysis of quantitative proteomics data was performed using MSqRob[54]. Label-free quantification intensities were normalized between samples by means of the median of peptide intensities. Timepoint (pre-treatment or post-treatment) was treated as a fixed effect, whereas peptide sequence and participant (biological replicate) were treated as random effects. Peptides belonging to contaminant protein lists (as annotated by MaxQuant/Andromeda), or proteins with fewer than two peptides were excluded from the analysis. For annotation of proteins ECM or non-ECM status, the protein tables generated by MaxQuant or MSqRob were screened for ECM core or affiliated components by comparison with MatrisomeDB, a curated database of ECM proteins[55,56]. Ontology analysis was performed on differential protein abundance data generated by MSqROB using clusterProfiler (v4.6.0)[57] gene set enrichment analysis, applying Benjamini–Hochberg correction and grouping proteins by 'Reactome pathway' annotations (v.65)[58].

## scRNA-seq library preparation and data processing

Single-cell suspensions of human breast cells were generated as described above and scRNA-seq was performed using the standard workflow for the 10X Genomics Single Cell 3′ RNA Kit V3 chemistry, in two batches for six participants. Batch 1 ($n = 6$) and batch 2 ($n = 6$) contained pre-treatment and post-treatment ulipristal acetate samples for three different participants. Next-generation sequencing was performed with the NovaSeq 6000 (Illumina) at the Cancer Research UK Cambridge Institute Genomics core. Raw sequencing reads were aligned using CellRanger (v3.02) using the GRCh38 human genome as reference. Quality control was performed using DropletUtils (v1.10.3) to detect empty droplets, and cells with less than 1,000 unique molecular identifier counts or more than 10% mitochondrial genes were filtered out using scuttle (v1.18.6). Gene symbols were mapped to Ensembl IDs using org.Hs.eg.db (v3.12.0). Batches were merged using batchelor (v1.6.3) and the batch-corrected dimension was used to build the shared nearest neighbour graph (scran v1.18.7), and batch integration was visually evaluated using igraph (v1.2.6). Scater (v1.18.6) was used to generate both principal component analysis and UMAP dimensionality reduction coordinates, with a minimum distance of 0.1 and nearest neighbours of 30 cells. Cell clusters were generated computationally with R package igraph, and cell-type assignment was performed manually using previously described mammary subtype canonical markers[35]. The secondary cell subclustering to match the 'level 2' annotation of the iHBCA was completed in Python (v3.10.13) using the Leiden algorithm through the scanpy package (v1.9.6)[59]. Differential abundance testing was performed using the propeller function from the Speckle R package[60].

## Memento differential expression analysis

Pairwise (post-treatment versus baseline) differential expression analysis of participant samples was performed using Memento (v0.1.0)[61] in Python (v3.9). The sequencing capture rate was estimated at 0.7 and Memento analysis was setup using a filter_mean_tresh and trim_percent of 10% to ensure robust expression of each gene carried to downstream analysis. Moments were computed using compute_1d_moments with min_perc_group of 50% ensuring at least half of these samples had sufficient expression of a gene while accounting for the potential of UA-mediated effects in 6 of 12 samples. Ht_1d_moments was used to compare baseline and post-treatment within each cell population with sample ID as a covariate. Memento pairwise differentially expressed results are summarized in Supplementary Table 4, and gene expression patterns were visualized with UpSet plots (UpsetR v1.4.0) with up to top 40 intersections shown.

## Single-cell pathway analysis

Pathway analysis was performed using gprofiler2 (v0.2.3), which selects pathways of interest based on gene overrepresentation. Analysis was restricted to Reactome pathways, corrected by false discovery rate, gene count within a pathway 5 or more, and an adjusted $P < 0.01$. The top 10 upregulated and downregulated pathways are presented in barplot visualizations.

## Cell–cell communication analyses

Ligand–receptor communication networks were first assessed using Cell Chat (v2.1.2)[36] to identify broad communication network alterations between treatment groups. To infer a 'per cell' contribution to cell–cell communication, each broad cell population was downsampled to 985 cells per treatment group (baseline and post-UA treatment). Communication probability between cell types was then assessed using a 3% truncated mean with a minimum of ten cells per group. Differential cell–cell communication was compared before or after UA treatment using default settings in the CellChat R package. This analysis was similarly repeated downsampling iHBCA-annotated BC-APPS1 data to 300 cells per subcluster to ensure coverage of each subcluster. Inference of downstream ligand–collagen target gene interactions was performed using NicheNet (v2.2.0)[37]. Differentially expressed LHS ligands (pairwise logFC ≤ −0.4 (de_coef) and $P < 0.05$ (de_pval)) were selected as input. Intersection of pairwise differentially expressed gene list with those annotated as 'ligand' in the NicheNet (v2.2.0) L–R network database ('lr_network_human_21122021.rds') facilitated identification of downregulated LHS ligands. Similarly, downregulated FB1–3 and BMYO1 targets genes were identified from Memento results (Supplementary Table 4, pairwise logFC ≤ −0.25 (de_coef) and $P < 0.05$ (de_pval)). These downregulated target genes were intersected with a collagen gene list to identify which collagens within each subcluster were candidate target genes. A liberal minimum threshold of 3% receptor expression in receiver cells was used, and no differential receptor expression criteria was applied, given that co-regulation of ligand and receptor is not essential for target gene regulation. NicheNet analyses were performed using all genes as background and no downsampling was used in this analysis.

## Imaging mass cytometry

For eight participants (02, 06, 12, 14, 17, 22, 24 and 31), 5-µm sections from FFPE tissue blocks were mounted onto a single glass slide. Sections were dewaxed with two sequential 10-min xylene incubations, then rehydrated in a graded series of ethanol in water (100%, 95%, 80%, 70% and 0%; 10 min each). Antigen retrieval was performed by incubating rehydrated sections in 'Target Retrieval Solution, pH 9' (S236784-2, Dako) for 20 min at 95 °C in a steam producing water bath, before removing the heat source and allowing the water bath to cool to room temperature over 20 min. Sections were washed three times in PBS (Maxpar PBS;

201058, Standard BioTools), then blocked in 3% BSA and 0.1% Triton X-100 in PBS for 1 h at room temperature. A cocktail of metal-conjugated antibodies and 0.5% BSA in PBS was made up and added to sections, incubating overnight at 4 °C. The antibodies used included α-smooth muscle actin (clone 1A4, 141Pr), E-cadherin (clone 24E10, 158Gd), Ki67 (clone B56, 168Er) and collagen I (polyclonal, 169Tm) from Standard BioTools (201508, Maxpar Human Immuno-oncology IMC panel kit), as well as fibronectin (polyclonal, 149Sm; ab23750, Abcam), collagen VI (polyclonal, 160Gd; ab6588, Abcam) and SOX9 (clone EPR14335, 147Sm; 3147022D, Standard BioTools). Fibronectin and collagen VI carrier-free antibodies were conjugated to metal isotopes using MaxPar X8 antibody labelling kits (149Sm, 201149 A; 160Gd, 201160B, Standard BioTools) as per the manufacturer's instructions. Following antibody incubation, sections were washed twice in 0.2% Triton X-100 in PBS and then twice in PBS. Nucleic acids were then stained by incubating section with Cell-ID Intercalator (201192A, Standard BioTools) for 30 min, then washed in ddH$_2$O and allowed to air dry.

Raw imaging mass cytometry data were acquired by ablating sections using a Hyperion Imaging System (Standard BioTools). For each slide, the histology of sections was inspected using the inbuilt light microscope and regions of interest were selected that contained lobular tissue. Approximately 1 mm$^2$ of tissue from at least three discrete regions of interest were ablated in a rastered pattern at 200 Hz per section. To ensure that the order of section acquisition did not create artefacts in the data, slides were randomly allocated to two equally sized groups, one group in which ablation was ordered baseline then post-treatment and a second group in which ablation was performed post-treatment then baseline. Raw data files (TXT) were converted to TIFF format and segmented into single-cell objects using the Steinbock pipeline[62]. Hot pixels were filtered using a threshold value of '50', whereas single-cell objects were segmented using the 'Deepcell, cell segmentation' function. Object intensity values were exported as CSV files and passed to the single-cell neighbourhood analysis pipeline. Selection of cells of interest and proximal ECM quantification was performed in MATLAB (please refer to the 'Code availability' section). In brief, after image noise removal, each channel was normalized between 0 and 1. Then, epithelium and void tissue regions were identified and masks for individual cells were generated. Cells of interest were then defined as those where mean intensity for a particular channel (for example, 'SOX9') is higher than a given threshold value. To define threshold values, for each image, an initial analyst, blinded to slide identity and referencing a common reference image, selected a pixel intensity threshold to captured cells exhibiting comparable staining intensity and cellular distribution. Thresholding was confirmed by a subsequent, independent analysis. To define SOX9$^{low}$ cells, SOX9$^{high}$ cells were first defined per image, then the equivalent number of the lowest expressing cells per image were classified as SOX9$^{low}$. Changing the SOX9$^{high}$ and SOX9$^{low}$ threshold analysis to either a fixed top or bottom 20% expression or a fixed-intensity value produced similar results to those reported (data not shown). Around each cell of interest, a circular region of interest of a given radius was defined in such a way that it did not include any neighbouring cells and the void regions, and mean intensity within the circular region of interest was calculated for each channel.

## VitroGel assay

Normal breast tissue organoids (3D microstructures) obtained from women at moderate or high risk of breast cancer were plated in 12-well plates in the presence of a polysaccharide-based synthetic hydrogel modified with the collagen-mimetic peptide GFOGER (VitroGel, TWG009, The Well Biosciences) as per the manufacturer's instructions. Organoids were cultured with VitroGel in DMEM/F-12 medium (Gibco) with 8% charcoal-stripped FBS (Gibco), 10 µg ml$^{-1}$ insulin (Sigma-Aldrich), 10 µg ml$^{-1}$ hydrocortisone (Sigma-Aldrich) and 5 ng ml$^{-1}$ EGF (Sigma-Aldrich). Using VitroGel dilution solution, the

hydrogel was used in a ratio of 1:4 and 1:2, which represents soft (600–900 Pa, Young's modulus) and stiff (1,800–3,000 Pa, Young's modulus) conditions, respectively. Organoids were treated with onapristone (100 nM) or DMSO (control) every 72 h. After 1-week incubation at 37 °C, cells were isolated from the hydrogel using VitroGel Cell Recovery Solution (MS03-100, The Well Biosciences) following the manufacturer's recommendations. Cells were counted before undergoing further analysis: mammosphere colony assay and RNA or protein extraction.

## Real-time PCR

RNA was extracted using the RNeasy Plus Micro Kit (74034, Qiagen) as described in the manufacturer's handbook. RNA quality was checked using a NanoDrop One spectrophotometer (Thermo Scientific). RNA reverse transcription to synthesize cDNA was done by using Omniscript RT kit (205111, Qiagen). Of each cDNA sample, 100 ng was then used in triplicate to perform quantitative PCR with Taqman Universal PCR master mix (4304437, Applied Biosystems) and Thermo Fisher gene expression assays for *TNFSF11* (Hs00243522_m1), *SOX9* (Hs00165814_m1) and *KIT* (Hs00174029_m1). Housekeeping gene, encoding β-actin, expression assay (Hs99999903_m1) was used to normalize the gene expression. Gene expression was measured using the QuantStudio 5 Real-time PCR system (Thermo Fisher) and analysed using QuantStudio Design and Analysis software (Desktop; v2.6.0).

## Western blot

Protein lysates were fractionated by SDS–PAGE and transferred to nitrocellulose membranes (Protran BA85, Whatman). The membranes were incubated in Tris-buffered saline containing 0.1% Tween 20 and 5% milk for 1 h at room temperature to block nonspecific antibody binding, and then probed with primary and secondary antibodies to identify the proteins of interest. The primary antibodies used were 1:1,000 dilution of anti-SOX9 rabbit poly-antibody (AB5535, Sigma), 1:1,000 dilution of anti-KIT mouse poly-antibody (MAB332, R&D Biosystems) and 1:5,000 dilution of anti-β-actin mouse monoclonal antibody (A1978, Sigma). Horseradish peroxidase-conjugated secondary antibodies of goat anti-rabbit (41424306, Dako) and anti-mouse (41424131, Dako) were used at a 1:5,000 dilution and proteins were detected by horseradish peroxidase reagents Classico, Forte (Millipore) or West Femto (Thermo Fisher). Proteins were visualized by acquiring digital images with ChemiDoc Touch Imaging System (Bio-Rad). Densitometry was carried out on the digital images using Image Lab (v6.1; BioRad) and proteins of interest were normalized to β-actin.

## Statistical analyses

If not stated otherwise, *P* values were generated using the 'stat_compare_means' function from the 'ggpubr' package (v0.6.0), applying the wilcox.test method. This performs either a Wilcoxon signed-rank test for paired samples or a Wilcoxon rank-sum test for independent samples. Exact *P* values are reported when there are no tied values. When ties are present, an approximate *P* value was calculated with a correction for ties. A value of $P < 0.05$ was considered to be statistically significant. Data are shown as median and interquartile range with connecting lines between paired data points, unless otherwise indicated.

## Ethics and inclusion statement

This research has included local researchers throughout: in the study design, study implementation, data ownership and authorship of publications. The clinical study was approved by the local ethics review committee.

## Reporting summary

Further information on research design is available in the Nature Portfolio Reporting Summary linked to this article.

## Data availability

All bulk RNA-seq and scRNA-seq data have been deposited in the Array Express database (https://www.ebi.ac.uk/biostudies/arrayexpress) and can be retrieved by the following access IDs: E-MTAB-13720 (bulk RNA-seq) and E-MTAB-13819 (scRNA-seq). The mass spectrometry proteomics data have been deposited to the ProteomeXchange Consortium via the PRIDE partner repository (https://www.ebi.ac.uk/pride/) with the dataset identifier PXD067122.

## Code availability

The scripts used to analyse the tissue bulk RNA-seq, the scRNA-seq and the imaging mass cytometry data are available on Zenodo[63] (https://zenodo.org/records/11369094).

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

**Acknowledgements** We thank all the participants who took part in the study, as well as the clinical research teams at the Manchester University NHS Foundation Trust and the Wolfson Molecular Imaging Centre for their support. Funding for the study was from a Breast Cancer Now Project Grant (2014MayCR003). This study has been delivered through the National Institute for Health and Care Research (NIHR) Manchester Biomedical Research Centre (BRC; NIHR203308). The views expressed are those of the authors and not necessarily those of Breast Cancer Now, the NIHR or the Department of Health and Social Care. B.M.S., S.M.A., E.F.H., D.G.E., A.P.G., R.B.C. and S.J.H. were supported by the Manchester NIHR BRC (NIHR203308 and IS-BRC-1215-20007). B.M.S. was also funded by The Christie Charity. A.D.R. and A.-J.T. were funded by the CRUK Programme Foundation Award (DCRPGF\100010) and the Wellcome LEAP Delta Tissue Programme to W.T.K. A.G. and S.C. were supported by UKRI Doctoral Training Programmes, funded by the BBSRC and EPSRC, respectively. A.J.W. and H.H. were supported by the Prevent Breast Cancer charity. A.S.-G. and K.S. were supported by the Breast Cancer Now charity. R.K. was funded by Breast Cancer Canada and Terry Fox Research Institute. C.W.M. was supported by a CRS Next Generation Scientist Award and a CIHR Banting Canada Postdoctoral Fellowship. R.J.T. was awarded a studentship from the National University of Singapore (NUS) Singapore Nuclear Research and Safety Initiative, which was supervised by M.J.S. The Manchester Cell-Matrix Centre is funded through the Wellcome Trust Discovery Research Platform for Cell-Matrix Biology (226804/Z/22/Z). We are grateful to G. Ashton and C. Behan (histology) and A. Banyard (flow cytometry) from CR-UK Manchester Institute core facilities, as well as to N. Hodson and S. Marsden (University of Manchester Bioimaging Facility) for their help with the AFM experiments.

**Author contributions** S.J.H., A.H., D.G.E., W.T.K. and R.B.C. designed the study. Participant recruitment and clinical management were done by S.J.H., A.H., D.G.E. and J.R. Clinical pathology was performed by S.P. Clinical radiology and biopsy were done by Y.L. Tissue digestion/processing and luminal progenitor/colony formation assays were performed by B.M.S., A.S.-G. and K.S. Immunohistochemical and immunofluorescence analyses were conducted by B.M.S., A.C., A.J.W., A.G., S.Alghamdi, S.P. and S.Andò. RNA-seq analysis was

done by B.M.S., S.Alghamdi and M.R. Single-cell RNA-seq was performed by B.M.S., C.W.M., M.R., A.D.R., A.-J.T. and W.T.K. MRI scan analyses were done by S.M.A., E.F.H. and S.S. Proteomics and imaging mass cytometry were conducted by R.P., Y.Z., M.A.K. and A.P.G. Hydrogel assays were performed by B.M.S., A.C., S.C., H.H., M.F. and J.L. Tissue composition and stiffness were done by B.M.S., R.J.T., J.R. and M.J.S. The manuscript was drafted by B.M.S. and S.J.H., and edited and redrafted by C.W.M., R.K., P.T. and R.B.C. All authors revised the manuscript and approved the final version.

**Competing interests** The authors declare no competing interests.

**Additional information**

**Correspondence and requests for materials** should be addressed to Bruno M. Simões, Robert B. Clarke or Sacha J. Howell.

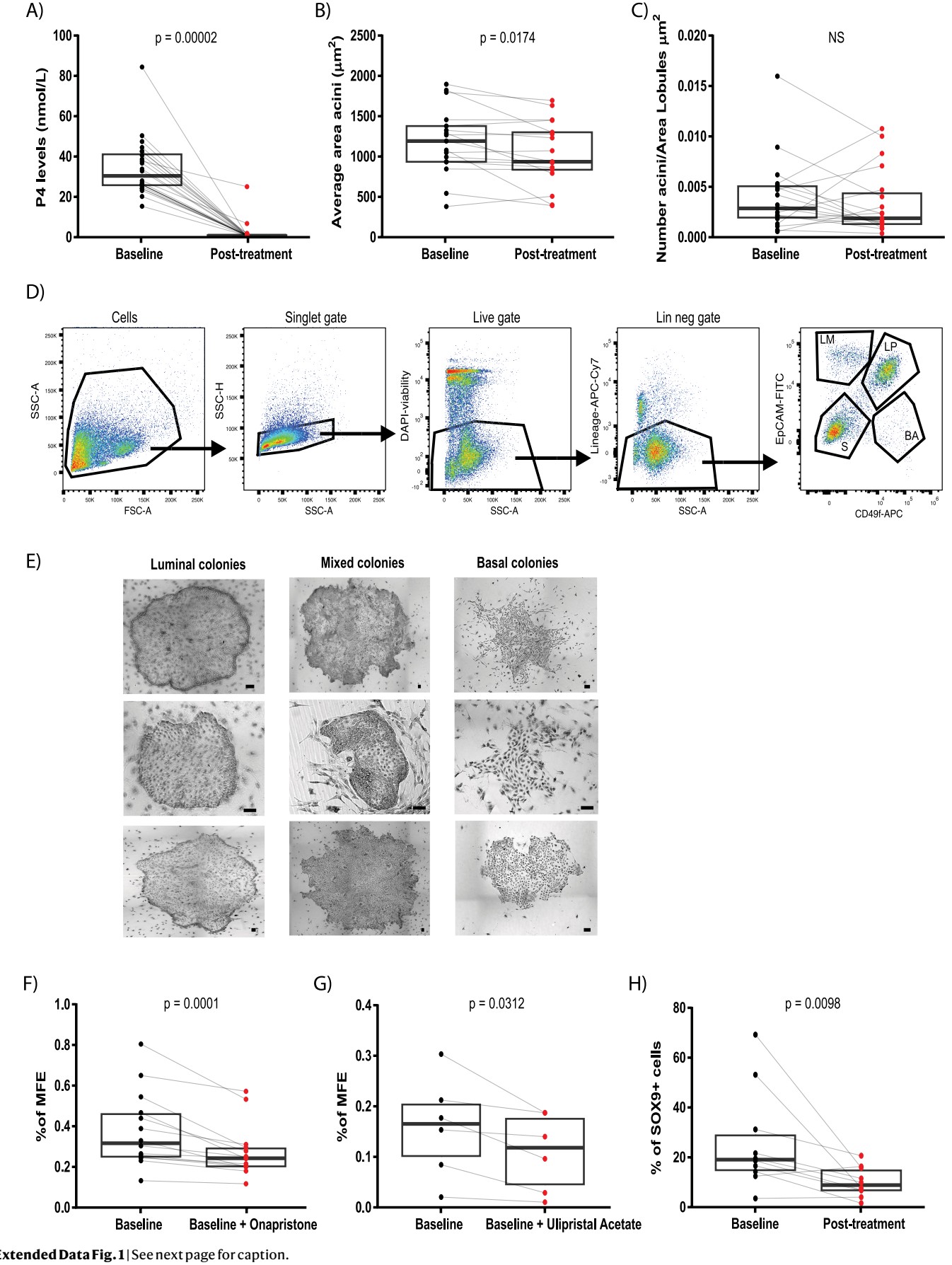

**Extended Data Fig. 1** | See next page for caption.

**Extended Data Fig. 1 | Anti-progestin treatment suppresses serum progesterone and reduces LP cell activity and frequency.** A) Measurement of serum progesterone levels before (baseline) and after (post-treatment) 3 months of ulipristal acetate (N = 24 pairs). B) Average area of acinar structures within lobules before (baseline) and after (post-treatment) 3 months of UA therapy (N = 17 tissue pairs). C) Number of acinar structures per area ($\mu m^2$) of lobules before (baseline) and after (post-treatment) 3 months of UA therapy (N = 19 tissue pairs). D) Gating strategy used to assess breast epithelial cell populations via flow cytometry. Live cell-sized singlets were selected from samples lineage depleted and stained with EpCAM and CD49f antibodies. E) Representative examples of luminal, mixed and basal colonies formed in the clonogenic assay are shown. Scale bar = 50 $\mu m$. N = 18 tissue pairs. F) Mammosphere formation efficiency (MFE %) data for baseline cell suspensions in the presence of onapristone (100 nM) versus control (DMSO). N = 14 baseline tissue. G) Mammosphere formation efficiency (MFE %) data for baseline cell suspensions in the presence of ulipristal acetate (2 nM) versus control (ethanol). N = 6 baseline tissue. H) Percentage of SOX9+ cells in pairs of tissue quantified by immunofluorescence. N = 10 tissue pairs. Box plots centre lines represent median values and box bounds indicate the 25th and 75th percentiles, with connecting lines between paired data points; p values are calculated with two-sided Wilcoxon matched-pairs signed rank test (A-C, F-H).

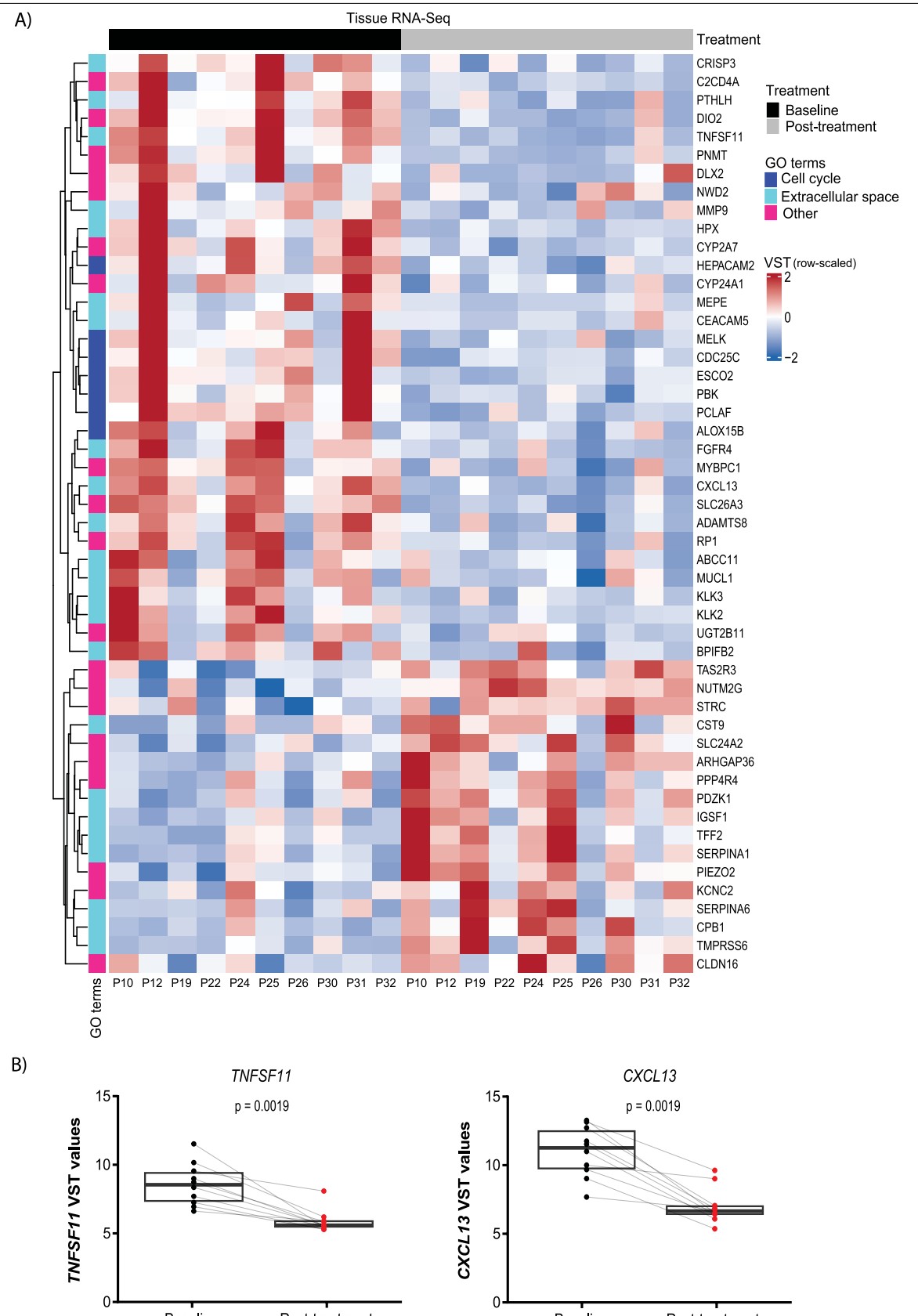

**Extended Data Fig. 2 | UA-induced transcriptional changes include down-regulation of PR target genes.** A) Heatmap from whole tissue RNA-Seq gene expression analysis showing the top 50 differentially expressed genes before (baseline) and after (post-treatment) UA therapy in 10 pairs of breast tissue (log2FC > 1.5, padj <0.05). Gene ontology (GO) term analysis shows genes associated with cell cycle, extracellular space and other GO terms. VST - Variance Stabilising Transformation. B) Gene expression of PR-regulated genes - *TNFSF11* and *CXCL13* - before (baseline) and after (post-treatment) UA therapy. Box plots centre lines represent median values and box bounds indicate the 25th and 75th percentiles, with connecting lines between paired data points; p values are calculated with two-sided Wilcoxon matched-pairs signed rank test. N = 10 tissue pairs.

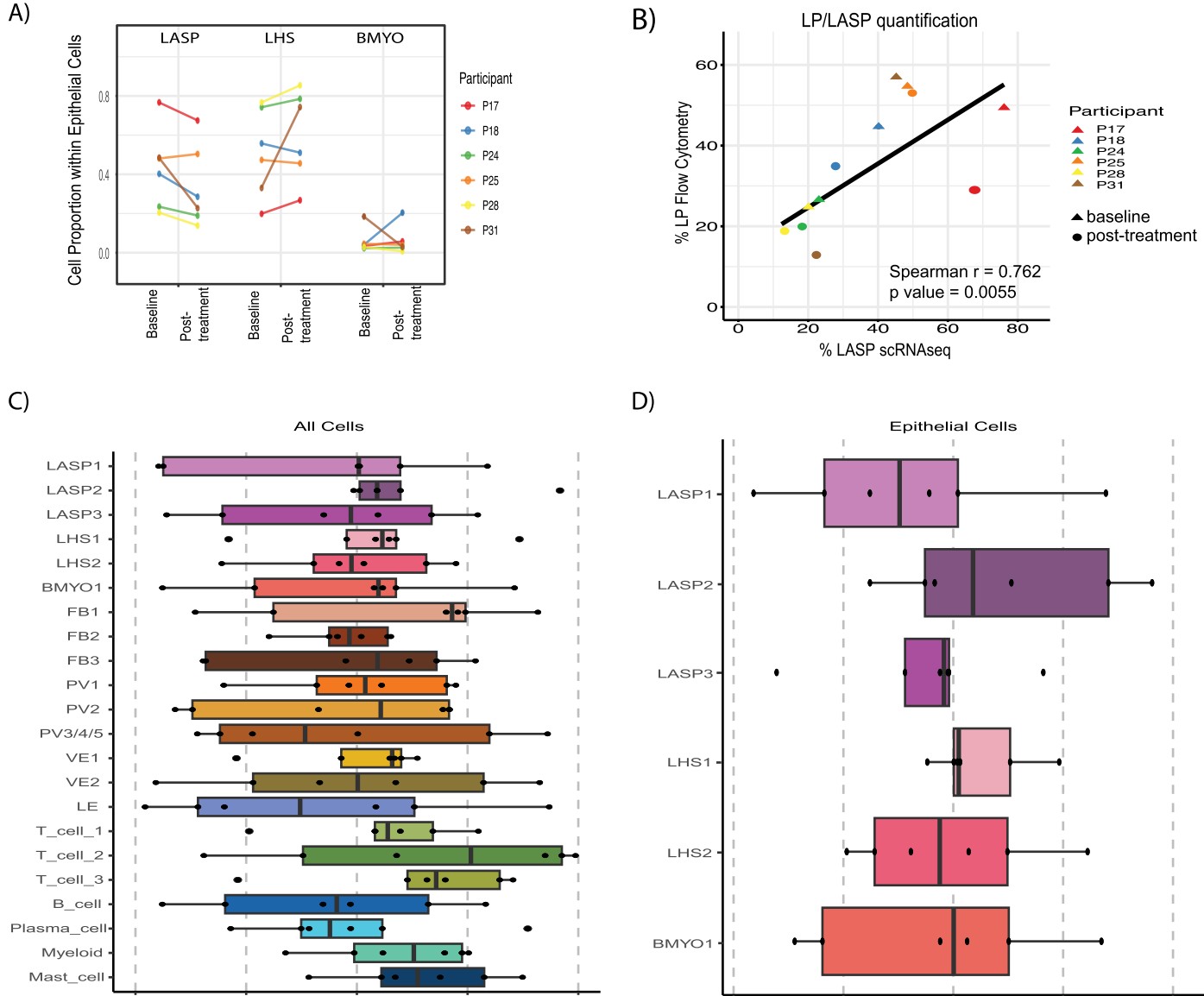

**Extended Data Fig. 3 | UA treatment reduces LASP/LP cell proportion with concordance between scRNAseq and flow cytometry.** A) Percentage of LASP, LHS and BMYO cell populations determined by scRNAseq in 6 pairs of tissue before (baseline) and after (post-treatment) UA therapy. B) Correlation (two-sided test) between the percentage of LP cells determined by flow cytometry and LASP cells determined by scRNAseq. N = 6 tissue pairs. C-D) Cell subcluster proportionality fold change (post-treatment / baseline) across all subcluster cell types (C) and with proportionality restricted to epithelial cell types (D). Positive (negative) changes denote subcluster enrichment (depletion) in samples after UA therapy. Box plots centre lines represent median values, box bounds indicate the 25th and 75th percentiles, and whiskers extend to the most extreme datapoint within 1.5 × IQR (inter-quartile range) of the outer hinge of the boxplot. N = 6 tissue pairs.

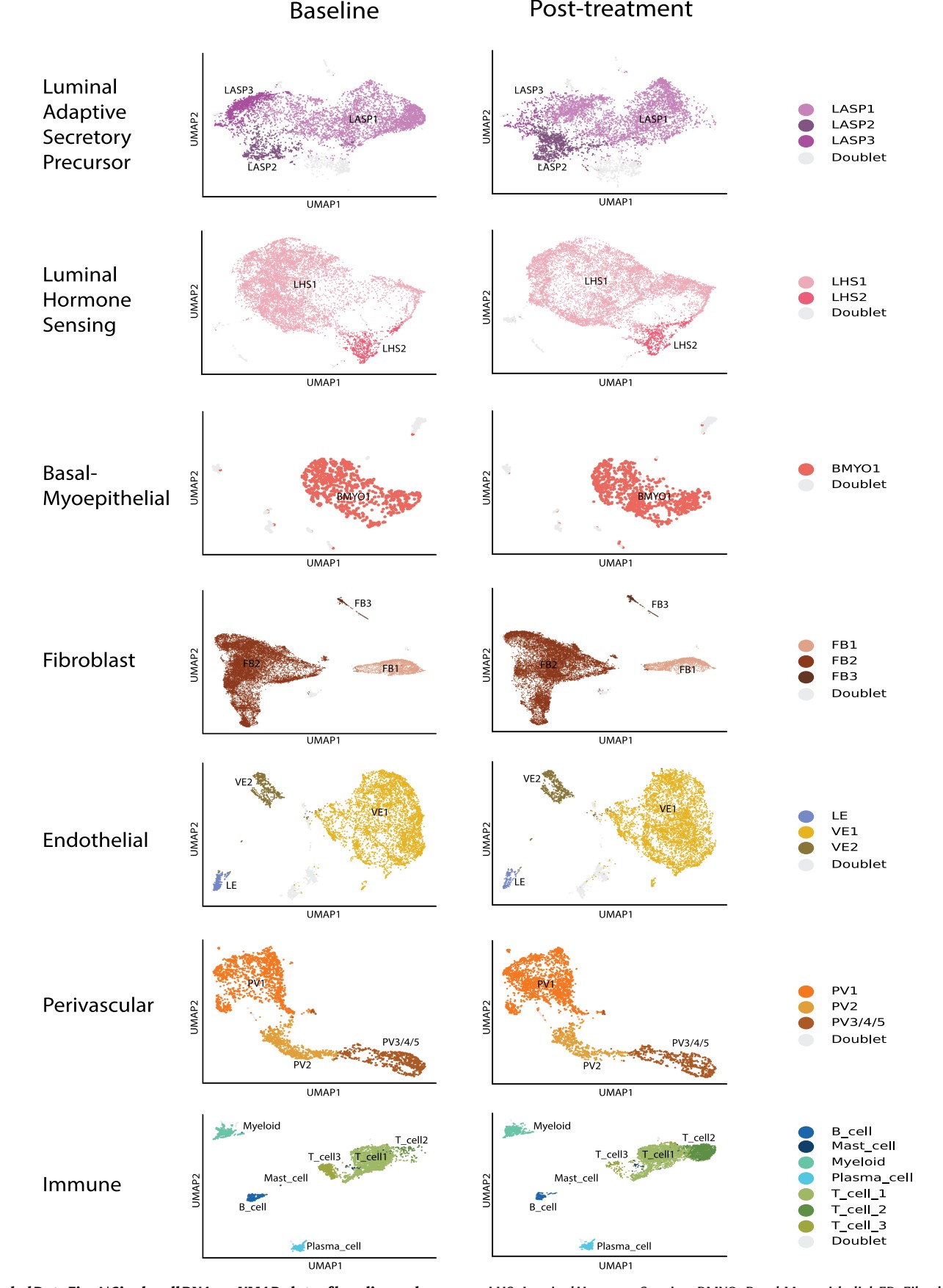

**Extended Data Fig. 4 | Single cell RNAseq UMAP plots of baseline and post-treatment samples showing the subcluster populations identified within the broad cell types.** LASP - Luminal Adaptive Secretory Precursor; LHS - Luminal Hormone Sensing; BMYO - Basal-Myoepithelial; FB - Fibroblast; LE - Lymphatic Endothelial; VE - Vascular Endothelial; PV - Perivascular.

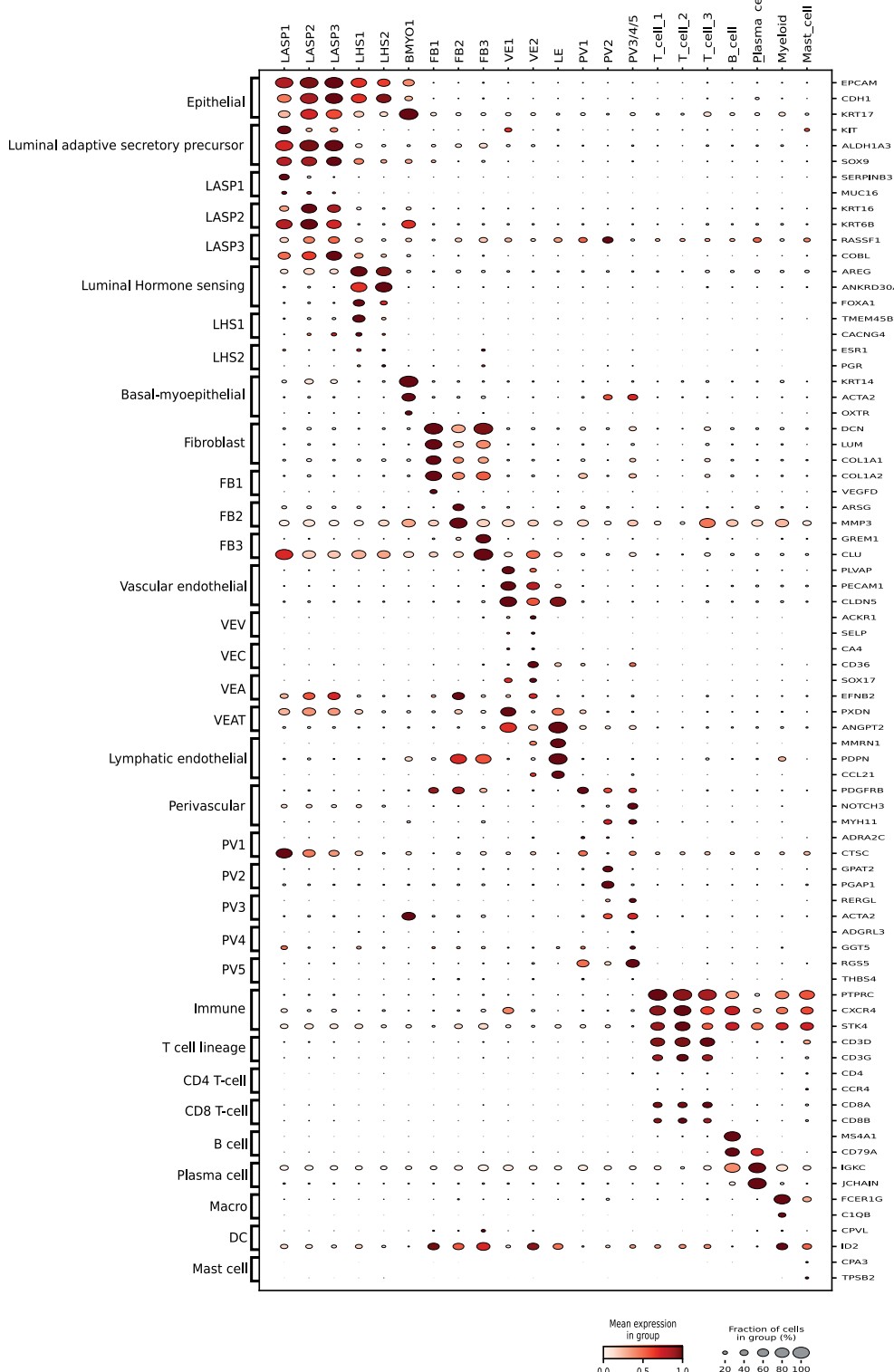

**Extended Data Fig. 5 | Dot plot summarises the selection of established cluster and subcluster cell type marker genes.** Each row corresponds to a key marker gene (expression normalized per gene) while brackets on the left side of the plot detail the cell type or subcluster that these genes mark. Each column corresponds to a specific subcluster cell population identified in our dataset.

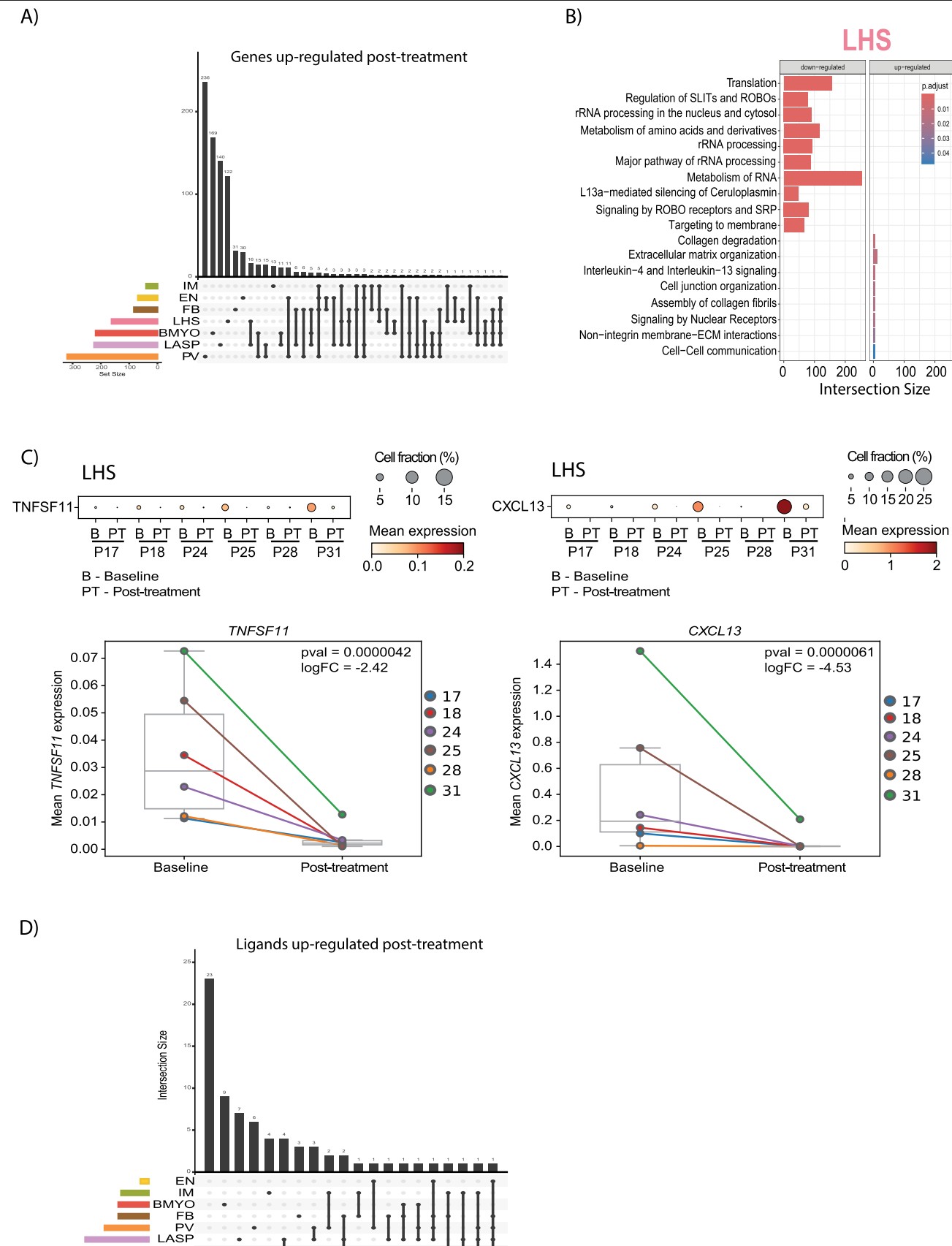

**Extended Data Fig. 6** | See next page for caption.

**Extended Data Fig. 6 | Progesterone induced genes are down-regulated post-UA treatment.** A) UpSet plot depicting up-regulated genes (>0.25 log FC, pval <0.05 - Memento analysis) post-UA treatment across all 7 broad cell types. Intersection size indicates the number of genes that are uniquely regulated within a single cell type or shared across two or more cell types. B) Pathway analysis of LHS cells comparing post-treatment to baseline. Intersection size indicates the number of genes within each pathway gene set. Fisher's one-tailed test p-values were corrected using Benjamini-Hochberg FDR, with adjusted p-value < 0.05 threshold for presented pathways. C) Dot plot (upper) and summary boxplots (lower) for *TNFSF11* and *CXCL13* gene expression across participant samples at baseline (B) and post-treatment (PT). Cell fraction represents the percentage of cells within LHS cells expressing each gene with mean expression overlaid (upper) or plotted on the y-axis (lower). Box plots centre lines represent median values, box bounds indicate the 25th and 75th percentiles, and whiskers extend to the most extreme datapoint within $1.5 \times IQR$ (inter-quartile range) of the outer hinge of the boxplot. P-value and log fold change (post-treatment/baseline) determined using Memento are overlaid. Memento significance is calculated using a nonparametric permutation test where observed mean differences are compared to a null distribution generated by permuting treatment group labels. N = 6 tissue pairs. D) UpSet plot depicting up-regulated genes that encode proteins that act as ligands (>0.25 log FC, pval <0.05 - Memento analysis) post-UA treatment across all 7 broad cell types.

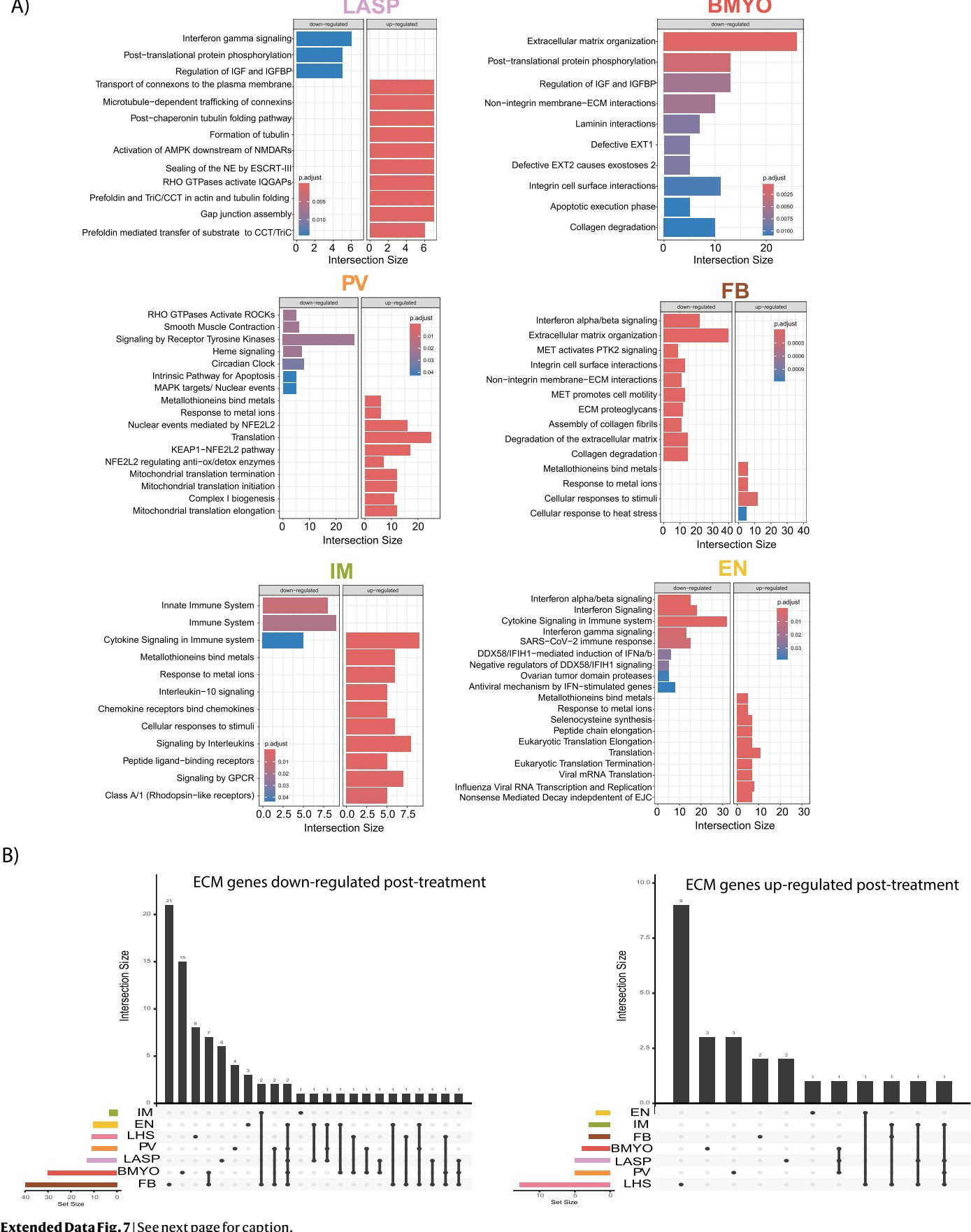

**Extended Data Fig. 7** | See next page for caption.

**Extended Data Fig. 7 | Pathway and differential expression analyses points towards ECM organisation as a main effect of UA exposure.** A) Pathway analysis of broad cell types comparing post-treatment to baseline. Intersection size indicates the number of genes within each pathway gene set. Fisher's one-tailed test p-values were corrected using Benjamini-Hochberg FDR, with adjusted p-value < 0.05 threshold for presented pathways. B) UpSet plots depicting down-regulated (left, < −0.25 log FC, pval <0.05 - Memento analysis) and up-regulated (right, > 0.25 log FC, pval <0.05 - Memento analysis) extracellular matrix (ECM) organisation genes post-UA treatment across all 7 broad cell types. Intersection size indicates the number of genes that are uniquely regulated within a single cell type or shared across two or more cell types.

A)

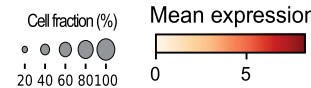

Extended Data Fig. 8 | See next page for caption.

**Extended Data Fig. 8 | Up-regulation of ECM regulatory genes, including MMPs, in LHS cells, and landscape of collagen gene expression across iHBCA annotated cell states.** A) Dot plot of up-regulated ECM genes in LHS cells showing gene expression at baseline and post-treatment. Cell fraction represents the percentage of cells within the LHS population expressing each gene with mean expression overlaid. B) Boxplots of up-regulated MMPs in LHS cells showing gene expression across participant samples at baseline and post-treatment. Box plots centre lines represent median values, box bounds indicate the 25th and 75th percentiles, and whiskers extend to the most extreme datapoint within 1.5 × IQR (inter-quartile range) of the outer hinge of the boxplot. P-value and log fold change (post-treatment/baseline) determined using Memento are overlaid. Memento significance is calculated using a nonparametric permutation test where observed mean differences are compared to a null distribution generated by permuting treatment group labels. N = 6 tissue pairs. C) Dot plot of collagen genes across all iHBCA annotated cell states. Cell fraction represents the percentage of cells within each cell state expressing each gene with mean expression annotated by colour.

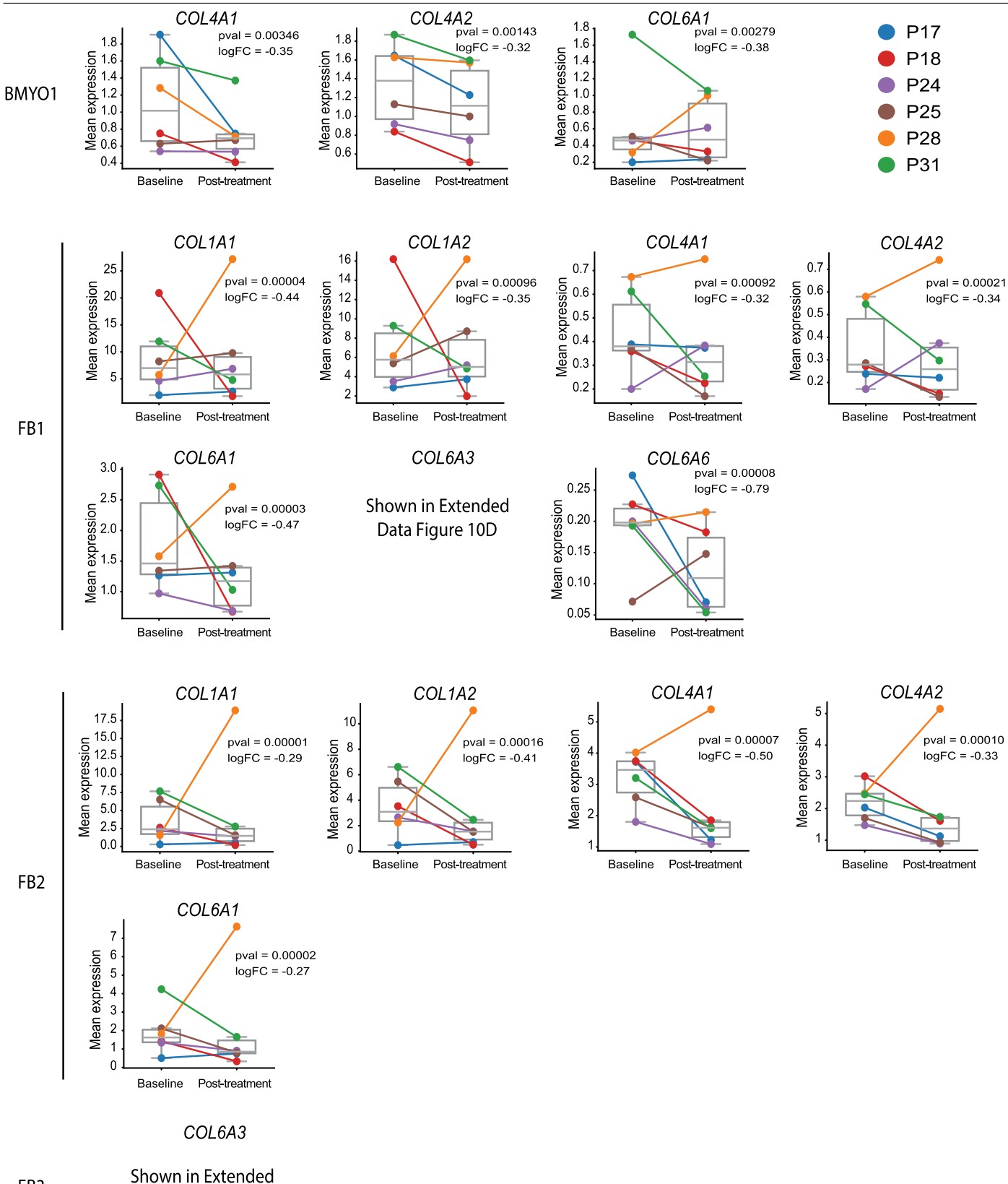

**Extended Data Fig. 9 | Down-regulation of collagen gene expression in fibroblast and basal-myoepithelial subclusters following UA treatment.** Boxplots of down-regulated collagens in BMYO1, FB1, FB2 and FB3 subclusters showing gene expression across participant samples at baseline and post-treatment. Box plots centre lines represent median values, box bounds indicate the 25th and 75th percentiles, and whiskers extend to the most extreme datapoint within 1.5 × IQR (inter-quartile range) of the outer hinge of the boxplot. P-value and log fold change (post-treatment/baseline) determined using Memento are overlaid. Memento significance is calculated using a nonparametric permutation test where observed mean differences are compared to a null distribution generated by permuting treatment group labels. N = 6 tissue pairs.

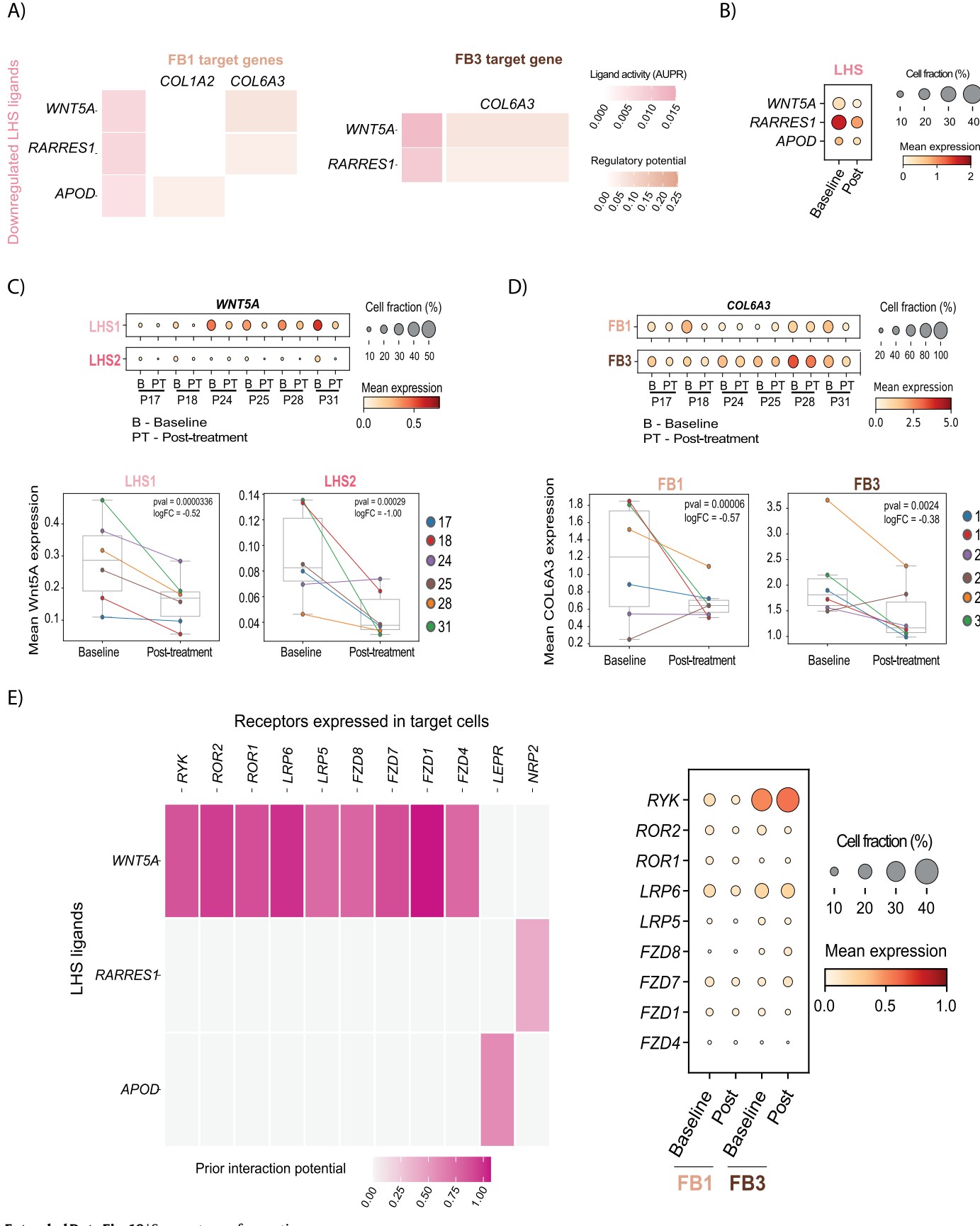

**Extended Data Fig. 10** | See next page for caption.

**Extended Data Fig. 10 | Ligand–receptor–target gene analysis links *WNT5A* to the regulation of *COL6A3*.** A) NicheNet ligand–target analysis of pairwise down-regulated LHS ligands post-treatment that are linked to collagen gene down-regulation in FB1 (left) or FB3 (right) cells. Ligand activity represents a combined score of magnitude of down-regulation in this dataset and established regulatory potential of each ligand:target link. B) Dot plot of LHS ligands at baseline and post-treatment that are linked to collagen gene regulation. Cell fraction represents the percentage of cells within LHS cells expressing each gene with mean expression overlaid. C-D) Dot plot (upper) and summary boxplots (lower) for *WNT5A* (C) and *COL6A3* (D) gene expression across participant samples at baseline (B) and post-treatment (PT) in LHS1-LHS2 cells (C) or FB1-FB3 cells (D). Cell fraction represents the percentage of cells within each subcluster expressing each gene with mean expression overlaid (upper) or plotted on the y-axis (lower). Box plots centre lines represent median values, box bounds indicate the 25th and 75th percentiles, and whiskers extend to the most extreme datapoint within 1.5 × IQR (inter-quartile range) of the outer hinge of the boxplot. P-value and log fold change (post-treatment/baseline) determined using Memento are overlaid. Memento significance is calculated using a nonparametric permutation test where observed mean differences are compared to a null distribution generated by permuting treatment group labels. N = 6 tissue pairs. E) Receptors for down-regulated LHS ligands expressed within target FB1 and FB3 cells. Prior interaction potential represents pre-existing links within the NicheNet L–R database (left). Dot plot of *WNT5A* linked receptors at baseline and post-treatment in FB1 and FB3 cells. Cell fraction represents the percentage of cells within FB1 or FB3 cells expressing each gene with mean expression overlaid.

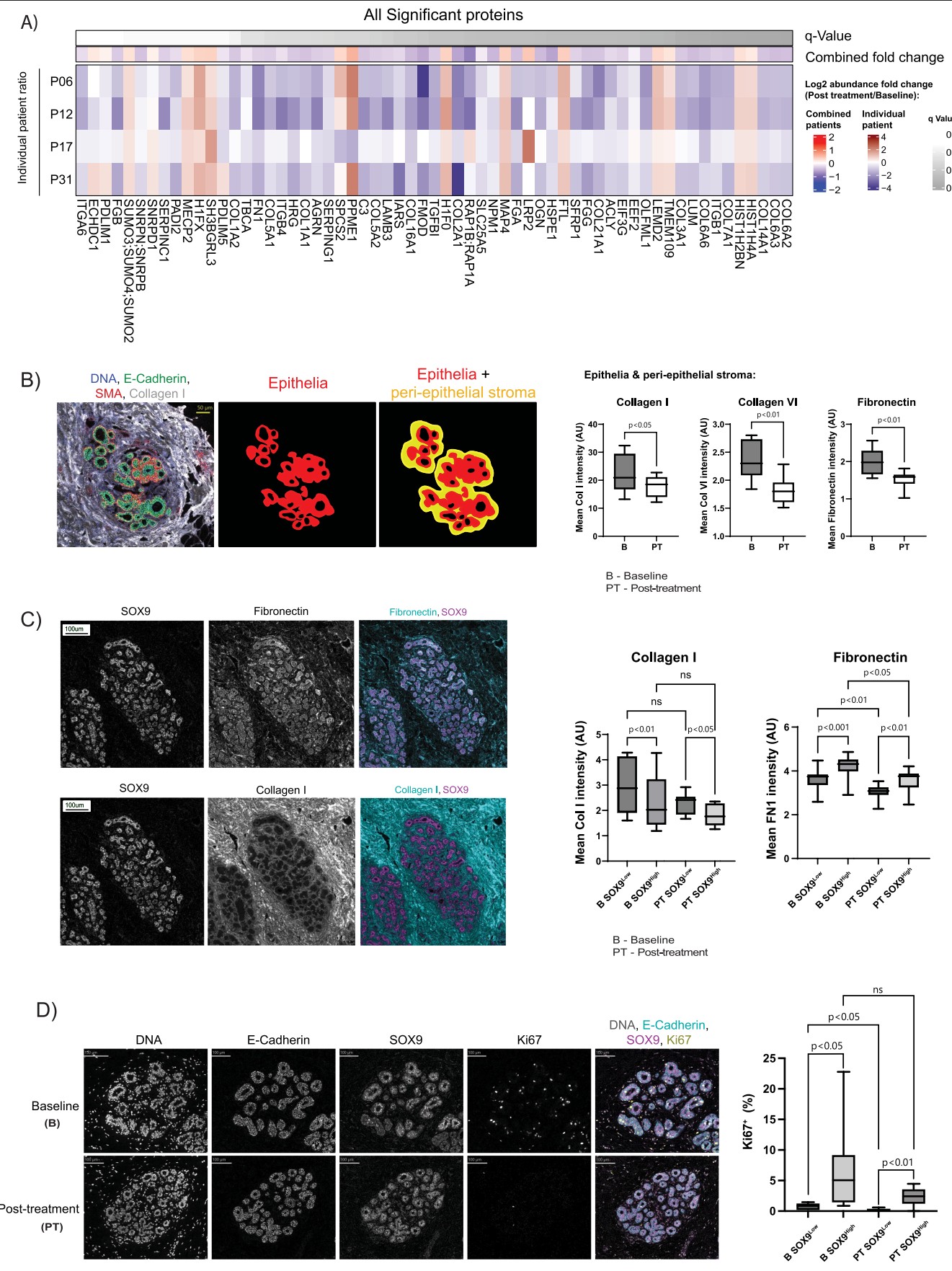

**Extended Data Fig. 11** | See next page for caption.

**Extended Data Fig. 11 | Anti-progestin treatment induces ECM remodelling.**
A) Heatmap of the 65 proteins identified as significantly differentially abundant after UA treatment. B) Representative imaging mass cytometry images showing staining with metal-conjugated antibodies against E-Cadherin (E-Cad; luminal cell marker), α-smooth muscle actin (SMA; basal cell marker) and Collagen I (Col-I; stromal marker). E-Cad and SMA markers were used to delineate the epithelial regions and define the peri-epithelial stroma (+25 μm from epithelia). Box plots compare Col-I, Col-VI and FN1 mean intensity in epithelia and peri-epithelial stroma of paired tissue sections before (B) and after (PT) UA treatment. Box plots centre lines represent median values, box bounds indicate the 25th and 75th percentiles, and whiskers denote minima and maxima values. Statistical analysis was performed using two-sided Wilcoxon matched pairs signed rank test. N = 8 tissue pairs. C) Single-cell neighbourhood analysis of pericellular collagen I (left hand panel) and fibronectin (right hand panel) abundance for SOX9[high] and SOX9[low] populations across paired BC-APPS1 samples at baseline (B) and post treatment (PT) timepoints. Single-cell neighbourhood analysis was performed as described in Fig. 3g. For each selected cell, Collagen-I or fibronectin staining intensity was quantified within a 10 μm radius. Box plots centre lines represent median values, box bounds indicate the 25th and 75th percentiles, and whiskers denote minima and maxima values. Statistical analysis was performed using a repeated measure one-way ANOVA followed by Sidak's multiple comparisons test. N = 8 tissue pairs. D) The percentage of Ki67+ cells in epithelial SOX9[high] and SOX9[low] cells populations were calculated across the samples described in Extended Data Fig. 11c. Box plots centre lines represent median values, box bounds indicate the 25th and 75th percentiles, and whiskers denote minima and maxima values. Statistical analysis was performed using repeated measure one-way ANOVA followed by Sidak's multiple comparisons test. N = 8 tissue pairs.

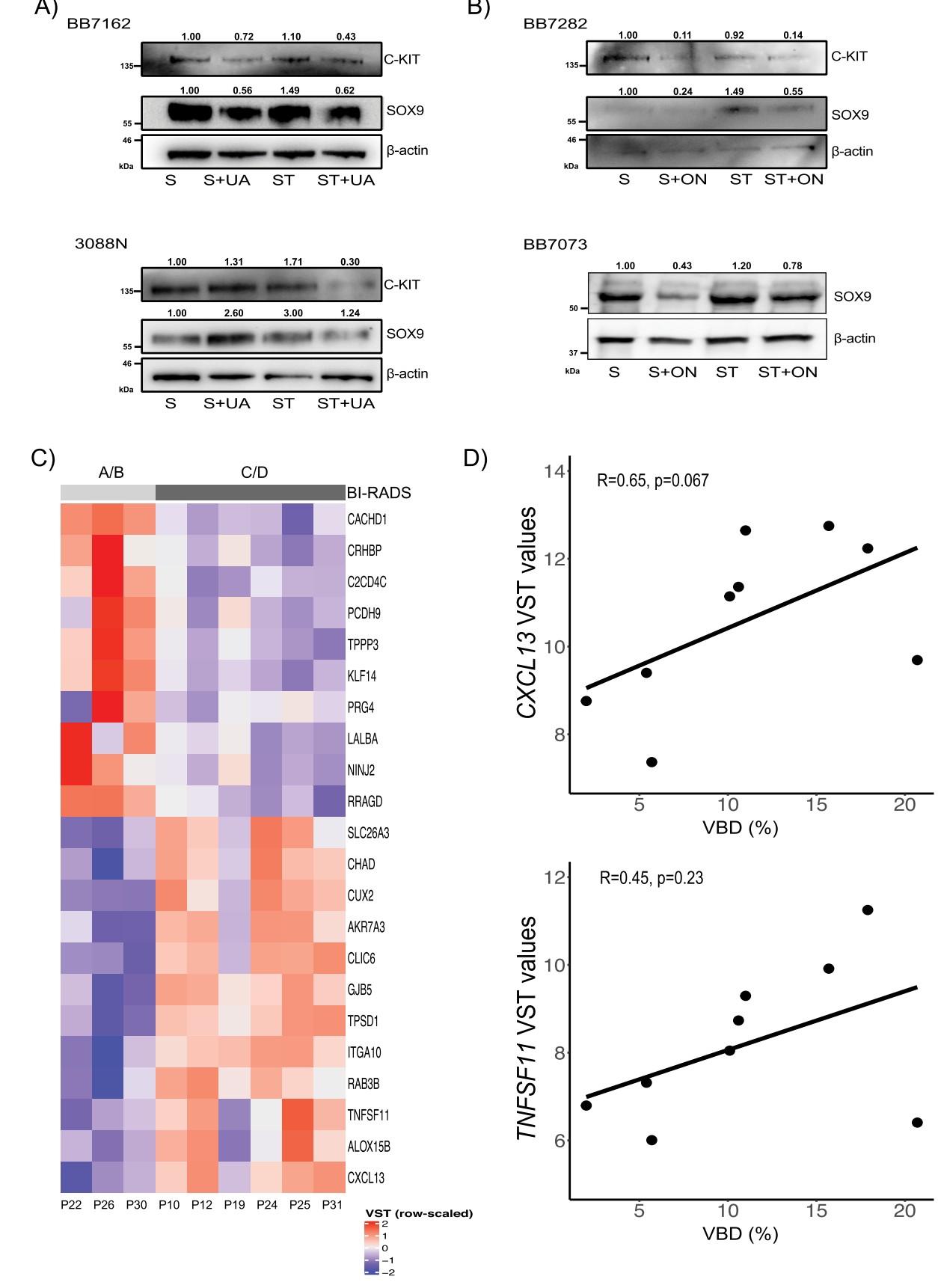

**Extended Data Fig. 12** | See next page for caption.

**Extended Data Fig. 12 | Anti-progestins block stiffness-induced SOX9 and C-KIT.** A-B) C-KIT and SOX9 protein detection in normal breast microstructures cultured in collagen-mimetic hydrogels under 'soft' (S) and 'stiff' (ST) conditions, treated for 7 days with: A) Ulipristal Acetate (UA, 2 nM), or B) Onapristone (ON, 100 nM). β-actin was used as a reference for the loading control. Densitometry quantification, normalised to β-actin, is shown at the top of each band. N = 3 breast samples. C) Heatmap from whole tissue RNA-Seq gene expression analysis showing all the differentially expressed genes between high MD (BI-RADS C/D, dark grey) and low MD (BI-RADS A/B, light grey) breast tissue at BC-APPS1 baseline (n = 9; p < 0.05). VST - Variance Stabilising Transformation. D) Correlation (two-sided test) between VBD percentage and gene expression of *CXCL13* and *TNFSF11* in baseline breast tissue of BC-APPS1 participants. N = 9 baseline tissue.

# Reporting Summary

## Statistics

For all statistical analyses, confirm that the following items are present in the figure legend, table legend, main text, or Methods section.

| n/a | Confirmed | |
|-----|-----------|---|
| ☐ | ☒ | The exact sample size (*n*) for each experimental group/condition, given as a discrete number and unit of measurement |
| ☐ | ☒ | A statement on whether measurements were taken from distinct samples or whether the same sample was measured repeatedly |
| ☐ | ☒ | The statistical test(s) used AND whether they are one- or two-sided<br>*Only common tests should be described solely by name; describe more complex techniques in the Methods section.* |
| ☐ | ☒ | A description of all covariates tested |
| ☐ | ☒ | A description of any assumptions or corrections, such as tests of normality and adjustment for multiple comparisons |
| ☐ | ☒ | A full description of the statistical parameters including central tendency (e.g. means) or other basic estimates (e.g. regression coefficient) AND variation (e.g. standard deviation) or associated estimates of uncertainty (e.g. confidence intervals) |
| ☐ | ☒ | For null hypothesis testing, the test statistic (e.g. $F$, $t$, $r$) with confidence intervals, effect sizes, degrees of freedom and $P$ value noted<br>*Give P values as exact values whenever suitable.* |
| ☒ | ☐ | For Bayesian analysis, information on the choice of priors and Markov chain Monte Carlo settings |
| ☒ | ☐ | For hierarchical and complex designs, identification of the appropriate level for tests and full reporting of outcomes |
| ☒ | ☐ | Estimates of effect sizes (e.g. Cohen's *d*, Pearson's *r*), indicating how they were calculated |

*Our web collection on statistics for biologists contains articles on many of the points above.*

## Software and code

Policy information about availability of computer code

| | |
|---|---|
| Data collection | Tissue bulk RNAseq: NextSeq 500 sequencer (Illumina), Nextflow (v19.10.0), nf-core/rnaseq (v1.3) using GRCh38 human genome as reference<br>scRNAseq: NovaSeq 6000 (Illumina), CellRanger (3.02) using GRCh38 human genome as reference<br><br>Mass Spectrometry: UltiMate 3000 Rapid Separation LC system (RSLC, Dionex Corporation) coupled to a Q Exactive HF™ mass spectrometer (Thermo Fisher), MaxQuant (v1.6.14.0)<br><br>Imaging Mass Cytometry: Hyperion Imaging System (Standard BioTools Inc., Fluidigm CyTOF Software v7.0) |
| Data analysis | All custom code and scripts used to generate analyses of the tissue bulk RNAseq, the single-cell RNA-seq, and the IMC data are available at the following link: https://zenodo.org/records/11369094<br><br>Other software packages include:<br><br>Tissue bulk RNAseq: Nextflow (v19.10.0); nf-core/rnaseq (v1.3) using GRCh38 human genome as reference; Ensembl BioMart (v101); DESeq2 (v1.26.0); pheatmap (v1.0.12); stats (v3.6.0), SummarisedExperiment (v1.16.1)<br><br>scRNAseq: DropletUtils (v1.10.3), scuttle (v1.18.6), org.Hs.eg.db (v3.12.0), batchelor (v1.6.3), SNN graph (scran v1.18.7), igraph (v1.2.6), scater (v1.18.6), Cell Chat (v2.1.2), NicheNet (v2.2.0), Memento (v0.1.0), UpsetR (v1.4.0), gprofiler2 (v0.2.3), python (v3.10.13), scanpy (v1.9.6)<br><br>Mass Spectrometry: MSqRob, clusterProfiler (v4.6.0), Reactome pathway (v.65) |

Flow cytometry: FACSDIVA (v8.0)

Histology: Aperio ImageScope Digital Pathology Slide viewer (v12.4.6), HALO Image Analysis Software (Indica Labs, v3.6.4134.314)

MRI scans: software ITK-SNP (v3.8.0)

Mammographic Density: Volpara density (v1.5.0)

Imaging Mass Cytometry: Steinbock (v0.15.0), MATLAB (v2022b)

Real-time PCR: QuantStudio Design and Analysis software (v2.6.0)

Western blot: Image Lab (v6.1, BioRad)

For manuscripts utilizing custom algorithms or software that are central to the research but not yet described in published literature, software must be made available to editors and reviewers. We strongly encourage code deposition in a community repository (e.g. GitHub). See the Nature Portfolio guidelines for submitting code & software for further information.

# Data

Policy information about availability of data

All manuscripts must include a data availability statement. This statement should provide the following information, where applicable:
- Accession codes, unique identifiers, or web links for publicly available datasets
- A description of any restrictions on data availability
- For clinical datasets or third party data, please ensure that the statement adheres to our policy

Datasets generated in the BC-APPS1 study:

All bulk and single cell RNA-sequencing data has been deposited in the Array Express data base (https://www.ebi.ac.uk/biostudies/arrayexpress) and can be retrieved by the following access IDs: E-MTAB-13720 (bulk RNAseq) and E-MTAB-13819 (scRNAseq).

The mass spectrometry proteomics data have been deposited to the ProteomeXchange Consortium via the PRIDE partner repository (https://www.ebi.ac.uk/pride/) with the dataset identifier PXD067122.

Datasets sourced from previously published studies:

RNAseq sequencing reads were aligned using the GRCh38 human genome as reference, available at https://www.ncbi.nlm.nih.gov/datasets/genome/GCF_000001405.39/

scRNAseq was compared to the scRNAseq dataset published by Reed et al, 2024 (doi: : 10.1038/s41588-024-01688-9; processed scRNAseq data can be downloaded at the CellXGene site - https://cellxgene.cziscience.com/collections/cd9a09e2- b440-4887-9163-6f8c684c7ced).

To assign peptides to protein groups, peptides were searched against the UniProt human proteome reference database (UP000005640) available at https://www.uniprot.org/proteomes/UP000005640

Proteins were classified as belonging to the "Matrisome" (structural ECM or ECM modifying) by searching against MatrisomeDB, a curated database of ECM proteins, available at https://sites.google.com/uic.edu/matrisome/matrisome-annotations/homo-sapiens

GSEA was performed using the Reactome Pathways database (v65), available at https://reactome.org/

# Research involving human participants, their data, or biological material

Policy information about studies with human participants or human data. See also policy information about sex, gender (identity/presentation), and sexual orientation and race, ethnicity and racism.

| Reporting on sex and gender | Only women (female sex) were eligible for this study. |
| Reporting on race, ethnicity, or other socially relevant groupings | Case selection was not based on any racial, ethnic or other demographic factors. Such data were not collected prospectively and are not reported in the manuscript. |
| Population characteristics | Patient demographics are presented in Supplementary Table 1. |
| Recruitment | All participants were recruited from the Family History Risk and Prevention Clinic at the Nightingale Centre, Wythenshawe Hospital, Manchester, UK. All participants were selected on their age (25–45 years), premenopausal status (including occurrence of regular menstrual cycles), absence of prior bilateral risk-reducing mastectomies, and having at least a moderately increased risk of breast cancer by virtue of a family history of the disease. Complete eligibility criteria are provided in the protocol (Supplementary Appendix 1). All eligible individuals were sent a letter of invitation and asked to contact the study team if interested in participating. Those who responded positively were provided with the participant information sheet and subsequently invited to clinic for consent, eligibility confirmation and recruitment to the study. As this was an investigational study for a drug with no proven benefit, requiring bilateral breast biopsies and two contrast-enhanced MRI scans, the participants were altruistic young women highly motivated to contribute to medical research. While this may |

reflect a degree of self-selection bias, there is no evidence to suggest this would influence objective molecular or radiological responses to the anti-progestin. Ethnicity and socioeconomic status of the participants were not recorded, and we therefore cannot comment on selection bias from those perspectives. Results may not, therefore, be generalisable to a broader population.

| Ethics oversight | The Breast Cancer - Anti-Progestin Prevention Study 1 (BC-APPS1) was a single arm single centre phase II study registered under the name "A pilot prevention study of the effects of the anti- progestin Ulipristal Acetate (UA) on surrogate markers of breast cancer risk" (EudraCT registration number: 2015-001587-19; registration date: 15/07/2015; Greater Manchester – South, Research Ethics Committee number 15/NW/0478). |
|---|---|

Note that full information on the approval of the study protocol must also be provided in the manuscript.

# Field-specific reporting

Please select the one below that is the best fit for your research. If you are not sure, read the appropriate sections before making your selection.

☒ Life sciences ☐ Behavioural & social sciences ☐ Ecological, evolutionary & environmental sciences

For a reference copy of the document with all sections, see nature.com/documents/nr-reporting-summary-flat.pdf

# Life sciences study design

All studies must disclose on these points even when the disclosure is negative.

| Sample size | The planned sample size was n=30. This was based on a prior study using an alternate anti-progestin (mifepristone) which demonstrated significant reduction in Ki67 (the primary endpoint in our study) in 8 patients with no difference in 6 placebo treated women (Engman, M., et al, DOI: 10.1093/humrep/den228). A 30-subject study was proposed to provide sufficient data points to explore variability in response across both primary and secondary endpoints. Ultimately, 26 women were recruited due to a change in licensing for the drug by the MHRA. The objective for all experiments described below was to detect biological signals of ulipristal acetate activity in the breast. No formal sample size calculations were performed but all available samples were used, except for LCM proteomics, AFM, and IMC experiments where costs limited analyses. In those cases, a minimum of 4 paired samples were analysed. <br><br> Sample Prioritisation for Live Cell Analyses: As cell yield varied between participants due to differences in tissue composition, single-cell suspensions were prioritised as follows: (1) mammosphere formation efficiency assays (n = 19 pairs), (2) 2D colony-forming assays (n = 18 pairs), (3) flow cytometry (FACS) with cryopreserved cells (n = 17 pairs), and (4) scRNAseq, performed only on samples with ≥300,000 cryopreserved cells at both baseline and post-treatment (n = 6 pairs). This prioritisation ensured consistent use of available material across assays. <br><br> Sample Prioritisation for FFPE Analyses: Analyses were conducted as follows: (1) Ki67 immunostaining (n = 24 pairs), (2) tissue morphometry on samples with ≥3 well-defined lobules each containing ≥10 acini (n = 19 pairs), (3) PSR staining for collagen quantification on samples with ≥3 well-defined lobules (n = 22 pairs), (4) LCM proteomics, limited to 4 pairs with high epithelial content due to cost and processing constraints, and (5) IMC (Hyperion) imaging, using the same 4 pairs plus an additional 4 high-epithelium-content pairs (n = 8 total) as a confirmatory cohort. <br><br> Sample Prioritisation for Snap-Frozen Tissue Analyses: Snap-frozen cores were used for: (1) total RNA extraction for bulk RNA sequencing (n = 10 pairs, selected based on RNA quality), and (2) atomic force microscopy (AFM) stiffness measurements in samples showing ≥10% reduction in PSR staining post-treatment (n = 4 pairs). |
|---|---|
| Data exclusions | Analyses of paired breast tissue samples were applied to selected samples depending on epithelial cell availability, the technology utilized and its feasibility requirements. Single cell RNAseq data was excluded based on filtering and QC criteria as outlined in the methods section. |
| Replication | No technical replicates were performed on the same tissue samples; however, paired biopsy samples collected before and after treatment were analysed, representing biological replicates. All experiments included a minimum of four paired samples (i.e., four biological replicates). The replication attempts using these biological replicates were successful and the individual results across experiments are presented in the manuscript. For reference, bulk RNA sequencing was performed on only 10 paired samples, as detailed above. In the case of mammosphere formation efficiency assays, two baseline samples contained sufficient cells for plating but did not form any mammospheres. |
| Randomization | Previous studies have shown ulipristal acetate to suppress menstruation and endogenous progesterone levels in the majority of women (Donnez J. et al, DOI: 10.1056/NEJMoa1103182). A placebo arm was therefore not appropriate, as it would have been abundantly clear which women were taking ulipristal acetate. Randomisation between the established preventive agent tamoxifen and ulipristal acetate would have been another option; however, as both drugs reduce proliferation in the normal breast, the sample size required would have been very large and beyond the scope of a single institution study. A single-arm phase 2 study with paired biopsies was indicated in order to determine the effects of ulipristal acetate in women at increased breast cancer risk, with the potential for a follow-on randomised multicentre trial thereafter. |
| Blinding | Blinding was not applicable in this study, as all participants received the same treatment (ulipristal acetate). There was therefore no comparator group and, consequently, no need for participant or investigator blinding. |

# Reporting for specific materials, systems and methods

We require information from authors about some types of materials, experimental systems and methods used in many studies. Here, indicate whether each material, system or method listed is relevant to your study. If you are not sure if a list item applies to your research, read the appropriate section before selecting a response.

## Materials & experimental systems

| n/a | Involved in the study |
|-----|------------------------|
| ☐ | ☒ Antibodies |
| ☐ | ☒ Eukaryotic cell lines |
| ☒ | ☐ Palaeontology and archaeology |
| ☒ | ☐ Animals and other organisms |
| ☐ | ☒ Clinical data |
| ☒ | ☐ Dual use research of concern |
| ☒ | ☐ Plants |

## Methods

| n/a | Involved in the study |
|-----|------------------------|
| ☒ | ☐ ChIP-seq |
| ☐ | ☒ Flow cytometry |
| ☒ | ☐ MRI-based neuroimaging |

## Antibodies

**Antibodies used**

Immunohistochemistry: Confirm Anti-Ki-67 (30-9) antibody (Roche,Ventana 790-4286);

Immunofluorescence: anti-SOX9 (Millipore, AB5535, 1:2000) and Ki67 (Dako, M 7240, 1:100);

Flow cytometry: anti-CD31 biotin-conjugated (eBioscience, 13-0319-82, 0.25 µg/ml), anti-CD45 biotin-conjugated (BioLegend, 304004, 0.125 µg/ml), APC-Cy7 streptavidin-conjugated (BioLegend, 405208, 0.2 µg/ml), anti-CD49f-APC (BioLegend, 313616, 0.5 µg/ml) and anti-EpCAM-FITC (StemCell Technologies, 10109, 1:5);

Imaging Mass Cytometry: anti-Alpha-Smooth muscle actin Pr-metal conjugated (#201508, Standard BioTools Inc., 1:1000), anti-SOX9 Sm-metal conjugated (#3147022D, Standard BioTools Inc., 1:75), anti-Fibronectin Sm-metal conjugated (#ab23750, Abcam, 1:75), anti-E-Cadherin Gd-metal conjugated (#201508, Standard BioTools Inc., 1:600), anti-Collagen VI Gd-metal conjugated (#ab6588, Abcam, 1:75), anti-Ki67 Er-metal conjugated (#201508, Standard BioTools Inc., 1:100) and anti-Collagen I Tm-metal conjugated (#201508, Standard BioTools Inc., 1:400);

Western blot: anti-SOX9 rabbit polyAb (#AB5535, Sigma, 1:1000), anti-C-KIT mouse polyAb (#MAB332, R&D Biosystems, 1:1000), anti-β-actin mouse mAb (#A1978, Sigma, 1:5000), goat anti-rabbit (#41424306, Dako, 1:5000) and goat anti-mouse (#41424131, Dako, 1:5000).

**Validation**

The commercial antibody validation can be found on each company's product sheet associated with the catalogue numbers. Additional validation details are summarised below.

Immunohistochemistry:
• Anti-Ki-67 (Roche/Ventana) – Manufacturer validation includes routine staining tests for sensitivity, specificity and precision on control tissues (lymph node, tonsil). Precision studies demonstrated between-lot reproducibility.

Immunofluorescence:
• Anti-SOX9 (Millipore) – Validated by Western blot in HepG2 lysates; control staining demonstrated in embryonic tissue and adult chondrocytes. Affinity-purified, reactive with human, mouse, rat, and chicken; widely cited (>1300 publications).
• Anti-Ki67 (Dako) – Extensively validated and cited (>4000 publications). Specific for Ki-67 with high lot-to-lot consistency, demonstrated specificity by Western blot and competitive binding assays. Control staining demonstrated in tonsillar sections.

Flow Cytometry:
• Anti-CD31 biotin-conjugated (eBioscience) – Validated for flow cytometry; tested on normal human peripheral blood cells.
• Anti-CD45 biotin-conjugated (BioLegend) – Each lot QC-tested by flow cytometry.
• Streptavidin-APC-Cy7 conjugate (BioLegend) – Each lot QC-tested by flow cytometry.
• Anti-CD49f-APC (BioLegend) – Each lot QC-tested by flow cytometry.
• Anti-EpCAM-FITC (StemCell Technologies) – Verified for flow cytometry applications.

Imaging Mass Cytometry:
• Anti-α-Smooth Muscle Actin (Standard BioTools Inc.) – Pathologist-verified for IMC on FFPE and frozen human tissue; QC-tested by IMC per lot; reactive with human and mouse.
• Anti-SOX9 (Standard BioTools Inc.) – Pathologist-verified for IMC on human FFPE tissue; QC-tested per lot; reactive with human, mouse, and rat.
• Anti-Fibronectin (Abcam) – Validated for IHC; >200 citations; strong specificity for human fibronectin.
• Anti-E-Cadherin (Standard BioTools Inc.) – Pathologist-verified for IMC on FFPE and frozen human tissue; QC-tested per lot; reactive with human, mouse, bovine.
• Anti-Collagen VI (Abcam) – Validated for Western blot on human tissue; strong specificity for type VI collagen, minimal cross-reactivity with other collagens; >150 citations.
• Anti-Ki-67 (Standard BioTools Inc.) – Pathologist-verified for IMC on FFPE and frozen human tissue; QC-tested per lot; reactive with human, mouse, rat, porcine.
• Anti-Collagen I (Standard BioTools Inc.) – Pathologist-verified for IMC on FFPE and frozen human tissue; QC-tested per lot; reactive with human and mouse.

Western Blot:

• SOX9 (Sigma/Merck) – Widely cited (>1300 publications); species reactivity includes human, mouse, rat.
• c-KIT (R&D Systems) – Detects human c-KIT in ELISA and WB; no cross-reactivity with mouse reported.
• β-actin (Sigma) – Validated for WB using extracts from human foreskin fibroblasts and chicken fibroblasts; each lot QC-tested by WB on these controls.

# Eukaryotic cell lines

Policy information about cell lines and Sex and Gender in Research

| | |
|---|---|
| Cell line source(s) | NIH 3T3 Swiss mouse embryo fibroblast cell line purchased from American Type Culture Collection (ATCC; CRL-1658) |
| Authentication | NIH 3T3 cell line was not independently authenticated as cells were utilised within 10 passages of acquisition from ATCC. |
| Mycoplasma contamination | NIH 3T3 cell line was not tested for mycoplasma contamination as cells were utilised within 10 passages of acquisition from ATCC. |
| Commonly misidentified lines (See ICLAC register) | N/A |

# Clinical data

Policy information about clinical studies

All manuscripts should comply with the ICMJE guidelines for publication of clinical research and a completed CONSORT checklist must be included with all submissions.

| | |
|---|---|
| Clinical trial registration | NCT02408770 |
| Study protocol | Study Protocol is provided in Supplementary Appendix 1 |
| Data collection | Recruitment ran from 29/03/2016 to 11/03/2019 at The Nightingale Centre, Wythenshawe Hospital, Manchester UK. Data collection from these participants continued until the last participant had completed study procedures (July 2019). Stored biological samples continued to be analysed thereafter. The study was formally closed on 31/12/2023. |
| Outcomes | The primary endpoint was the change in epithelial cell proliferation measured by %Ki67 staining before and after treatment - see "Ki67 staining" section of Methods. Secondary endpoints were (1) percentage of luminal, basal and mixed colonies by morphological analysis of adherent feeder layer assay - see "2D Human mammary colony forming assay" section of Methods; (2) percentage of luminal progenitor cells (EPCAM+/CD49f+) by FACS analysis - see "Flow cytometry analysis" section of Methods; (3) tissue stiffness assessed as the reduced indentation modulus by atomic force microscopy - see "Tissue stiffness by atomic force microscopy" section of Methods; (4) mean tissue section percentage fibrillar collagen assessed by picrosirius red staining and polarised light microscopy - see "Picrosirius red staining and polarised light microscopy" section of Methods; (5) background parenchymal enhancement assessed by magnetic resonance imaging (MRI) - see "Magnetic resonance imaging (MRI)" section of Methods; (6) the side effect profile of UA in this patient population assessed by CTCAE v4.03. |

# Plants

| | |
|---|---|
| Seed stocks | N/A |
| Novel plant genotypes | N/A |
| Authentication | N/A |

# Flow Cytometry

## Plots

Confirm that:

☒ The axis labels state the marker and fluorochrome used (e.g. CD4-FITC).

☒ The axis scales are clearly visible. Include numbers along axes only for bottom left plot of group (a 'group' is an analysis of identical markers).

☒ All plots are contour plots with outliers or pseudocolor plots.

☒ A numerical value for number of cells or percentage (with statistics) is provided.

## Methodology

| | |
|---|---|
| Sample preparation | Normal breast tissue was minced into ~2mm³ fragments and incubated in a dissociation medium containing phenol red-free DMEM/F12, 25% BSA Fraction V, 1mg/mL collagenase/hyaluronidase, and 5μg/mL insulin. After overnight digestion at 37°C with shaking, the cell suspension was washed and centrifuged at 450 x g for 5 minutes at 4°C. The epithelial pellet was treated with 0.05% Trypsin-EDTA and 5 mg/mL dispase, resuspended in HBSS/Hepes/FBS, and filtered through 100μm and 40μm sieves to obtain a single cell suspension. Cells were counted and frozen in Bambanker freezing media. Cells from paired samples were then stained for flow cytometry using the following antibodies: CD31 biotin-conjugated (eBioscience, 13-0319-82), CD45 biotin-conjugated (BioLegend, 304004), APC-Cy7 streptavidin-conjugated (BioLegend, 405208), CD49f-APC (BioLegend, 313616) and EpCAM-FITC (StemCell Technologies, 10109). |
| Instrument | BD™ LSR II flow cytometer |
| Software | BD FACSDiva™ (v8.0) |
| Cell population abundance | N/A |
| Gating strategy | Following singlet, live/dead and lineage CD45+/CD31+ exclusion, mammary gland epithelial lineages were determined as Luminal Progenitor (LP, CD49f+/EPCAM+), Luminal Mature (LM, CD49f-/EPCAM+) or Basal (B, CD49f+/EPCAM-/lo). |

☒ Tick this box to confirm that a figure exemplifying the gating strategy is provided in the Supplementary Information.

