## [Peer Review File · Nature]

Anti-progestin therapy targets hallmarks of breast cancer risk

Corresponding Author: Dr Bruno Simões

Version 0:

Reviewer comments:

Referee #1

(Remarks to the Author)

A. Summary of the key results

The authors report the results of the clinical trial BC-APPS1. This window trial aimed at assessing the effects of the progesterone receptor antagonist ulipristal acetate in breast cancer prevention. Twenty-four premenopausal women with inferred high risk because of their family history were treated for 12 weeks with daily UA and pre- and post-treatment tissue OMICs and imaging analyses were performed.

Decrease in progenitor cell activity was decreased as assessed by colony formation, mammosphere assays and through scRNA seq analysis. Breast density, considered surrogate marker of breast cancer risk, decreased and the expression of ECM encoding genes was decreased and this is attributed to changes in fibroblasts and basal cells. Experiments with hydrogels of different stiffness show that stiffness induced PR target gene expression is abrogated by UA treatment. Hyperion imaging mass cytometry showed that luminal progenitors identified as SOX9^{high} cells are in close proximity to regions with high expression of the non-fibrillar COLVI.

The authors conclude that the progesterone receptor antagonist UA, holds promise as a preventative treatment strategy in premenopausal patients.

B. Originality and significance: if not novel, please include reference

The role of PR signaling in breast carcinogenesis is a topic of ongoing debate. As such the present study with new clinical data will be a very valuable contribution to the breast cancer field. The design of the trial with consideration of the luteal phase and hormone measurements is original and improves over previous designs.

C. Data & methodology: validity of approach, quality of data, quality of presentation

There is little information for the genetic basis of the increased BC risk of the women included in this study. To what extent can it be attributed to germline mutations in BRCA1 or BRCA2? Those are likely to be involved and could limit the extrapolation of the findings to the population at large as it has been shown that BRCA1 germ line mutation carriers have an increased compartment of aberrant progenitor cells (Lim Nat Med 2009) which could confound analysis and interpretation of the results.

Approach testifies to close collaboration between disciplines and the authors are complemented on coordination of sample collection with the participants' luteal phase. For the post treatment point, is there still a luteal phase to speak of, after 12 weeks of UA treatment? The authors should explain what happens to the participants' endogenous hormone levels upon treatment.

D. Appropriate use of statistics and treatment of uncertainties

The Extended Data Table illustrates well the challenges of translational research studies with the difficulty of obtaining multiple datasets from individual patient samples. The authors may consider adding it to the main manuscript. Any genetic information and more details about the family history should be added to this table. The known role of BRCA1 in the progenitor cell population may limit interpretability of results from a subset of samples but may also add insights if considered as additional variable.

E. Conclusions: robustness, validity, reliability

The conclusion on the general utility for premenopausal prevention is overdrawn as the women included in the present study have high familial risk. Some of this (BRCA1mt) has been linked to aberrant progenitor populations and confounds the study.

F. Suggested improvements: experiments, data for possible revision

Patients with normal risk should be included.

Minor: The following statement is somewhat misleading: Lines 71ff "Supplementation of progestin, either as a contraceptive or hormone replacement Magnetic Resonance Imaging (MRI) measurements of fibro glandular volume (FGV) correlate well with automated volumetric MD, and FGV is greater in the luteal vs follicular phase of the menstrual cycle. The progestins are usually administered together with an estrogenic compound.

G. References: appropriate credit to previous work?

A lot of previous work has been overlooked such as very pertinent studies from Seema Khan, Claudia Lanari and Martin Widschwendter as well as work from Susan Clare, Cathrin Brisken, Pepper Schedin, Geoff Lindeman (Lim et al).

H. Clarity and context: lucidity of abstract/summary, appropriateness of abstract, introduction and conclusions

The manuscript is clear and the abstract, introduction and conclusions are appropriate. Previous window trials, albeit in the therapy setting should be discussed. The conclusions overdrawn .

Referee #2

(Remarks to the Author)

The manuscript titled, "BC-APPSI: Anti-progestin therapy targets hallmarks of breast cancer risk through epithelial-stromal remodeling", demonstrates anti-progestin mediated alterations in the breast microenvironment of women at high risk for breast cancer. The authors provide a multi-omic analysis of paired biopsies from pre- and post-treatment breast tissues. The results show a significant reduction in the proliferation and activity of luminal progenitor cells. This is associated with a robust change in ECM composition, architecture, and tissue stiffness. Together, this data shows promise for targeting prolactin signaling as a prevention for breast cancers in premenopausal women. The manuscript is well written, the data is original, strong, and clearly presented, and the statistical analysis and references are appropriate. However, there are a few points that should be addressed to improve the manuscript prior to publication.

Major concerns:

1. The authors suggest that changes in mammographic density may be used as a surrogate for the risk reduction of anti-progestin therapy. In addition, mammographic density is implicated as a potential biomarker for response to anti-progestin therapy. The cohort of patients included in this study have a range of initial mammographic density scores and a range of treatment responses. How do the data (changes in ECM (proteomics), AFM/stiffness, and luminal progenitors) relate to the mammographic density scores? The data points in the graphs could be presented in a color-coded fashion (similar to Fig 4A) to represent response by mammographic density score. Correlation analysis between density score and anti-progestin response would strengthen the argument for mammographic density as a biomarker for anti-progestin treatment.
2. While the authors provide evidence that increased mammographic density scores have higher progesterone signaling, there is no investigation into Progesterone Receptor (PR) protein expression or which cells are expressing PR. Does mammographic density alter the expression of PR on the epithelium or fibroblasts and how does PR expression relate to treatment response? PR expression along with mammographic density scores could provide a signature biomarker for treatment.
3. A major finding of the manuscript is that anti-progestin treatment remodels the ECM (stiffness and architecture), however, the presentation of these results could be improved. The change in architecture as measured by birefringence seems very subtle and is difficult to see in the images presented in Figure 4. A high magnification inset of the polarized light image (without the fiber analysis markup) might be beneficial. Additionally, there is no mention of how thick the samples are cut for AFM analysis. The results section states that all 4 paired samples measured by AFM had a significant decrease in stiffness. The data in Fig 4 shows that only 3 of 4 samples achieved significance ($P < 0.05$). This should be reconciled in the text (page 11). It would also be informative to know whether the AFM results are related to the original mammographic density score and/or treatment response. Finally, the immunofluorescence staining of Col VI seems to have high background (low signal to noise) and some of the signal seems to be within epithelial cells. Is this to be expected? How specific is this antibody?
4. A discussion of how this systemic treatment may impact other hormone-responsive organs/tissues could be valuable for broader implications of this prevention strategy.

Referee #3

(Remarks to the Author)

The findings reported in this manuscript are extremely timely given the increasing emphasis on prevention as an essential strategy to manage the growing cancer burden across the world. As the leading global cause of cancer-related mortality in women, interventions to prevent breast cancer, particularly in individuals at elevated risk are urgently required. Although

several drugs are recommended for this purpose, they act by blocking the action of estrogen in breast tissue and efficacy is restricted to the prevention of estrogen-receptor positive breast cancers. This manuscript describes an alternative approach through the use of anti-progestins in premenopausal women at increased risk and the authors demonstrate that ulipristal acetate (UA) can target, both directly and indirectly via stromal remodelling, the luminal progenitor cells, which are the proposed precursor of aggressive triple negative breast cancer.

It is important to note that the focus on analysing normal breast tissue from high-risk women who do not have cancer is a major strength of this study. These tissues and individuals represent the true target for preventive interventions, but to date very few published trials (across all cancers) have been able to access these types of patients and/or conduct such a comprehensive analysis of tissues.

Using a combination of molecular analyses, imaging and tissue micro-mechanics approaches, the authors present exciting new mechanistic insight showing that 12 weeks of daily UA reduces proliferation of breast epithelial cells and, more specifically, decreases the proportion, proliferation and ex vivo colony forming capacity of luminal progenitor cells in women at increased risk of breast cancer. High mammographic breast density is increasingly being used clinically as a biomarker of risk, but the mechanisms underlying the association are not well defined; the findings from this study provide new insight to help explain these links. MRI imaging showed reduced fibroglandular volume after UA, and this was consistent at a molecular and tissue level with extracellular remodelling and reduced collagen organisation and tissue stiffness, as shown by gene and protein expression changes, histology and atomic force microscopy. Most notably, inhibition of progesterone receptor signalling by UA led to downregulation of Collagen VI, which was spatially associated with LP cells, establishing a link between collagen organization and LP activity. The authors propose a model whereby progesterone paracrine signalling influences fibroblasts and basal cells within normal breast tissue, leading to remodelling of the extracellular matrix and increased stiffness; this in turn further activates progesterone receptor signalling in a positive feedback loop. Anti-progestins disrupt this mechanism, reducing collagen expression and stiffness, which results in decreased numbers of LP cells and since these are the purported cell of origin for triple negative breast cancer, this leads to reduced cancer risk. Overall, the findings support the important concept of precision prevention whereby anti-progestin therapy could be specifically targeted to women with high mammographic breast density, as these individuals are likely to experience the greatest benefit.

Other strengths of the manuscript include the design and conduct of the BC-APPS1 trial; with its strong translational work assessing diverse endpoints, it represents an excellent template for the design of other early phase prevention trials, which are an essential first step in developing preventive therapies. Additionally, in generating new molecular insight on how breast density and stiffness increase breast cancer risk through progesterone receptor signalling, the study provides a strong mechanistic rationale for using measures of breast density as surrogate biomarkers in early phase prevention trials. In doing so, the work addresses one of the key barriers to the development of preventive therapies, namely a lack of surrogate biomarkers. The findings could therefore pave the way for many future trials in breast cancer prevention. Whilst the demonstration that modulation of the tissue microenvironment can contribute to reduced cancer risk will provide the stimulus for more work in this area across other cancer types.

The manuscript is very well written, the results are clearly and appropriately presented. The interpretation of results and conclusions drawn are entirely appropriate. Relevant references seem to have been included to provide a balanced assessment of the field and impact of the findings.

Suggested improvements:

The BC-APPS1 trial with its strong translational work and diverse range of biomarkers represents an excellent template for the design of other early phase prevention trials. Although this point is made in the discussion, it should also be highlighted in the abstract/introduction.

Results, line 126: please include a reference here for the method used to assess risk of breast cancer.

Results, line 130. Briefly describe the relevance of Ki67 as the primary endpoint of the trial and given an indication of its limitations to highlight the value of the new findings.

Results, line 142. The section on epithelial colony formation would benefit from additional explanation on the methods as it was not entirely clear whether each sample could form each type of colony on the same plate. Also, were the same number of cells plated for each sample? It is not clear from the images (Fig 1E) what constitutes a single colony; some additional annotation or inclusion of further images would be helpful.

Results, line 147: The authors state that 'although the relative proportion of luminal colonies increased, the total fraction of LP cells decreased, suggesting lower absolute LP progenitor activity following treatment.' However, isn't the reduction in LP cells taken into account already by plating the same number of cells from pre/post treatment samples? I do not follow the argument based on the information given, unless there is an overall total reduction in the number of colonies. This section needs further explanation.

Results, line 153: The authors use SOX9 as a marker of LP cells, but this is different to the markers used to designate the different populations in Figure 1D and initially show that the LP population was reduced after treatment. How do the two LP populations overlap, are all the SOX9+ cells within the CD49f+/EPCAM+ population?

Results, line 157: The statement that the data demonstrate that treatment reduces the colony formation capacity of LP cells needs to be supported through clarification of the results, as indicated above.

Results, line 165: The manuscript states that 'RNA quality was suboptimal in at least 1 sample from 14 participants and data are presented for the paired samples from 10 participants that met quality standards'. Please clarify if samples were prepared from all participants that completed the trial and why only 10 paired samples were eventually analysed; what is the justification for excluding the other samples/data excluded?

In this respect, it would be helpful to explain in the methods how biopsy samples were prioritised for all the different types of analyses, given that not all analyses were conducted on all samples, and to understand the selection process for inclusion of patient samples for each type of analysis. The methods stated that 10 cores were obtained pre/post intervention but did all patients provide this many?

Results, line 185: Please include a key for Ext Figure 3A, or annotate the plots so it is clear where the LP cells are to support the quantitative data shown in Ext Figure 3C.

Results, line 265: It would be worth stating that the persistence of SOX9 cells with Co1-VI/FN1 was in the post treatment samples, to aid clarity.

Results, line 278: comment on the in vivo relevance of the two collagen mimetic hydrogels, how does the stiffness relate to human tissue?

Results, line 281: Explain why a different anti-progestin, other than UA was used for some experiments.

Results, line 285: The authors state that the 'results indicate that anti-progestin treatment prevents stiffness-induced upregulation of progesterone signalling and progenitor cell activity'. However, there was also an effect on SOX9 protein levels in the soft conditions, so it doesn't appear to be just a stiffness-induced upregulation; this needs to be commented on. What happens to the genes (TNFSG11 and C-KIT) at the protein level under these different conditions?

Discussion, line 357: the authors suggest that reduced proliferation of LP cells due to treatment of anti-progestins may counteract the increased mutational burden resulting from recurrent progesterone stimulation of breast epithelium. If suitable samples were still available from the present study it would be interesting to test this hypothesis by comparing the somatic mutational burden in LP cells before and after treatment with UA. Even if a difference was not detected, this type of analysis could provide insight on the duration of intervention needed to have an impact on the mutation burden in these cells.

Discussion, Section beginning line 368: The authors should comment on how universal the effects they observe are likely to be across other anti-progestins. Furthermore, the impact of UA associated liver toxicity should be discussed in terms of how this work should be progressed clinically.

Discussion: Please comment on the magnitude of effect with respect to Ki67 and MD as biomarkers for prevention trials; how do the effects observed compare with other trials where these markers have been used and correlated with a measure of clinical efficacy?

Discussion: Did the authors see any evidence of differential effects between participants in their response to UA across the various types of analyses conducted? This would be worth a comment.

Methods, BC-APPS1 study: Please include a reference/website for the version of the Tyrer-Cuzick risk estimation programme used.

Methods, Ki67 staining: Please clarify whether the slides were scored automatically using software and the pathologist just verified the results, or whether the pathologist's scores correspond to the data presented in this manuscript. If the latter, were the slides (for both Ki67 and analysis of acinar/lobular ratio) scored blind?

Referee #4

(Remarks to the Author)

This manuscript explores how anti-progestin therapy (ulipristal acetate, UA) influences breast cancer formation. Leveraging longitudinal samples from the BC-APPS1 clinical trial, the authors apply multiple modalities – including MRI, transcriptomics, proteomics, histology and microscopy – to study the impact of UA treatment. These data suggest a reduction in the number of the cancer-precursor "luminal progenitor" (LP) cells and altered stroma-epithelium interactions. With scRNAseq, the authors further examine shifts in cell-cell signaling, identifying alterations in ECM-related pathways. By performing spatially resolved proteomic analyses (LCM and imaging mass cytometry), they corroborate these observations. By performing functional assays with organoids and AFM on tissue samples they further implicate tissue stiffness as a target of UA associated with breast cancer phenotypic features.

Overall, this is a unique and interesting study examining potential mechanisms by which UA can prevent breast cancer formation in premenopausal women at high risk. However, there are numerous issues with the way in which the work is described (e.g., the manuscript cannot be read without jumping back and forth to between the main text and methods; it is

often unclear what is being plotted without detailed investigative work by the reader) and the data analyses were performed (which may impact interpretation). While the study and data appear interesting and valuable, there is substantial work that needs to be done which could influence the findings. Until these points are resolved, it is hard for us to advise on topic such as impact.

Comments by Figure

Figure 1 and Extended Data Figure 1:

1. The data in Figures 1 and S1 are tied. Please fix your statistics to consider this. This is true for several other analyses as well.

2. Figure 1E - how were colonies classified?

Figure 2 and Extended Data Figures 2 & 3

1. Figure 2 - The annotation for the major cell populations should be more accurate and precise, especially for the cell types defined as perivascular and immune (Extended data Figure 2). For the clusters defined as immune, there are clearly 3-4 different clusters in the UMAP and finer annotation for these clusters would be useful for downstream analysis. Also, the reference signature the authors use is from breast milk. Please use a reference dataset that comes from similar or comparable sample collection (e.g., Murrow et al., 2022). In addition, by directly running differential gene analysis for each cluster can help identify the marker genes for each cluster.

2. Moreover, the characterization of the data is underdeveloped. There is no discussion of changes in cell abundance or activity, just a rapid transition to a receptor-ligand analysis. E.g., normalization for cell numbers is mentioned at line 190 but those numbers are never shown making it hard to understand how significant the shifts in overall signaling really are.

3. Figure 2C - Please fix your legend

4. Lists of DE genes are missing to enable interpretation of the enrichments (only LRs are provided).

5. Links between 2D & F and 2E & G are missing.

6. Line 214 - Why did you restrict the target genes to the Reactome ECM gene set? This is not justified and trivializes the results in lines 212 - 233. Also, the comment in line 227 - 229 comes out of nowhere.

7. Extended Data Table 3 - Please be quantitative and use proper statistics.

8. Figure 2H - For the ligand-receptor interaction of collagen, are the receptors also down-regulated after UA treatment? This panel is hard to interpret. Please show single-cell expression data to demonstrate that collagen is downregulated post-UA.

9. Figure 2I - For the NicheNet analysis, the ligand expression is presented in binary. Please show a dotplot to give a sense for the percentage of cells expressing the ligands/target genes + average expression.

10. Extended Fig 2A - Something seems off here. No genes look clearly DE. Please double check your analysis. Also, what is the scale? Please make sure these are always added (e.g., Extended Fig 2C). And, how were the enrichments determined. The analytical steps are opaque.

11. Extended Data Table 1 and Extended Data Figure 3: is "HR" the same as "LM"?

12. Extended Data Figure 3D - The LP population shows substantial expression heterogeneity with very few cells expressing the selected genes. What are the cell numbers here? What are the expression levels? This panel is not very convincing. Please also show per patient values pre- and post.

13. Extended data figure 3F - the legend title is probably wrong, as q values should not be higher than 1.

14. We understand why you focus on FB and BA cells, but a more comprehensive analysis is needed and warranted.

Figure 3 and Extended Data Figure 4

1. Line 241-242. Unexpected why? Was the direction of change consistent?

2. Figure 3C - Why was patient treated as a random effect rather than explicitly considered when it was modeled in the bulk RNA work?

3. Figure 3D - Could use a more comprehensive analysis.

4. Extended Data Figure 4A - This would make much more sense shown as a volcano plot, especially to examine consistency for lines 249-252.

5. Extended Data Figure 4B - Markers selected how?

6. Figure 3F and Extended Data Figure 4C - Please do not use green-red.

7. Line 259 - How were cells classified as high vs low? Also, what fraction are Ki67+?

8. The strong spatial association between the Collogens and SOX9hi LP cells is interesting. Does colocalization shift as a result of treatment? We'd like to better understand how treatment folds into the results shown in Figure 4G and Extended Data Figure 4C&D.

Figure 4 and Extended Data Figure 5

1. Figure 4A&B - How was expression measured? There are several details required to understand your analyses that are missing. Also, please do a tied analysis (Ditto for 4D & 4F).

2. Figure 4C - Please quantify and show replicates.

3. Lines 284-286 - There are logic gaps here.

4. Figure 4G & Extended Data Figure 5 - We would suggest replacing this with standard methods for finding biomarkers (ROC), filtering for expression change (since P19 would be missed). Also, these markers need to be validated.

Other points:

1. How were potential confounders (like birth control) accounted for in the analyses?

2. The resolutions among figures are very different and some are very hard to visualize.

Version 1:

Reviewer comments:

Referee #1

(Remarks to the Author)

The authors addressed all my major concerns.

For the sake of not propagating incorrect (albeit widely used) nomenclature, please correct the following sentence in the introduction:

In both mouse and human mammary glands, lobular-alveolar development results from progesterone induced proliferation of the stem/progenitor cell population.

Should read: "increased branching or increased ductal complexity" instead of "lobular-alveolar development"

Referee #2

(Remarks to the Author)

The authors have substantially revised the manuscript and addressed the prior concerns of this reviewer. The manuscript is now acceptable for publication.

Referee #3

(Remarks to the Author)

The authors have addressed all my comments very well in their revised manuscript and letter of response. Furthermore, the changes made and considerable new data added as a result of the feedback from the other referees have led to an even stronger manuscript.

This study remains extremely important to the field; representing an excellent example of an early phase prevention trial, underpinned by high quality translational research. It supports the important concept of precision prevention whereby anti-progestin therapy could be specifically targeted to women with high mammographic breast density and provides a strong mechanistic rationale for using measures of breast density as surrogate biomarkers. I look forward to following the undoubted impact that this manuscript will have on breast cancer prevention and more widely on other tumour types.

Karen Brown

Referee #4

(Remarks to the Author)

Overall, the authors' revisions have addressed many of our previous concerns. The narrative has become clearer, the data presentation largely improved, and the methods section is now more thorough. However, a few issues of differing significance remain (see below; organized according to our original points).

Figure 1 and Extended Data Figure 1:

1. The data in Figures 1 and S1 are tied. Please fix your statistics to consider this. This is true for several other analyses as well.

We appreciate authors' clarifications around the use of paired statistics. Adding lines connecting data points from the same patient, pre and post, would aid interpretation (like current extended figure 10C). Please do this consistently.

2. Figure 1E - how were colonies classified?

This point has been well addressed but one clarification is needed. If the cells are seeded at two different densities, which one is used in the results section? This should be specified.

Figure 2

The authors have done a better job examining their single-cell data but the paper would benefit substantially from better linkage between the findings described in Fig 2A-F and Fig 2G-K (i.e., how does the downregulation observed connect to the interactions presented?).

6. Line 214 - Why did you restrict the target genes to the Reactome ECM gene set? This is not justified and trivializes the results in lines 212 - 233. Also, the comment in line 227 - 229 comes out of nowhere.

Current narrative is clearer. Two remaining points: 1. Legend in 2D refers to a star for significance but gives a p-value –

please proofread; 2. For Figure 2H, is there a statistical test to confirm the decrease of strength of these pathways?

8. Figure 2H - For the ligand-receptor interaction of collagen, are the receptors also down-regulated after UA treatment? This panel is hard to interpretate. Please show single-cell expression data to demonstrate that collagen is downregulated post-UA.

Figure 2J does not make clear how much collagen decreases post treatment. Plots like those in Extended Data 10D would be more convincing. In addition, Figure 2K does not help the statement in line 248-251 regarding the downregulation of collagen production. To highlight that the collagen is mostly acting on epithelial clusters, you should either use the shade/size of the arrows to indicate strength or at least highlight the epithelial portion of the circos plot. Another approach to check the interaction strength is to look at the expression of collagen receptors on the receptor cell types, rather than only looking at ligand expression.

One additional comment on Figure 2: the authors mention briefly that LHS upregulates MMP genes after treatment. This helps support the idea of an overall stiffness decrease, but is only reported in the text and Supplemental Table 2. It might be worth to bring into a main figure or supplemental figure.

Figure 3

1. Line 241-242. Unexpected why? Was the direction of change consistent?

To be clear, the overlap of detected proteins does not indicate the reliability of the data. We suggest removing this statement.

2. Figure 3C - Why was patient treated as a random effect rather than explicitly considered when it was modelled in the bulk RNA work?

As pre and post treatment samples are paired from the same patient, your statistical tests should use fixed effect. It would be best to build a GLM that models patient.

7. Line 259 - How were cells classified as high vs low? Also, what fraction are Ki67+?

This requires better quantitation, and the threshold testing should be done based on intensity.

Figure 4

Figure 4A&B - How was expression measured? There are several details required to understand your analyses that are missing. Also, please do a tied analysis (Ditto for 4D & 4F).

See comment in Figure 1 point #1 about tie lines.

3. Lines 284-286 - There are logic gaps here.

Mostly addressed but the authors should be clear in line 329-331 that onapristone does not decrease MFE in the soft condition.

Small comments:

1. Figure quality and consistency could be improved. For instance, related main and supplemental figures should have the same color range (q values in Figure 3E and Suppl Figure 11A) and consistent nomenclature (T1/T2 versus baseline/post-treatment)

2. Some abbreviations are not explained in the figure legends. For example, VST in extended figure 2A and cell types in extended figure 5. Please fix.

Version 2:

Reviewer comments:

Referee #4

(Remarks to the Author)

The authors have satisfactorily addressed our remaining concerns.

Rebuttal Letter for BC-APPS1 Manuscript (#2024-05-10498A)

We are very grateful to the reviewers for their insightful and constructive feedback on our manuscript “*BC-APPS1: Anti-progestin therapy targets hallmarks of breast cancer risk through epithelial-stromal remodelling*”. We have addressed the comments thoroughly and believe that the changes made have significantly strengthened the manuscript.

In our original submission, we reported findings from the Breast Cancer Anti-Progestin Prevention Study 1 (BC-APPS1), a clinical trial evaluating the progesterone receptor antagonist ulipristal acetate (UA) as a primary preventive agent in premenopausal women at increased familial risk of breast cancer. Our multidisciplinary team across Manchester, Toronto, and Cambridge integrated clinical imaging, single-cell transcriptomics, functional assays, and spatial proteomics to establish key mechanisms by which anti-progestin therapy exerts preventive effects. Specifically, we demonstrated that: **1)** UA treatment reduced the proliferation, number and activity of luminal progenitors (LP), the cell of origin of aggressive breast cancers; **2)** Marked remodelling of the ECM was induced, reducing collagen alignment and tissue stiffness, and downregulating collagen VI, which was spatially associated with Sox9^{high} LP cells; **3)** Serial MRI scans revealed a significant reduction in fibroglandular volume, a surrogate for mammographic density, a major breast cancer risk factor.

The reviewers’ comments and suggestions have led us to undertake additional analyses and experiments that have provided novel insights into the mechanistic links between luminal hormone sensing (LHS) cell - fibroblast paracrine signalling, determined that similar effects are seen with an alternative PR antagonist - suggesting this is a class effect - and shown that the effect is largely limited to those women with raised mammographic density, providing a potential predictive biomarker to be tested in future clinical trials. The major revisions to the manuscript are summarised below, followed by a detailed point by point response addressing the reviewers’ comments:

1. Clarification of genetic risk and exclusion of BRCA1/2 carriers

We have expanded the clinical description of our study population to demonstrate that none of the 24 participants were known carriers of BRCA1 or BRCA2 pathogenic variants (PVs). Using the BOADICEA v7 algorithm, we calculated the probability of BRCA1/2 PVs for each participant and found a median BRCA1 PV probability of only 0.43%. The cumulative probability that all participants were BRCA1 PV-negative is 83.53%. These data are now presented in **Extended Data Table 1** and argue against the suggestion that the observed effects are driven by aberrant BRCA1 heterozygous progenitors.

2. Enhanced scRNAseq analysis with Human Breast Cell Atlas mapping

To improve the analysis and resolution of our scRNAseq dataset, we have incorporated label mapping from the integrated Human Breast Cell Atlas (iHBCA; Reed et al., 2024). This refinement improved cell annotations and the precision of our analyses (**Extended Data Figure 5**). Key changes include:

- Dot plots illustrating marker genes for main clusters (**Figure 2C**) and subclusters (**Extended Data Figure 6**), as well as expression of collagen genes across subclusters (**Figure 2J, Extended Data Figure 9**).
- Differential cell abundance analysis showing proportionality shifts across clusters, including a decrease in luminal adaptive secretory precursor (LASP) cells following UA treatment (**Figure 2D, Extended Data Figure 4C–D**).
- Differential gene expression analysis using Memento, highlighting significant downregulation in luminal hormone sensing (LHS) cells gene expression (**Figure 2E, Extended Data Figure 7A**).
- Pathway analysis of post-UA effects across the seven main cell types (**Figure 2F, Extended Data Figure 8A**).

- Cell–cell communication analysis using CellChat, now with subcluster-level resolution, identifying ECM-related signalling changes (particularly collagen pathways) as the principal driver of the observed cellular responses to UA treatment (**Figure 2I–K**).
- Presentation of gene expression analyses for key LHS ligands (*WNT5A*) and target genes (*COL6A3*) (**Extended Data Figure 10**).

3. Expanded Imaging Mass Cytometry (IMC) analyses

We have reformatted and clarified the ECM intensity quantification plots for Collagen VI, Collagen I, and Fibronectin within SOX9^{high} and SOX9^{low} populations (**Figure 3G, Extended Data Figure 11C**). We also include a per-participant heatmap illustrating the direction of protein changes following UA treatment (**Extended Data Figure 11A**). In addition, we now demonstrate that Ki67+ cells are significantly more frequent in SOX9^{high} compared to SOX9^{low} populations at both baseline and post-treatment, confirming the higher proliferative activity of the SOX9^{high} population (**Extended Data Figure 11D**).

4. Confirmation that the in vitro effects are comparable between two independent progesterone receptor antagonists (PRA)

We show that UA treatment of baseline cells led to a reduction in mammosphere-forming efficiency (MFE), consistent with the effects previously demonstrated using onapristone (**Extended Data Figure 1G**). Furthermore, we validated that both PRAs reduce both MFE and expression of progenitor markers (SOX9, C-KIT) in soft and stiff hydrogel cultures of breast epithelial organoids (**Figure 4B–C, Extended Data Figure 12A–B**) confirming this to be a class effect of PRAs.

5. Stratification of responses

We examined participants' baseline characteristics - age, breast cancer risk, parity and BMI - to assess whether they influenced response to UA in terms of reduced proliferation (Ki67) or changes in LP/LASP cell populations. No significant correlations were observed. We also examined baseline Volpara Density Grades (VDG 1–4), corresponding to BIRADS categories A–D, and observed statistically significant reductions in Ki67 only among participants with high breast density (BIRADS C/D, $n=17$), but not in those with low density (BIRADS A/B, $n=6$), as shown in **Figure 4G**, identifying mammographic density as a potential biomarker of PRA activity that could be used for participant selection in future studies.

6. Additional clarifications and figure enhancements

In response to reviewers' comments, we have:

- Revised the Methods section to clarify protocols for epithelial colony formation, single-cell gene expression and communication analyses, atomic force microscopy and statistical analysis.
- Provided our prioritisation strategy for allocating biopsy samples across various assays.
- Updated figure colour schemes to improve visual clarity (e.g. avoiding red-green contrasts).
- Included other enhancements such as extra colony photos (**Extended Data Figure 1E**) and the per-participant ECM protein heatmaps (**Figure 3E; Extended Data Figure 11A**).
- Revised the final model in **Figure 4I** to depict the *WNT5A–COL6A3* signalling axis uncovered by our analyses.

We hope that the comprehensive revisions and additional analyses we have undertaken satisfactorily address the reviewers' comments and enhance the clarity and robustness of our study. Our findings establish that anti-progestin therapy target both epithelial and stromal cell populations involved in early breast cancer development, providing a rationale for molecularly guided prevention strategies.

The table below summarises the key revisions made to the figures and tables in the manuscript, including addition of 15 new or revised main figure panels, along with a brief description of the data or analysis presented in each.

Updated or New Figure/Table	Summary of updated or new content
Extended Data Table 1	BRCA1/2 mutation probabilities per participant, confirming low PV genetic risk across the cohort.
Extended Data Figure 1E	Additional colony formation assay images.
Extended Data Figure 1G	UA reduces mammosphere-forming efficiency validating onapristone findings.
Figure 2C	Dot plots of marker genes for main cell clusters.
Figure 2D, Extended Data Figure 4A	Cell abundance analysis showing proportional changes in cell populations revealing reduced LASP cells post-UA.
Extended Data Figure 4C–D	Cell abundance analysis showing no significant proportional changes in subcluster cell populations.
Extended Data Figure 5	Cell type annotations of subclusters refined by Human Breast Cell Atlas mapping.
Extended Data Figure 6	Dot plots of marker genes for cell subclusters.
Figure 2E	Differential expression analysis highlighting downregulated genes in LHS cells post-UA.
Extended Data Figure 7A	UpSet plot of up-regulated genes across cell types after UA treatment.
Figure 2F, Extended Data Figure 8A	Pathway analysis of UA effects across main cell types.
Extended Data Figure 8B	UpSet plot of ECM genes down- and up-regulated across cell types after UA treatment.
Figure 2I	CellChat analysis of BMYO and FB sub-clusters.
Figure 2J, Extended Data Figure 9	Expression of collagen genes across subclusters.
Figure 2K	CellChat analysis of collagen signalling in BMYO and FB sub-clusters.
Extended Data Figure 10A	NicheNet analysis of LHS ligands regulating collagen genes in FB sub-clusters.
Extended Data Figure 10B–D	Gene expression plots for LHS ligands, WNT5A (LHS) and COL6A3 (FB).
Extended Data Figure 10E	Gene expression of receptors for LHS ligands in FB.
Figure 3C	Recoloured volcano plot to identify Matrisome proteins.
Figure 3E, Extended Data Figure 11A	Heatmap of ECM protein changes per-participant post-UA.
Figure 3F, Extended Data Figure 11C	Updated colour scheme to avoid red-green contrast.
Figure 3G, Extended Data Figure 11C	Reformatted ECM intensity plots to show expression in SOX9 ^{high} and SOX9 ^{low} populations in T1 vs T2.
Extended Data Figure 11D	Ki67+ cell frequency is higher in SOX9 ^{high} vs SOX9 ^{low} and is reduced by UA.
Figure 4B–C	UA reduces MFE and progenitor markers (SOX9, C-KIT) in soft and stiff hydrogels.
Extended Data Figure 12A–B	Additional data from organoids showing reduced SOX9/C-KIT expression after UA treatment.
Figure 4G	Reduced Ki67 post-UA only in participants with high breast density (BIRADS C/D).
Figure 4I	Updated model to include WNT5A–COL6A3 signalling as an UA-regulated axis.

Referees' comments

Referee #1 (Remarks to the Author):

A. Summary of the key results

The authors report the results of the clinical trial BC-APPS1. This window trial aimed at assessing the effects of the progesterone receptor antagonist ulipristal acetate in breast cancer prevention. Twenty-four premenopausal women with inferred high risk because of their family history were treated for 12 weeks with daily UA and pre- and post-treatment tissue OMICs and imaging analyses were performed.

Decrease in progenitor cell activity was decreased as assessed by colony formation, mammosphere assays and through scRNA seq analysis. Breast density, considered surrogate marker of breast cancer risk, decreased and the expression of ECM encoding genes was decreased and this is attributed to changes in fibroblasts and basal cells. Experiments with hydrogels of different stiffness show that stiffness induced PR target gene expression is abrogated by UA treatment. Hyperion imaging mass cytometry showed that luminal progenitors identified as SOX9high cells are in close proximity to regions with high expression of the non-fibrillar COLVI.

The authors conclude that the progesterone receptor antagonist UA, holds promise as a preventative treatment strategy in premenopausal patients.

B. Originality and significance: if not novel, please include reference

The role of PR signaling in breast carcinogenesis is a topic of ongoing debate. As such the present study with new clinical data will be a very valuable contribution to the breast cancer field. The design of the trial with consideration of the luteal phase and hormone measurements is original and improves over previous designs.

C. Data & methodology: validity of approach, quality of data, quality of presentation

There is little information for the genetic basis of the increased BC risk of the women included in this study. To what extent can it be attributed to germline mutations in BRCA1 or BRCA2? Those are likely to be involved and could limit the extrapolation of the findings to the population at large as it has been shown that BRCA1 germ line mutation carriers have an increased compartment of aberrant progenitor cells (Lim Nat Med 2009) which could confound analysis and interpretation of the results.

Thank you for this important observation. The population recruited for this study were from the Breast Cancer Family History clinic in South Manchester and none were known to carry a BRCA1 or BRCA2 pathogenic variant (PV) at entry or have been shown to be carriers subsequently.

Two participants had undergone personal testing (01 and 19) both of whom were negative for gene panels (12 and 30 gene panels respectively performed outside of the NHS). The NHS National Institute for Health and Care Excellence guidelines (CG164) stipulate that genetic testing of an affected individual in the family is prioritised and undertaken if the *a priori* probability of identifying such a PV is at least 10%. In seven BC-APPS1 participants an affected family member had undergone genetic testing. In 3 (05, 10 and 13) a BRCA2 PV had been identified in an affected parent, with normal BRCA1 sequencing, but the BRCA2 PV was not identified in the study participants. In one of these cases (10) both parents were affected with BC but testing had been undertaken only in the paternal line. In the other 4 families the affected parent (participants 6,11 and 12) or second degree relative (maternal aunt of participant 7) had undergone negative screening for PVs in both BRCA1 and BRCA2 and,

appropriately, no additional genetic screening was undertaken in these participants. In the remaining 15 participants with paired VAB samples, no genetic testing was undertaken as the *a priori* probability of identifying a BRCA1/2 PV was less than the NICE CG164 threshold of 10%. We have calculated the probabilities of BRCA1 and BRCA2 PVs using the BOADICEA v7 algorithm and provided these in Extended Data Table 1. The median probability of a BRCA 1 PV is only 0.43% and the probability that all 24 individuals are BRCA1 PV negative is 83.53%. These data strongly refute the suggestion that the effect is driven by aberrant progenitors in BRCA1 PV carriers.

Approach testifies to close collaboration between disciplines and the authors are complemented on coordination of sample collection with the participants' luteal phase. For the post treatment point, is there still a luteal phase to speak of, after 12 weeks of UA treatment? The authors should explain what happens to the participants' endogenous hormone levels upon treatment.

This is a good point and one we believe is covered in the second results paragraph where we describe the effect of UA treatment on serum progesterone levels. *“Mean serum progesterone levels reduced with treatment from 36 nmol/L (95% CI, 29.4 – 41.6 nmol/L) at baseline to <3 nmol/L (95% CI, 0.3 – 4.6 nmol/L; p<0.0001) (Extended Data Figure 1A).”* We have also added *“effectively abrogating the luteal phase”* to the revised manuscript for clarity. (Lines 139-142)

D. Appropriate use of statistics and treatment of uncertainties

The Extended Data Table illustrates well the challenges of translational research studies with the difficulty of obtaining multiple datasets from individual patient samples. The authors may consider adding it to the main manuscript. Any genetic information and more details about the family history should be added to this table. The known role of BRCA1 in the progenitor cell population may limit interpretability of results from a subset of samples but may also add insights if considered as additional variable.

Thank you for this suggestion, as above we have amended the Extended Data Table 1 in supplementary to include the probability of BRCA1/2 PV and we have also added the number of first, second and third degree relatives affected by breast and ovarian cancer. We cannot add each age of the relatives as this would make individual participants identifiable. As the table is now very large we believe it remains best placed in supplementary.

E. Conclusions: robustness, validity, reliability

The conclusion on the general utility for premenopausal prevention is overdrawn as the women included in the present study have high familial risk. Some of this (BRCA1mt) has been linked to aberrant progenitor populations and confounds the study.

As above we believe we have strong evidence to support the participants not being carriers of BRCA1 PVs and, therefore, that the effect is not driven by aberrant BRCA1 heterozygous progenitors. These data are now included in Extended Data Table 1.

F. Suggested improvements: experiments, data for possible revision

Patients with normal risk should be included.

Thank you for this suggestion. As the study has closed it was not possible for us to modify eligibility and recruit normal risk women and treat them with UA. However, we have obtained normal risk tissue samples from women undergoing breast reduction surgery and subjected these to *in vitro* treatment with UA in in soft and stiff gels. The results, presented in figure 4, are comparable in effect size and direction to the samples from women at increased familial risk.

	BC risk	Soft	Soft + UA	Stiff	Stiff+UA
BB7162T1N	Increased	0.18	0.101667	0.51	0.235
BB7164T1N	Increased	0.211667	0.176667	0.311667	0.25
1923PM	Increased	0.088333	0.076	0.11	0.061667
1715PM	Increased	0.301667	0.28	0.375	0.223333
1989N	Average	0.205	0.103333	0.365	0.235
3088N	Average	0.25	0.083333	0.276667	0.14

Table: Results for percentage of mammosphere-forming efficiency (MFE) are shown for breast samples at increased or average cancer risk as detailed in Extended Data Table 3.

The percentage of MFE was determined for four breast tissue samples from women at increased risk and compared with two samples from women at average risk. Although the sample size is small, the trends are consistent across both groups: there is an increase in MFE in cells previously grown under stiff matrix conditions compared to those grown in soft conditions. Additionally, treatment with ulipristal acetate leads to an overall reduction in MFE under both soft and stiff matrix conditions.

Furthermore, we used normal breast samples from individuals at average risk (1989N and 3088N) to demonstrate that expression of SOX9 and C-Kit proteins increases under stiff matrix conditions. This increase is inhibited by treatment with ulipristal acetate (Figure 4B and Extended Data Figure 12A).

We believe these data substantiate our initial assertions in the manuscript that the effect is likely to be generalisable to those at increased risk through factors other than a family history. We have strengthened the statement in the conclusions to highlight this requirement for future translational studies: *“This should include further testing of the hypothesis that PR antagonism may reverse MD associated BC risk more broadly, beyond those at increased familial risk.”* (Lines 444-446)

MInor: The following statement is somewhat misleading: Lines 71ff “Supplementation of progestin, either as a contraceptive or hormone replacement Magnetic Resonance Imaging (MRI) measurements of fibro glandular volume (FGV) correlate well with automated volumetric MD, and FGV is greater in the luteal vs follicular phase of the menstrual cycle.

The progestins are usually administered together with an estrogenic compound.

Thank you for this observation, which is of course correct for the use of HRT in women with an intact uterus. However, the majority of women using hormonal contraceptives, use progestin only pills, depots and intrauterine devices which have all been shown to increase the risk of breast cancer. Reference 10 in our manuscript (Morch et al 2017) was one such study confirming this relationship. Since our original submission further supporting data have been published and are now referenced in the manuscript (Reference 12 - Tuesley KM et al. Long-acting, progestin-based contraceptives and risk of breast, gynaecological, and other cancers. J Natl Cancer Inst. 2025, PMID: 39805314). We have expanded our previous statement to: *“Supplementation of progestin, as a contraceptive or hormone*

replacement therapy (HRT), both with or without supplementary estrogen, increases BC incidence^{10, 11, 12} and stimulates epithelial proliferation and hyperplasia in preclinical models¹³." (Lines 74-77)

G. References: appropriate credit to previous work?

A lot of previous work has been overlooked such as very pertinent studies from Seema Khan, Claudia Lanari and Martin Widschwendter as well as work from Susan Clare, Cathrin Brisken, Pepper Schedin, Geoff Lindeman (Lim et al).

Thank you for this suggestion. We were constrained by word limits and reference numbers for the initial submission but have expanded the introduction and discussion to include these valuable additional data. Of note, from the suggested list of authors we had already cited the work of Lindeman (2 references) and Widschwendter. We have added a reference to Seema Khan's paper on the inhibition of mouse mammary tumourigenesis by telapristone to the introduction (PMID: 27080304, Reference 17), and that on telapristone's inhibition of ECM-associated gene transcription to the discussion (PMID: 31771627, Reference 44). We have also incorporated Cathrin Brisken's paper on progesterone induced proliferation and hyperplasia development in human breast tissue samples in the elegant MIND model (PMID: 34042278, Reference 13).

H. Clarity and context: lucidity of abstract/summary, appropriateness of abstract, introduction and conclusions

The manuscript is clear and the abstract, introduction and conclusions are appropriate. Previous window trials, albeit in the therapy setting should be discussed. The conclusions overdrawn .

Thank you for this comment. We believe we have addressed the concern about overdrawn conclusions by demonstrating that the observed effects are not attributable to BRCA1 or BRCA2 status, and are likely related to breast tissue density. Regarding previous window trials, we have revised a statement in the Introduction and cited additional studies, including telapristone (Lee et al., 2020, Reference 18) and mifepristone (Elia et al., 2023, Reference 19) window trials, to provide broader context.

The revised sentence reads: "*Conversely, inhibiting PR or its downstream pathways in mouse models results in significant reduction in mammary carcinogenesis, with studies showing suppressed mammary LP/stem cell activity^{6, 14, 15, 16, 17} and in clinical studies reduced proliferation has been seen with window studies in both normal and cancerous tissue^{9, 18, 19, 20}."* (Lines 77-80)

Referee #2 (Remarks to the Author):

The manuscript titled, "BC-APPSI: Anti-progestin therapy targets hallmarks of breast cancer risk through epithelial-stromal remodeling", demonstrates anti-progestin mediated alterations in the breast microenvironment of women at high risk for breast cancer. The authors provide a multi-omic analysis of paired biopsies from pre- and post-treatment breast tissues. The results show a significant reduction in the proliferation and activity of luminal progenitor cells. This is associated with a robust change in ECM composition, architecture, and tissue stiffness. Together, this data shows promise for targeting prolactin signaling as a prevention for breast cancers in premenopausal women. The manuscript is well written, the data is original, strong, and clearly presented, and the statistical analysis and references are appropriate. However, there are a few points that should be addressed to improve the manuscript prior to publication.

Major concerns:

1. The authors suggest that changes in mammographic density may be used as a surrogate for the risk reduction of anti-progestin therapy. In addition, mammographic density is implicated as a potential biomarker for response to anti-progestin therapy. The cohort of patients included in this study have a range of initial mammographic density scores and a range of treatment responses. How do the data (changes in ECM (proteomics), AFM/stiffness, and luminal progenitors) relate to the mammographic density scores? The data points in the graphs could be presented in a color-coded fashion (similar to Fig 4A) to represent response by mammographic density score. Correlation analysis between density score and anti-progestin response would strengthen the argument for mammographic density as a biomarker for anti-progestin treatment.

Thank for this very helpful suggestion. We first analysed the %VBD as a continuous variable and tested for correlations with baseline factors and none showed a statistically significance (see table below).

Baseline Variable	Number of participants	R value	P value
Ki67	23	-0.0016	0.99
LP (FACS)	16	-0.32	0.23
Mammosphere forming efficiency	18	-0.16	0.52
SOX9+ by immunofluorescence	10	-0.14	0.71

We next examined the correlation between %VBD and the fold change in these variables with treatment.

Fold change in:	Number of participants	R value	P value
Ki67	23	-0.0051	0.96
LP (FACS)	16	-0.05	0.85
Mammosphere forming efficiency	18	-0.42	0.083
SOX9+ by immunofluorescence	10	-0.55	0.096
Organised collagen by Picrosirius	21	0.12	0.61

None of these showed statistical significance although there were borderline significant results for greater reduction in MFE and SOX9+ cells with increasing VBD. We next examined the clinically utilised cut-offs for high vs low density by translating the %VBD to Volpara Density Grades which correlate well with to Breast imaging-reporting and data system (BIRADS) categories. BIRADS A/B are considered low breast density with categories C/D considered high. Using this approach we saw statistically significant reductions in Ki67, in those with high (n=17) but not low (n=6) breast density (Figure 4G). Similar patterns were seen in % of LP by FACS, MFE and % of Sox9+ cells, albeit with lower numbers in these analyses. This also leads into the RNAseq analysis well as the same categorisation (A/B vs C/D) was already used in the original manuscript.

The additional results have been added to the results section as follows:

*“In our study we did not find significant correlations between baseline percentage of Volumetric Breast Density (%VBD) as a continuous variable and baseline or fold change in any variable examined with UA treatment (data not shown). However, categorisation of %VBD into Volpara Density Grades (1-4) to approximate BIRADS categories (A-D) demonstrated statistically significant reduction in %Ki67 in those with high but not low MD (BIRADS C/D vs A/B; **Figure 4G**). A similar pattern was observed in LP frequency by FACS, MFE, and SOX9+ cell percentages (data not shown).”* (Lines 346-353)

With only 4 samples examined by AFM we do not have statistical power to examine %VBD correlation with baseline stiffness or its response to UA treatment but the results are included here for completeness.

Study ID	19	21	22	25
%VBD	20.7	6.3	5.7	17.9
AFM Baseline	0.2038	0.1303	0.1055	0.1287
AFM Fold change	0.476594701	0.952417498	0.220663507	0.595648796

2. While the authors provide evidence that increased mammographic density scores have higher progesterone signaling, there is no investigation into Progesterone Receptor (PR) protein expression or which cells are expressing PR. Does mammographic density alter the expression of PR on the epithelium or fibroblasts and how does PR expression relate to treatment response? PR expression along with mammographic density scores could provide a signature biomarker for treatment.

We thank the reviewer for the comment regarding PR protein expression and its potential relevance to mammographic density and treatment response.

To investigate PR protein expression, we performed IHC on 18 paired BC-APPS1 biopsy samples using an automated staining platform (BenchMark Ultra, Ventana Medical Systems). The Confirm Anti-PR antibody (clone 1E2, #790-2223) was used with the UltraVIEW universal DAB detection kit. Slides were scanned using a Leica SCN400 slide scanner, and quantification of PR-positive epithelial cells was carried out using HALO™ Image Analysis software (v3.2.1851), with a minimum of 2500 cells assessed per sample. PR expression at baseline showed substantial variability, ranging from 3% to 42% of epithelial cells, and there was no consistent trend in PR expression change following treatment (representative images and graph showing % PR cells for N=18 is shown below).

Baseline PR+ IHC staining

Post-treatment PR+ IHC staining

Figure: Percentage of PR-positive epithelial cells at baseline and post-treatment for n = 18 paired samples. No consistent change in PR expression was observed following treatment.

Our IHC analysis revealed that PR expression was exclusively detected in epithelial cells within the acinar structures, with no observable staining in the stromal compartment. This was confirmed both visually on low-magnification images and through automated quantification. Whilst we acknowledge the possibility of low-level PR expression in other cell populations, we did not detect any PR-positive staining in the stromal regions of our normal breast biopsies.

Figure: Low magnification image showing PR expression restricted to epithelial cells within the acinar structures of breast tissue with no detectable expression in the surrounding stromal compartment.

This observation is supported by our single-cell RNA sequencing data, which further demonstrated that PR expression is predominantly restricted to the luminal hormone-sensitive (LHS) epithelial cells (11% PR+), with minimal expression detected in fibroblasts (FB, 1.3%) or other populations (LASP 1.6%, BA 2%, IM 0.1%, VA 1.7%, EN 0.2%).

We found no correlation between baseline PR expression and change in Ki67, suggesting that PR levels do not predict treatment response in this setting.

Figure: Correlation plot of fold-change in percentage of Ki67+ cells versus percentage of PR-positive epithelial cells at baseline. No significant association was observed.

We also examined whether PR expression correlated with mammographic density scores (VBD) at baseline. As shown in the plot below, no association was observed, indicating that PR protein levels

are not linked to the relationship between mammographic density and progesterone signalling activity observed in our study.

Figure: Correlation plot of percentage VBD versus percentage of PR-positive epithelial cells at baseline. No significant association was observed.

In summary, our data demonstrate that PR expression is almost entirely confined to the epithelial compartment, does not vary consistently with treatment, and is not associated with either mammographic density or treatment response. Thus, whilst PR is a key mediator of progesterone signalling, its expression alone does not appear to serve as a predictive biomarker in this context.

3. A major finding of the manuscript is that anti-progestin treatment remodels the ECM (stiffness and architecture), however, the presentation of these results could be improved. The change in architecture as measured by birefringence seems very subtle and is difficult to see in the images presented in Figure 4. A high magnification inset of the polarized light image (without the fiber analysis markup) might be beneficial. Additionally, there is no mention of how thick the samples are cut for AFM analysis.

To enhance clarity we have now included high-magnification insets of the polarized light images in Figure 4D, without the fibre orientation markup overlays to better visualise the changes in collagen birefringence following UA treatment.

We have also updated the Methods section to specify that breast tissue sections used for AFM measurements were cut at a thickness of 7 μ m, along with additional details. This was optimised to preserve native tissue architecture while providing suitable integrity for AFM analysis.

In the manuscript Methods section “Tissue stiffness by atomic force microscopy”, the following text has been added:

“7 μ m-thick sequential cryosections were obtained for each participant sample. Three peri-lobular regions (100 x 100 μ m) were identified for each participant sample from a H&E-stained section, and the same region was then located on an unstained sequential slice to be probed. Immediately prior to the experiment, each participant sample was allowed to thaw and dry at RTP for 2 hours, followed by five quick washes in deionised water to remove the Optimal Cutting Temperature (OCT) compound.”
(Lines 932-937)

The results section states that all 4 paired samples measured by AFM had a significant decrease in stiffness. The data in Fig 4 shows that only 3 of 4 samples achieved significance ($P < 0.05$). This should be reconciled in the text (page 11). It would also be informative to know whether the AFM results are related to the original mammographic density score and/or treatment response.

We have revised the text to clarify that while all four samples demonstrated a decrease in stiffness following UA treatment, only three of the four reached statistical significance ($P < 0.05$). The fourth sample showed a similar trend but did not meet the threshold for significance, likely due to the variability in stiffness measurements within that sample. To clarify this point, we have incorporated the changes highlighted below in bold in the Results section “Tissue stiffness amplified progesterone response and LP activity are inhibited by anti-progestins”:

*“Atomic force microscopy (AFM) of four paired samples that had at least 10% reduction in collagen organisation by PSR showed **a consistent decrease in tissue stiffness, with three reaching statistical significance** (Figure 4E).”* (Lines 338-340)

We also explored whether AFM-derived changes in stiffness were associated with baseline mammographic density (VBD) or treatment response (as assessed by changes in Ki67). No clear correlation was observed in this small subset ($n=4$) likely due to limited sample size (see table above - page 9).

Finally, the immunofluorescence staining of Col VI seems to have high background (low signal to noise) and some of the signal seems to be within epithelial cells. Is this to be expected? How specific is this antibody?

Imaging mass cytometry (IMC) was employed to visualize Collagen VI in the lobular and peri-lobular regions of donor tissues. For detection via IMC, gadolinium-160 was conjugated to a polyclonal antibody targeting the Collagen alpha-1(VI) chain (COL6A1) (Abcam, #ab6588). This antibody has been cited in 151 publications to date, including several immunohistochemistry (IHC) applications [https://www.abcam.com/en-us/products/primary-antibodies/collagen-vi-antibody-ab6588?srltid=AfmBOoq5_k9bRRArch8W-N-CTqfEGNpt6GgcNKXN5MBRvPOxB2oNxx2q#drawer=publications].

Below we present additional IHC data from breast tissues using an alternative, independent anti-COL6A1 antibody [Proteintech, 17023-1-AP] – middle and right panels. This IHC staining exhibits a comparable distribution and intensity of Collagen VI staining in the lobule and peri-lobular stroma, which aligns with our IMC data (Figure 3F). Notably, staining is observed in the epithelial compartment, where it is distributed non-uniformly, as demonstrated in Figure 3F.

Figure: IHC staining using an independent anti-COL6A1 antibody (Proteintech, 17023-1-AP) confirms Collagen VI expression consistent with IMC data. Red arrows indicate acini with high Collagen VI expression, while blue arrows indicate acini with low expression.

Moreover, the scRNA-seq data (Extended Data Figure 9, see below) reveals expression of *COL6A1* and *COL6A2* in epithelial cell populations, particularly in basal myoepithelial (BMYO) and luminal adaptive secretory precursor (LASP) cells, as well as in stromal compartments, including fibroblasts (FB) and perivascular (PV) cells.

Figure: scRNA-seq data showing expression of COL6 genes in epithelial (BMYO, LASP) and stromal (FB, PV) cell populations.

In addition, the Human Protein Atlas (HPA), a reputable external repository of IHC data generated with antibodies validated through orthogonal methods (<https://www.proteinatlas.org/>), provides additional breast tissue examples stained with an independent anti-COL6A1 antibody, showing the presence of heterogeneous staining in the epithelial compartment [<https://www.proteinatlas.org/ENSG00000142156-COL6A1/tissue/breast#img>].

We believe that the consistency of staining patterns across three independent antibodies and assays provides strong evidence that the anti-Collagen VI antibody used in our IMC panel accurately reflects the true distribution and abundance of Collagen VI in human breast tissue, a conclusion further supported by our BC-APPS1 scRNA-seq data.

4. A discussion of how this systemic treatment may impact other hormone-responsive organs/tissues could be valuable for broader implications of this prevention strategy.

This is a very useful suggestion. Reviewer 3 also requested contextualisation of the hepatotoxicity of UA in the final discussion paragraph. We have added a short section in the final paragraph that highlights that further work is required to fully establish the effects on other organs and the safety of SPRM therapy. Space limitations prevent more detailed expansion but numerous reviews are available for interested readers.

The final paragraph now reads as follows with the alterations highlighted in bold:

*“In summary, our work identifies progesterone signalling as a key regulator of the BC-precursor LP/LASP cell population in the normal breast, and establishes a complex interplay between anti-progestin treatment, ECM remodelling, and LP/LASP cell dynamics. **Comparable in vitro effects with two independent anti-progestins, along with pre-existing clinical data on other PR antagonists^{9, 18}, suggest these may be class-effects of PR antagonism. Longer term studies are required to evaluate safety, particularly hepatotoxicity and effects on other hormone sensitive tissues such as the endometrium. BC is the most common cause of cancer death globally and such studies need to be progressed urgently. This should include further testing of the hypothesis that PR antagonism may reverse MD associated BC risk more broadly, beyond those at increased familial risk. MD reporting is now mandated in all US states by the FDA, albeit without any recommendation on methods to reduce it. Collectively, our study offers a roadmap for strategic design of molecularly informed primary prevention clinical trials.**”* (Lines 437-448)

Referee #3 (Remarks to the Author):

The findings reported in this manuscript are extremely timely given the increasing emphasis on prevention as an essential strategy to manage the growing cancer burden across the world. As the leading global cause of cancer-related mortality in women, interventions to prevent breast cancer, particularly in individuals at elevated risk are urgently required. Although several drugs are recommended for this purpose, they act by blocking the action of estrogen in breast tissue and efficacy is restricted to the prevention of estrogen-receptor positive breast cancers. This manuscript describes an alternative approach through the use of anti-progestins in premenopausal women at increased risk and the authors demonstrate that ulipristal acetate (UA) can target, both directly and indirectly via stromal remodelling, the luminal progenitor cells, which are the proposed precursor of aggressive triple negative breast cancer.

It is important to note that the focus on analysing normal breast tissue from high-risk women who do not have cancer is a major strength of this study. These tissues and individuals represent the true target for preventive interventions, but to date very few published trials (across all cancers) have been able to access these types of patients and/or conduct such a comprehensive analysis of tissues.

Using a combination of molecular analyses, imaging and tissue micro-mechanics approaches, the authors present exciting new mechanistic insight showing that 12 weeks of daily UA reduces proliferation of breast epithelial cells and, more specifically, decreases the proportion, proliferation and ex vivo colony forming capacity of luminal progenitor cells in women at increased risk of breast cancer. High mammographic breast density is increasingly being used clinically as a biomarker of risk, but the mechanisms underlying the association are not well defined; the findings from this study provide new insight to help explain these links. MRI imaging showed reduced fibroglandular volume after UA, and this was consistent at a molecular and tissue level with extracellular remodelling and reduced collagen organisation and tissue stiffness, as shown by gene and protein expression changes, histology and atomic force microscopy. Most notably, inhibition of progesterone receptor signalling by UA led to downregulation of Collagen VI, which was spatially associated with LP cells, establishing a link between collagen organization and LP activity. The authors propose a model whereby progesterone paracrine signalling influences fibroblasts and basal cells within normal breast tissue, leading to remodelling of the extracellular matrix and increased stiffness; this in turn further activates progesterone receptor signalling in a positive feedback loop. Anti-progestins disrupt this mechanism, reducing collagen expression and stiffness, which results in decreased numbers of LP cells and since these are the purported cell of origin for triple negative breast cancer, this leads to reduced cancer risk.

Overall, the findings support the important concept of precision prevention whereby anti-progestin therapy could be specifically targeted to women with high mammographic breast density, as these individuals are likely to experience the greatest benefit.

Other strengths of the manuscript include the design and conduct of the BC-APPS1 trial; with its strong translational work assessing diverse endpoints, it represents an excellent template for the design of other early phase prevention trials, which are an essential first step in developing preventive therapies. Additionally, in generating new molecular insight on how breast density and stiffness increase breast cancer risk through progesterone receptor signalling, the study provides a strong mechanistic rationale for using measures of breast density as surrogate biomarkers in early phase prevention trials. In doing so, the work addresses one of the key barriers to the development of preventive therapies, namely a lack of surrogate biomarkers. The findings could therefore pave the way for many future trials in breast cancer prevention. Whilst the demonstration that modulation of the tissue microenvironment can

contribute to reduced cancer risk will provide the stimulus for more work in this area across other cancer types.

The manuscript is very well written, the results are clearly and appropriately presented. The interpretation of results and conclusions drawn are entirely appropriate. Relevant references seem to have been included to provide a balanced assessment of the field and impact of the findings.

Suggested improvements:

The BC-APPS1 trial with its strong translational work and diverse range of biomarkers represents an excellent template for the design of other early phase prevention trials. Although this point is made in the discussion, it should also be highlighted in the abstract/introduction.

Thank you for this suggestion, we have added the following to the final sentence of the abstract:

"This study offers a template for biologically informed early phase therapeutic cancer prevention trials and demonstrates the potential for premenopausal BC prevention with progesterone receptor antagonists through stromal remodelling and LP suppression." (Lines 53-56)

Results, line 126: please include a reference here for the method used to assess risk of breast cancer.

We have added "*Tyrer Cuzick v7.02*" (Line 131) to the appropriate results section and have added the weblink to the IBIS risk evaluator software in the methods (Line 735) as also requested by reviewer 1.

Results, line 130. Briefly describe the relevance of Ki67 as the primary endpoint of the trial and given an indication of its limitations to highlight the value of the new findings.

Thank you for this suggestion. We have added the following statement at the opening of the paragraph:

"The primary endpoint of the BC-APPS1 study was epithelial proliferation assessed by Ki67 immunohistochemistry, chosen primarily to power the study statistically, as Ki67 is not a recognised surrogate for BC risk." (Lines 135-137)

Results, line 142. The section on epithelial colony formation would benefit from additional explanation on the methods as it was not entirely clear whether each sample could form each type of colony on the same plate. Also, were the same number of cells plated for each sample? It is not clear from the images (Fig 1E) what constitutes a single colony; some additional annotation or inclusion of further images would be helpful.

We thank the reviewer for their helpful suggestions regarding the epithelial colony formation assay. We have revised the Methods section to clarify that the same number of cells were plated at two different densities (200 and 400 cells/cm²) for each sample, to account for potential differences in colony number or growth size. We also clarified that all three colony types (luminal, myoepithelial and mixed) can be observed on the same plate.

We have elaborated on the morphological criteria used to categorise colonies and clarified the definition of a single colony (≥ 50 cells). Specifically, luminal colonies appeared as tightly packed, cobblestone-like clusters with smooth, well-defined edges; myoepithelial colonies were composed of dispersed, teardrop-shaped, spindle-like cells with visible gaps between them; and mixed colonies

exhibited both cell types, with irregular, non-uniform edges. Each image shown represents a single colony, and to make this clearer, we have now added an additional panel showing three further examples of each colony type (Extended Data Figure 1E).

We did not observe any consistent differences in the overall number or size of colonies following UA treatment. However, quantitative analysis revealed a reduction in the proportion of mixed colonies after treatment, suggesting a selective loss of colony-forming capacity in bipotent cells.

To clarify this point, we have incorporated the changes highlighted below in bold in the Methods section “2D Human Mammary Colony-Forming Assay”:

*“Single epithelial cells were cultured in adherence in Human EpiCult-B media (Stem Cell Technologies) supplemented with 5% FBS (Gibco), 0.48 µg/mL hydrocortisone (Sigma) and 2 mM L-Glutamine (Gibco). **For each participant sample, cells were plated at two different seeding densities (200 and 400 cells/cm²) to account for potential differences in colony number or growth size.** Irradiated NIH 3T3 feeder cells (50,000 cells/mL; 50 Gy) were added to each plate. Three separate culture dishes per condition were plated and incubated at 37°C in 5% CO₂ for 10-12 days. Cells were fixed with acetone:methanol (1:1), air-dried, rinsed with distilled water, and stained with Giemsa (Sigma) for 2–3 minutes. **Colonies were defined as discrete clusters of ≥50 cells and classified according to established morphological criteria. All three colony types could be observed on the same plate: luminal colonies appeared as tightly packed, cobblestone-like clusters with smooth, well-defined edges; myoepithelial colonies consisted of dispersed, teardrop-shaped, spindle-like cells with visible gaps between them; and mixed colonies displayed features of both, with irregular, non-uniform edges³³.**”*(Lines 862-874)

Results, line 147: The authors state that ‘although the relative proportion of luminal colonies increased, the total fraction of LP cells decreased, suggesting lower absolute LP progenitor activity following treatment.’ However, isn’t the reduction in LP cells taken into account already by plating the same number of cells from pre/post treatment samples? I do not follow the argument based on the information given, unless there is an overall total reduction in the number of colonies. This section needs further explanation.

We agree that this sentence lacked clarity and may have led to confusion. It has been removed from the revised manuscript, and the paragraph has been rephrased to improve interpretation. Following anti-progestin treatment, the proportion of LP cells is reduced, as shown by FACS analysis, and this is accompanied by decreased LP activity, evidenced by reduced mammosphere formation efficiency. Together, these findings suggest that the reduction in mixed colonies in treated samples is likely due to diminished LP numbers/activity.

In the manuscript Results section “Preventive anti-progestin treatment reduces luminal progenitor activity”, the following sentence has been rephrased: “*Anti-progestin treatment reduced the proportion of mixed colonies from 70% (95% CI, 60% – 80%) to 55% (95% CI, 44% – 67%; p<0.05) - Figure 1E, Extended Data Figure 1E.*” (Lines 151-153)

Results, line 153: The authors use SOX9 as a marker of LP cells, but this is different to the markers used to designate the different populations in Figure 1D and initially show that the LP population was reduced after treatment. How do the two LP populations overlap, are all the SOX9+ cells within the CD49f+/EPCAM+ population?

Regarding the use of SOX9 as a marker for LP cells, data previously published by the first author (Simões BM) demonstrates that SOX9 protein predominantly marks FACS-sorted LP cells, specifically the CD49f+/EpCAM+ and ALDH+ populations, in the normal human breast (Domenici et al., *Oncogene*, 2019; PMID: 30622340). As shown by Western blot and immunofluorescence of FACS-sorted cells, the vast majority of SOX9+ cells fall within the CD49f+/EpCAM+ population, with a small proportion found in the LM (CD49f-/EpCAM+) population and even fewer in the BA (CD49f+/EpCAM-). Furthermore, in our scRNA-seq analysis SOX9 gene expression was predominantly seen in the LASP population, as shown in Figure 2C of our manuscript. This provides evidence of a clear overlap between SOX9+ cells and the LP/LASP populations.

In the manuscript Results section “Preventive anti-progestin treatment reduces luminal progenitor activity”, the following sentence cites the paper by Domenici et al. (reference 34): “SOX9 is a marker of LP cells²⁸ and...” (Line 157)

Results, line 157: The statement that the data demonstrate that treatment reduces the colony formation capacity of LP cells needs to be supported through clarification of the results, as indicated above.

As clarified above, the reduction in the proportion of mixed colonies is likely due to a decrease in LP cell numbers and/or activity. In addition, we observed a reduction in mammosphere formation efficiency, a 3D colony assay that reflects LP activity. Since both 2D and 3D colony assays are used to measure LP activity we have rephrased the sentence as follows: “Overall, these data demonstrate that anti-progestin treatment reduces the proportion, proliferation, and **activity** of LP cells in the normal breast tissue of women at increased BC risk.” (Lines 160-162)

Results, line 165: The manuscript states that ‘RNA quality was suboptimal in at least 1 sample from 14 participants and data are presented for the paired samples from 10 participants that met quality standards’. Please clarify if samples were prepared from all participants that completed the trial and why only 10 paired samples were eventually analysed; what is the justification for excluding the other samples/data excluded?

We initially performed RNA extraction for the 24 paired samples from participants who completed the trial. Of these, 4 pairs were excluded prior to sequencing due to low RNA integrity (RIN < 7). Following sequencing, an additional 10 pairs were excluded due to quality control metrics. Specifically it was identified that a number of the samples had a high percentage of reads mapped to intergenic regions of the genome. Additionally, strandedness analysis revealed a high percentage of reads mapped to the sense strand in these samples, which led us to conclude that these samples were likely contaminated with genomic DNA. While the STAR aligner would only have counted reads that mapped to genes, some of the reads may have mapped to contaminating DNA. On the advice of our Bioinformatics Facility, we applied an intergenic reads threshold of 10% and excluded the samples that exceeded this threshold from the analysis. This unfortunately resulted in the exclusion of 10 paired samples from further analysis.

In this respect, it would be helpful to explain in the methods how biopsy samples were prioritised for all the different types of analyses, given that not all analyses were conducted on all samples, and to understand the selection process for inclusion of patient samples for each type of analysis. The methods stated that 10 cores were obtained pre/post intervention but did all patients provide this many?

We agree it's important to clarify how biopsy samples were prioritised for the various analyses as all analyses were not performed on all samples. We obtained 10 core biopsies per participant both at baseline and post-treatment but not all samples yielded the same tissue composition/cell numbers due to natural heterogeneity in breast tissue (varying proportions of collagen, fat and epithelium). As described in the Methods, cores were fixed in formalin and paraffin-embedded, snap-frozen and the remaining were digested into single cell suspensions.

Analyses were prioritised based on tissue/cell availability and quality as follows:

From single-cell suspensions (live cells):

1. Mammosphere formation efficiency - data obtained from 19 paired samples.
2. 2D colony assays - performed on 18 paired samples.
3. Flow cytometry (FACS) - cells were frozen for FACS where possible; data available for 17 paired samples.
4. scRNAseq - only conducted on pairs with at least 300K spare cells both pre- and post-treatment; 6 paired samples met this condition.

From FFPE blocks:

1. Ki67 immunostaining – primary endpoint and undertaken on all 24 paired samples.
2. Tissue morphometry - performed on 19 pairs with ≥ 3 well-defined lobules (each with at least 10 acini).
3. PSR staining - conducted on 22 pairs with ≥ 3 well-defined lobules.
4. LCM proteomics - done on 4 pairs with high epithelial/lobular content; due to cost and complexity.
5. IMC (Hyperion) - used the same 4 pairs as LCM plus an additional 4 pairs with high epithelial/lobular content selected as a confirmatory cohort.

From snap-frozen tissue:

1. Bulk RNAseq - performed on 20 paired samples but only 10 paired samples passed RNA quality control.
2. AFM stiffness assays - carried out on 4 paired samples that showed $\geq 10\%$ reduction in PSR staining.

Due to space limitations in the manuscript, the information above, clarifying how sample inclusion was determined, was included in the "Life Sciences Study Design" section of the mandatory Nature Reporting Summary file accompanying the submission.

Results, line 185: Please include a key for Ext Figure 3A, or annotate the plots so it is clear where the LP cells are to support the quantitative data shown in Ext Figure 3C.

To clarify the data presented in Extended Figure 3 and support the quantification shown in Extended Data Figure 4A (formerly Extended Figure 3C), we have annotated the plots to indicate the location of the LP/LASP population. Additionally a key has been added to define all major epithelial and stromal cell populations more clearly.

Results, line 265: It would be worth stating that the persistence of SOX9 cells with Col1-VI/FN1 was in the post treatment samples, to aid clarity.

To clarify this point, we have incorporated the changes highlighted below in bold in the results section “Anti-progesterin treatment remodels the breast matrix”:

*“Single-cell neighbourhood analysis of SOX9^{high} and SOX9^{low} cells at baseline identified the SOX9^{high} cells to be in close proximity to regions of high Col-VI and FN1 but not Col-I expression **compared with SOX9^{low} cells, a finding that persisted following UA treatment** (Figure 3F-G and Extended Data Figure 11C).”* (Lines 303-307)

*“These data identify stromal remodelling as an early event in breast tissue perturbed by anti-progesterin treatment, although the **persistent spatial association of SOX9^{high} cells with Col-VI/FN1 post-treatment** suggests some continued co-localisation despite short term UA therapy (Extended Data Figure 11C).”* (Lines 312-315)

Results, line 278: comment on the in vivo relevance of the two collagen mimetic hydrogels, how does the stiffness relate to human tissue?

We acknowledge that reconciling in vitro measurements of matrix stiffness with in vivo human breast tissue values is challenging. To approximate physiologically relevant conditions, we used commercially available synthetic collagen-mimetic hydrogels at two stiffness levels: “soft” (1:4 dilution) and “stiff” (1:2 dilution). The respective moduli of the collagen mimetic hydrogels ‘soft’ (600-900 Pa, Young’s Modulus) and ‘stiff’ (1800-3000 Pa, Young’s Modulus) were calculated based on values reported by the manufacturer. These values, in turn, were determined by measurement of G' (shear modulus) with a dynamic rheometer, where non-diluted VitroGel equates to a stiffness 4,000 Pa (4kPa): <https://www.thewellbio.com/docs/what-is-the-elastic-modulus-of-the-high-concentration-different-dilution-vitrogels/>. Rheometers commonly measure mechanical properties over bulk/macroscopic length scales. [Redacted]

[Redacted]

[Redacted]

As a consequence, estimates of breast tissue stiffness vary considerably depending, in part, on the instrumentation used and the tissue region/area measured. For example, Patel et al (DOI: 10.1007/s10549-022-06607-2), using magnetic resonance imaging and elastography, report moduli of 2.5kPa and 2.0 kPa for breast parenchyma and fatty tissue respectively. However, using mechanical indentation (area not defined), Paszek et al (<https://doi.org/10.1016/j.ccr.2005.08.010>) report breast tissue stiffness ranging from ~200Pa for normal mammary gland to 4000Pa for tumour. In our 2016 (DOI: 10.1186/s13058-015-0664-2) study we used localized atomic force microscopy indentation of periductal regions to measure stiffness values of ~200-700 Pa. We are confident therefore, that our chosen hydrogel compositions will result in material stiffness which approximates that of healthy breast tissue and provide a biologically relevant framework for modelling the mechanical microenvironment of the breast.

Results, line 281: Explain why a different anti-progestin, other than UA was used for some experiments.

At the time of the initial experiments, UA was not available in our laboratory, and we therefore used the anti-progestin onapristone. However, we agree with the reviewer that testing with UA is important for consistency and clinical relevance. To address this, we performed a series of follow-up experiments using UA and confirmed the findings obtained with onapristone:

- We treated frozen baseline cells from 8 BC-APPS1 samples with UA in mammosphere colony-forming assays. While only 6 of the 8 samples formed mammospheres, we observed a consistent reduction in mammosphere-forming efficiency, replicating the effects previously seen with onapristone (Extended Data Figure 1G).
- We also cultured normal breast microstructures/organoids in soft versus stiff collagen-mimetic hydrogels and confirmed our prior findings: UA treatment reduced both MFE and the expression of progenitor cell markers Sox9 and C-KIT, consistent with results obtained using onapristone. These have now been included in results (Figure 4B–C and Extended Data Figure 12A).

These results further strengthen the conclusion that the attenuation of LP activity and PR-mediated responses in both soft and stiff matrix conditions are class effects of PRAs.

Results, line 285: The authors state that the ‘results indicate that anti-progestin treatment prevents stiffness-induced upregulation of progesterone signalling and progenitor cell activity’. However, there was also an effect on SOX9 protein levels in the soft conditions, so it doesn’t appear to be just a stiffness-induced upregulation; this needs to be commented on. What happens to the genes (TNFSF11 and C-KIT) at the protein level under these different conditions?

We thank the reviewer for this comment. We have since repeated these experiments using additional breast tissue samples cultured in soft versus stiff hydrogels, and in the presence of two different anti-progestins - onapristone and UA. In these analyses, we were able to assess SOX9 and also obtain reliable data for CKIT protein levels. However, TNFSF11 could not be detected by western blot as it is a secreted protein. Additionally, densitometry quantification normalized to beta-actin has now been included and is shown at the top of each western blot image.

We agree with the reviewer’s observation that the effects of anti-progestin treatment are evident not only under stiff conditions but also under soft conditions. Both SOX9 and CKIT protein levels were

reduced in the presence of either anti-progestin across both matrix stiffness conditions. These results suggest that there is basal PR activation within the 3D hydrogel culture system, even in softer matrices.

Consistent with our previous findings, increased stiffness led to upregulation of SOX9 and CKIT expression and this effect was attenuated by anti-progestin treatment. Together, these results support the conclusion that stiffness-induced enhancement of LP marker expression is mitigated by PR blockade, and also point to a basal level of PR activity in this in vitro system.

To clarify this point, we have revised the sentence in the Results section (“Anti-progestin treatment remodels the breast matrix”) as follows, with the changes highlighted in bold:

*“Overall, these results **establish** that anti-progestin treatment **attenuates** stiffness-induced up-regulation of progesterone signalling and progenitor cell activity, **and also reduces the basal level of PR activity seen in softer gels in this in vitro system.**”(Lines 331-334)*

Discussion, line 357: the authors suggest that reduced proliferation of LP cells due to treatment of anti-progestins may counteract the increased mutational burden resulting from recurrent progesterone stimulation of breast epithelium. If suitable samples were still available from the present study it would be interesting to test this hypothesis by comparing the somatic mutational burden in LP cells before and after treatment with UA. Even if a difference was not detected, this type of analysis could provide insight on the duration of intervention needed to have an impact on the mutation burden in these cells.

This is a very valid point. In a previous publication with Martin Widschwendter (Bartlett et al., 2022, Reference 54), we observed that both UA and mifepristone (another progesterone receptor antagonist) reduced DNA methylation signatures associated with LP cells and mitotic age. We also reported that TP53 mutation frequency decreased in breast tissue as a function of mifepristone exposure (where appropriate tissue samples were available), specifically in women who showed a reduction in the mitotic age index. Unfortunately, we have no remaining paired samples to test this hypothesis further in BC-APPS1 but such analyses would certainly be integrated into future studies.

Discussion, Section beginning line 368: The authors should comment on how universal the effects they observe are likely to be across other anti-progestins. Furthermore, the impact of UA associated liver toxicity should be discussed in terms of how this work should be progressed clinically.

Thank you for this suggestion, we have changed the final paragraph of the discussion to incorporate the suggested change in addition to that from reviewer 2 requested information on other hormonal responsive organs. Space does not permit a huge amount of detail but we have highlighted the issues that will need to be addressed in future studies.

*“In summary, our work identifies progesterone signalling as a key regulator of the BC-precursor LP/LASP cell population in the normal breast, and establishes a complex interplay between anti-progestin treatment, ECM remodelling, and LP/LASP cell dynamics. **Comparable in vitro effects with two independent anti-progestins, along with pre-existing clinical data on other PR antagonists^{9, 18}, suggest these may be class-effects of PR antagonism. Longer term studies are required to evaluate safety, particularly hepatotoxicity and effects on other hormone sensitive tissues such as the endometrium. BC is the most common cause of cancer death globally and such studies need to be progressed urgently. This should include further testing of the hypothesis that PR antagonism may reverse MD associated BC risk more broadly, beyond those at increased familial risk. MD reporting***

is now mandated in all US states by the FDA, albeit without any recommendation on methods to reduce it. Collectively, our study offers a roadmap for strategic design of molecularly informed primary prevention clinical trials.” (Lines 437-448)

Discussion: Please comment on the magnitude of effect with respect to Ki67 and MD as biomarkers for prevention trials; how do the effects observed compare with other trials where these markers have been used and correlated with a measure of clinical efficacy?

We have added to the opening paragraph of the discussion to include data on density reduction with tamoxifen in IBIS-1 (Cuzick et al, 2011, Reference 43) and the recent data from the KARISMA project that shows correlation with MD response and baseline density and Ki67 (Gabrielson et al, 2024, Reference 44). Space and reference number constraints mean it is not possible to go into detail about the inconsistent results of studies that have looked into tamoxifen response in the normal breast but we have included a statement to describe the sources of variability and importance of translational study design.

“In previous studies, BC risk reduction with the selective estrogen receptor modulator tamoxifen associated positively with MD reduction⁴³ and MD reduction associated positively with baseline MD and Ki67 expression⁴⁴. However, studies evaluating change in normal epithelial Ki67 with tamoxifen have provided inconsistent results, likely due to failure to control for menstrual cycle phase⁸ and the combination of pre and postmenopausal populations, highlighting the importance of study design in translational prevention research.” (Lines 397-402)

Discussion: Did the authors see any evidence of differential effects between participants in their response to UA across the various types of analyses conducted? This would be worth a comment.

We examined several baseline characteristics - age, breast cancer risk, parity and BMI - to assess whether there was any evidence of differential effects among participants in response to UA. Specifically, we evaluated treatment response in terms of fold-change reduction in cell proliferation (Ki67) and fold-changes in the proportion or activity of LP/LASP cells (% of LP cells by FACS, % of Sox9⁺ LP cells by immunofluorescence, and mammosphere formation efficiency). Across these analyses, we did not observe any significant correlation between baseline characteristics and response to UA. However, it is important to note that our dataset includes only 24 women which may limit our ability to detect associations.

As discussed in response to reviewer #2 we looked for correlations between the above responses and %VBD as a continuous variable and saw no significant associations. However, when categorising high vs low MD (BIRADS A/B vs C/D) we saw significant reductions in Ki67, MFE and LP fraction by FACS, only in the high MD cohort. The Ki67 data has been added to the manuscript as detailed in response to reviewer 2.

In addition to the changes made in response to Reviewer 2, we have added the following sentence to the Discussion:

“The response to UA did not show any correlation with age, breast cancer risk, parity or BMI.” (Lines 433-434)

Methods, BC-APPSI study: Please include a reference/website for the version of the Tyrer-Cuzick risk estimation programme used.

This was Version 7.02 which has been included in the Methods with a weblink.

Methods, Ki67 staining: Please clarify whether the slides were scored automatically using software and the pathologist just verified the results, or whether the pathologist's scores correspond to the data presented in this manuscript. If the latter, were the slides (for both Ki67 and analysis of acinar/lobular ratio) scored blind?

Regarding the original Ki67 staining presented in the manuscript, the slides were scored by a pathologist. To validate these findings, a researcher blinded to participant number and timepoint analysed Ki67 using HALO automated analysis (please see results tabulated below). This analysis confirmed the overall results, with 22 out of 24 samples showing agreement in the direction of change (i.e. fold change after treatment) between the pathologist's assessment and the HALO analysis; only samples 21 and 22 showed slight differences.

HALO QUANTIFICATION				PATHOLOGIST QUANTIFICATION			
	Pre-treatment	Post-treatment	Fold-change		Pre-treatment	Post-treatment	Fold-change
BAP01	0.40	0.14	0.35	BAP01	2.30	0.83	0.36
BAP02	1.17	0.24	0.20	BAP02	4.65	1.45	0.31
BAP03	2.57	0.08	0.03	BAP03	8.15	1.30	0.16
BAP05	0.95	2.38	2.51	BAP05	1.95	2.90	1.49
BAP06	5.17	0.84	0.16	BAP06	9.90	1.55	0.16
BAP07	2.25	0.55	0.24	BAP07	4.60	1.15	0.25
BAP10	2.22	0.73	0.33	BAP10	4.50	2.75	0.61
BAP11	7.65	1.36	0.18	BAP11	16.00	3.25	0.20
BAP12	5.30	0.55	0.10	BAP12	17.60	2.35	0.13
BAP13	1.33	0.28	0.21	BAP13	4.75	2.05	0.43
BAP14	12.12	0.50	0.04	BAP14	34.00	2.35	0.07
BAP16	1.13	0.38	0.33	BAP16	5.40	2.05	0.38
BAP17	1.81	0.37	0.20	BAP17	13.50	1.35	0.10
BAP18	5.88	0.58	0.10	BAP18	10.59	2.43	0.23
BAP19	2.95	1.18	0.40	BAP19	4.97	3.19	0.64
BAP21	1.59	1.19	0.75	BAP21	3.25	4.56	1.41
BAP22	1.55	1.76	1.14	BAP22	4.35	3.39	0.78
BAP24	0.78	0.36	0.46	BAP24	4.81	2.39	0.50
BAP25	1.06	0.83	0.78	BAP25	6.57	2.57	0.39
BAP26	3.44	2.36	0.69	BAP26	8.09	5.31	0.66
BAP28	0.85	1.69	1.98	BAP28	4.42	10.16	2.30
BAP30	0.96	0.44	0.46	BAP30	3.51	1.65	0.47
BAP31	12.71	1.72	0.14	BAP31	15.73	5.96	0.38
BAP32	1.11	0.93	0.83	BAP32	3.57	2.57	0.72

Table: Ki67 staining scores assessed by a blinded researcher using HALO software and by a pathologist. Fold changes after treatment confirm consistency between manual and automated quantification.

Whilst there were some discrepancies in the absolute percentages, with HALO identifying a lower number of Ki67+ cells, this is likely due to differences in the thresholds used to define Ki67 positivity. Pearson correlation analysis (R = 0.86, see graph below) indicated a strong positive correlation and Cohen's Kappa score (0.62) confirmed substantial agreement between the two approaches, providing confidence in the robustness of the results reported.

Regarding the acinar/lobular ratio analysis, it was performed entirely blind to treatment allocation.

Figure: Correlation plot comparing Ki67 fold changes after treatment as assessed by a pathologist and by HALO software.

To address these issues in the manuscript, we have made the changes below:

- added the following sentence to the “Ki67 staining” section of Methods: *“Ki67 quantification was independently performed by a researcher fully blinded to participant number and timepoint using the HALO software, with Cohen’s Kappa score (0.62) confirming substantial agreement between the two assessments.”* (Lines 822-825)
- added the following sentence to the “Tissue morphometry” section of Methods: *“This analysis was performed blind to participant number and timepoint.”* (Line 833)

Referee #4 (Remarks to the Author):

This manuscript explores how anti-progestin therapy (ulipristal acetate, UA) influences breast cancer formation. Leveraging longitudinal samples from the BC-APPS1 clinical trial, the authors apply multiple modalities – including MRI, transcriptomics, proteomics, histology and microscopy – to study the impact of UA treatment. These data suggest a reduction in the number of the cancer-precursor “luminal progenitor” (LP) cells and altered stroma-epithelium interactions. With scRNAseq, the authors further examine shifts in cell-cell signaling, identifying alterations in ECM-related pathways. By performing spatially resolved proteomic analyses (LCM and imaging mass cytometry), they corroborate these observations. By performing functional assays with organoids and AFM on tissue samples they further implicate tissue stiffness as a target of UA associated with breast cancer phenotypic features.

Overall, this is a unique and interesting study examining potential mechanisms by which UA can prevent breast cancer formation in premenopausal women at high risk. However, there are numerous issues with the way in which the work is described (e.g., the manuscript cannot be read without jumping back and forth to between the main text and methods; it is often unclear what is being plotted without detailed investigative work by the reader) and the data analyses were performed (which may impact interpretation). While the study and data appear interesting and valuable, there is substantial work that needs to be done which could influence the findings. Until these points are resolved, it is hard for us to advise on topic such as impact.

Comments by Figure

Figure 1 and Extended Data Figure 1:

1. The data in Figures 1 and S1 are tied. Please fix your statistics to consider this. This is true for several other analyses as well.

Thank you for this comment which we have interpreted to mean that some participants have equal (tied) values in certain analyses. P-values throughout the manuscript were generated using the `stat_compare_means` function from the `ggpubr` package (v0.6.0), applying the `wilcox.test` method. This function automatically selects the appropriate test depending on the data structure: a Wilcoxon signed-rank test for paired samples or a Wilcoxon rank-sum test for independent samples. All baseline and post-treatment sample comparisons were performed using the Wilcoxon signed-rank test for paired samples. When tied values are present in the data, the function calculates approximate p-values and applies a correction for ties. Therefore, the statistical tests already account for the presence of ties in the datasets shown in Figures 1, S1 and elsewhere in the manuscript.

We have added the following text to the “Statistical analyses” section of Methods: *“If not stated otherwise, p-values were generated using the “stat_compare_means” function from the “ggpubr” package (v0.6.0), applying the wilcox.test method. This performs either a Wilcoxon signed-rank test for paired samples or a Wilcoxon rank-sum test for independent samples. Exact p-values are reported when there are no tied values. When ties are present, an approximate p-value is calculated with a correction for ties.”* (Lines 1164-1168)

2. Figure 1E - how were colonies classified?

This point was also raised by Reviewer 3 and has been addressed in our response above (see pages 17-18).

Figure 2 and Extended Data Figures 2 & 3

1. Figure 2 - The annotation for the major cell populations should be more accurate and precise, especially for the cell types defined as perivascular and immune (Extended data Figure 2). For the clusters defined as immune, there are clearly 3-4 different clusters in the UMAP and finer annotation for these clusters would be useful for downstream analysis. Also, the reference signature the authors use is from breast milk. Please use a reference dataset that comes from similar or comparable sample collection (e.g., Murrow et. al., 2022). In addition, by directly running differential gene analysis for each cluster can help identify the marker genes for each cluster.

We thank the reviewer for this helpful suggestion. In our revised analysis, we have incorporated label mapping from the integrated Human Breast Cell Atlas (iHBCA; Reed et al., 2024, Reference 35) into our BC-APPS1 single-cell RNA-seq dataset. This has enhanced the granularity of our cell type annotations and better situates our findings within current gold-standard cellular nomenclature in the field. We agree that the iHBCA provides a more appropriate and relevant reference signature than datasets derived from breast milk, and we have updated our reference.

In Extended Data Figure 5, we have improved the annotation of both perivascular and immune cell clusters. Specifically, we now identify distinct subclusters within the perivascular population (PV1, PV2, PV3/4/5) and within the immune compartment (B cells, T cells, myeloid cells, plasma cells and mast cells).

To support these annotations, we performed differential gene expression analysis to identify cluster-specific marker genes, visualised using dot plots in Figure 2C (main clusters) and Extended Data Figure 6 (subclusters). While our primary analyses focus on broader cell populations, where robust gene expression changes were observed in response to UA treatment, we now complement this with subcluster-level analysis (guided by iHBCA annotations) to better understand which specific cell states may be driving our identified phenotypes.

2. Moreover, the characterization of the data is underdeveloped. There is no discussion of changes in cell abundance or activity, just a rapid transition to a receptor-ligand analysis. E.g., normalization for cell numbers is mentioned at line 190 but those numbers are never shown making it hard to understand how significant the shifts in overall signaling really are.

We agree with the reviewer's observation and have substantially deepened our analysis to improve the clarity and characterisation of the dataset. We now include cell abundance proportionality analysis across both broad cell populations and iHBCA-mapped subclusters. This revealed no significant alterations in proportionality across most populations (Figure 2D). However, when restricting the analysis to the epithelial compartment, we observed a decrease in the proportion of LASP cells post-UA treatment, consistent with our flow cytometry findings (Figure 2D). Further analysis of iHBCA-mapped subclusters showed no individual LASP subpopulation with a significant change in proportion post-treatment (Extended Data Figure 4D). For all other clusters and subclusters (Extended Data Figure 4C), no notable shifts in abundance were detected, suggesting that UA primarily influences gene expression programmes, rather than directly depleting specific cell states.

To better characterise transcriptional changes, we performed pairwise differential expression testing for all broad cell populations using the Memento framework, which identified striking downregulation of genes, particularly within LHS cells. These results are summarised in Figure 2E and Extended Data Figure 7A (UpSet plots). Supplementary Table 1 includes the list of all differentially expressed genes in each population.

To avoid biases toward the most populous cell populations as key drivers of cell signalling in our CellChat analyses, we performed cell number normalisation. We have now clarified in the Methods section “Cell:cell communication analyses”, that we used a down-sampling approach, analysing 985 cells per broad cluster and 300 per iHBCA-mapped subcluster, to assess ‘per cell’ signalling contributions.

3. Figure 2C - Please fix your legend

Thank you for spotting this. It has been fixed, now Figure 2G.

4. Lists of DE genes are missing to enable interpretation of the enrichments (only LRs are provided).

A full list of pairwise DE analysis across the 7 main cell types is presented in Supplementary Table 1. These lists were then subsetted by pvalue (de_pval) or logFC (de_coef) for downstream analyses including ligands, targets and pathway analyses.

5. Links between 2D & F and 2E & G are missing.

Thank you for highlighting this lack of clarity. We have amended the flow and now present pathway analysis during the bird’s eye analyses of UA-effects in the 7 main cell types identified. This appears in Figure 2F and Extended Data Figure 8A.

6. Line 214 - Why did you restrict the target genes to the Reactome ECM gene set? This is not justified and trivializes the results in lines 212 - 233. Also, the comment in line 227 - 229 comes out of nowhere.

As part of the process to improve clarity and flow we now present the data as an initial confirmation that the greatest impact on individual gene expression changes was seen in LHS cells (new Figure 2E). However, the majority of the gene expression changes in LHS were related to cell intrinsic processes. Using cell-chat to explore cell-cell communication networks, we identified that the most profound changes in outgoing pathways (a surrogate for paracrine signalling) were seen in BMYO and FBs where extracellular matrix organisation, and in particular collagen signalling, emerged as the major pathways showing consistent and significant alterations following UA treatment (now presented in Figure 2H). Proteomic and bulk RNA-seq data analyses also supported these findings. We have modified the text and figures extensively which now lead more robustly to the focus on ECM regulation, given the strong and consistent collagen suppression observed across all data modalities post-UA treatment.

The original comment (lines 227–229) has also been removed to improve the clarity and cohesion of this section.

7. Extended Data Table 3 - Please be quantitative and use proper statistics.

A full list of pairwise DE analysis across the 7 main cell types is presented in Supplementary Table 1. This included p-values for both fold change and variance (de_pval, dv_pval), logFC (de_coef), and variance (dv_coef). We have also included a table (Supplementary Table 2) of these results subsetted to ECM genes through intersection with the REACTOME Extracellular Matrix Organization geneset.

8. Figure 2H - For the ligand-receptor interaction of collagen, are the receptors also down-regulated after UA treatment? This panel is hard to interpretate. Please show single-cell expression data to demonstrate that collagen is downregulated post-UA.

We agree that the binary presentation of gene expression in the original manuscript limited interpretation and the ability to determine the magnitude of effects. To address this, we now present dot plots throughout, where dot size reflects the proportion of cells expressing a gene within a cluster, and dot colour indicates mean expression level. For key genes of interest (including COL6A3 and WNT5A), we show per-participant gene expression across all 6 participants, with accompanying summary plots showing the matched baseline/posttreatment pairs, logFC and pvalue. These revisions are shown in main Figure 2J and Extended Data Figure 10B/C/D.

As receptor expression is not typically co-regulated with its ligand, and downregulation is not necessarily required to observe downstream effects, we focused our analysis on differential expression of ligands and downstream target genes, applying a threshold whereby the associated receptor was expressed in at least 3% of the target population. This liberal receptor cutoff was selected to retain potentially meaningful interactions, though in most cases, the target receptor was expressed in around 10% across target cells - Extended Data Figure 10E. The dot plot on Extended Data Figure 10E shows the expression of WNT5A-associated receptors within FB1 and FB3 subclusters.

9. Figure 2I - For the NicheNet analysis, the ligand expression is presented in binary. Please show a dotplot to give a sense for the percentage of cells expressing the ligands/target genes + average expression.

As noted above, we now present dotplots for all expression data where size indicates the fraction of a cluster expressing a gene, with colour indicating mean expression.

10. Extended Fig 2A - Something seems off here. No genes look clearly DE. Please double check your analysis. Also, what is the scale? Please make sure these are always added (e.g., Extended Fig 2C). And, how were the enrichments determined. The analytical steps are opaque.

We thank the reviewer for this helpful comment. We acknowledge that the initial heatmap representation in Extended Fig. 2A lacked clarity, which may have made it difficult to visually identify differentially expressed genes.

Without changing the underlying analysis, we have updated the visual presentation of the heatmap to improve interpretability:

- Initially, genes were grouped by Gene Ontology (GO) category without any additional ranking. In the revised version, we have applied default hierarchical clustering parameters, which has resulted in a clearer structure, with up-regulated genes post-treatment now appearing at the top of the heatmap and down-regulated genes at the bottom.
- We have changed the plotting package from “pheatmap” to “ComplexHeatmap”, which provides improved colour contrast and sharper visualisation.
- The colour scale is now clearly indicated, addressing the concern about scale labelling.

To further clarify the data shown:

- Differential expression analysis following UA treatment was performed using DESeq2.
- The heatmap displays genes differentially expressed between treatment groups, with an absolute log₂ fold change ≥ 1.5 and an adjusted p-value ≤ 0.05 .
- DESeq2 variance-stabilising transformation (VST) values were used, and rows were scaled so that the heatmap colour range spans from -2 to 2.
- Gene Ontology terms were downloaded from Ensembl BioMart (<https://www.ensembl.org/biomart/martview/>) using Ensembl Genes 103 and Human Genes GRCh38.p13, selecting the attributes "Gene stable ID", "Gene name", and "GO term name".
- Genes were annotated by functional category as follows:
 - "Cell cycle" if any GO term contained the phrase "cell cycle";
 - "Extracellular matrix" if any GO term contained "extracellular matrix";
 - "Other" if the gene did not fall into either of the above categories.

We have added further details to the "Tissue bulk RNAseq analysis" section of Methods to clarify the data analysis. (Lines 970-973)

11. Extended Data Table 1 and Extended Data Figure 3: is "HR" the same as "LM"?

Yes, HR and LM refer to the same cell population. We have now updated the label to "LHS" throughout to maintain consistency with the nomenclature used in the iHBCA paper (Reed et al., 2024, Reference 35).

12. Extended Data Figure 3D - The LP population shows substantial expression heterogeneity with very few cells expressing the selected genes. What are the cell numbers here? What are the expression levels? This panel is not very convincing. Please also show per patient values pre- and post.

We agree with the reviewer that UMAP plots can be misleading in conveying both the proportion of cells expressing a gene and the magnitude of expression within a cell population. To address this, we now present dot plots across all expression analyses.

For key genes such as TNFSF11 (RANKL) and CXCL13, we present both per-participant expression levels and summary mean expression data, as shown in Extended Data Figure 7B. These genes are well-established progesterone response markers and serve as useful indicators of UA activity. As described in the revised Results section "Transcriptome network analyses reveal luminal mature cells orchestrate the matrisome landscape of basal cells and fibroblasts", both TNFSF11 and CXCL13 are often difficult to detect in single-cell datasets, including the iHBCA dataset. Nevertheless, even though they are detected at low levels in terms of transcripts expressed/cell, per-participant pairwise mapping revealed a consistent and robust reduction in expression post-UA treatment, an effect that aligns with the anticipated pharmacological action of UA.

13. Extended data figure 3F - the legend title is probably wrong, as q values should not be higher than 1.

In the SCPA package that was initially used for pathway analysis, the SCPA authors implemented their own metric, called a "q value", which is not synonymous to conventional q values typically used for

assessing false discovery rate. Given the confusion related to this non-standard approach and that standard gene set enrichment analyses identified similar Reactome pathways, we decided to change our analysis to gProfiler-based pathway analysis, where top significant pathways are presented according to the intersection size (i.e. the number of overlapping genes) within a Reactome geneset. Full details are provided in the Single Cell Pathway Analysis section of the Methods.

14. We understand why you focus on FB and BA cells, but a more comprehensive analysis is needed and warranted.

We agree that a more comprehensive analysis is important. We believe that our detailed engagement with the points raised above have greatly improved the rigour, clarity, and comprehensiveness of analyses on the single-cell RNA seq data. In particular, we have expanded our investigation beyond the fibroblast and basal populations to include proportionality and differential expression analyses across all major and subclustered cell types, guided by integration with the iHBCA dataset. These additions provide a more complete characterisation of the cellular and transcriptional effects of UA treatment.

Figure 3 and Extended Data Figure 4

1. Line 241-242. Unexpected why? Was the direction of change consistent?

Lines 241-242 reads “Among these 1,519 proteins, 1,454 (96%) were consistently detected both before and after treatment (data not shown), and 1,373 (90%) were identified in all four participants (Figure 3B). This extensive overlap underscores the reliability of the data. We identified 65 proteins regulated by UA treatment with q-value <0.05 (Figure 3C).”

We are unclear as to what the referee is addressing with the comment “unexpected why?”, however in relation to the consistency of the direction of change we have generated a heatmap illustrating the direction of change for each significant protein in individual participants [Extended Data Figure 11A]. We note that all ECM components that were identified as significantly altered in response to UA treatment exhibit a consistent downregulation across all participants, also addressing the referee’s comment concerning lines 249-252 [Figure 3C&3E].

The text in lines 241–242 (lines 285–286 in the updated manuscript) has been slightly simplified to improve clarity, as follows: “Among these 1,519 proteins, 1,454 (96%) were consistently detected before and after treatment (data not shown) with 1,373 (90%) identified in all four participants, confirming data reliability (Figure 3B). We identified 65 proteins regulated by UA treatment with q-value <0.05 (Figure 3C).”

2. Figure 3C - Why was patient treated as a random effect rather than explicitly considered when it was modeled in the bulk RNA work?

Data from the LCM-proteomic experiment were analysed using MSqRob, a peptide-level robust ridge regression method for analysing relative protein quantification (DOI: 10.1074/mcp.M115.055897; DOI: 10.1016/j.jprot.2017.04.004; DOI: 10.1021/acs.analchem.9b04375; DOI: 10.1074/mcp.RA119.001624). In the peer-reviewed tutorial for MSqRob, the authors treated biological replicates as random effects, stating “The effect of each single biological repeat will differ each time one would repeat the experiment” [DOI: 10.1016/j.jprot.2017.04.004]. We acknowledge that the distinction between fixed or random effects is ambiguous, and in the context of this study, the

participant (biological replicate) could also be considered a fixed effect. To assess the impact of classifying “participant” as a fixed effect, we reanalysed the data in MSqRob, defining the participant as a fixed effect (see figure below).

Figure: Volcano plots showing significantly differentially abundant proteins after UA treatment comparing results from random effect analysis (left) versus fixed effect analysis (right).

Notably, the proteins central to our key findings, the Collagen VI chains (COL6A3 and COL6A6) remained among the most significantly downregulated in response to UA treatment, regardless of whether "participant" was modelled as a fixed or random effect. Additional Collagen VI chains were identified as significantly downregulated only under one modelling approach: COL6A2 in the random effect analysis and COL6A1 in the fixed effect analysis. COL1A1 also remained significantly downregulated across models, while the q-value for COL1A2 increased to 0.058 and FN1 was not identified as significantly downregulated in the fixed effect analysis. Despite these minor differences, the overall pattern of differentially abundant proteins, both upregulated and downregulated, was highly consistent between models, with comparable q-values (all <0.05) and fold changes (effect sizes). This indicates that modelling "participant" as either a random or fixed effect does not alter the interpretation of our core finding: Collagen VI and other matrisome proteins (primarily collagens) are significantly downregulated in response to UA treatment. This conclusion is further supported by orthogonal validation methods. Imaging Mass Cytometry confirmed the downregulation of COL1, COL6 and FN1 proteins (Extended Data Figure 11B), while scRNA-seq data confirmed the downregulation of COL1A1, COL1A2, COL6A1, COL6A2, COL6A3, and FN1 in fibroblasts (data available in Supplementary Table 2).

3. Figure 3D - Could use a more comprehensive analysis.

Data from the LCM-proteomics experiment was analysed using Gene Set Enrichment Analysis (GSEA) with Reactome pathway annotations to provide a high-level overview of biological pathway alterations in response to UA treatment. Subsequent figure panels delve into extracellular matrix changes, which

are the key findings of our study, in greater detail. Figure 3D focuses on pathway-level changes that show a q-value of <0.05 , as displaying all pathway alterations would require an impractical amount of space.

We believe that our original approach provides a broad overview of the biological changes induced by UA treatment at the proteome level, while maintaining narrative focus on the most impactful findings of our study. We present below an expanded gene set enrichment analysis (GSEA) of the dataset, including Gene Ontology (GO) Biological Process terms, GO Molecular Function terms, and KEGG pathways. These additional analyses further support that extracellular matrix (ECM) organization is among the most significantly affected processes after UA treatment.

We also present below a visualization of the Reactome pathway enrichment analysis - independent of q-value thresholds - as a comprehensive examination of the LCM-proteomics data. ECM organization emerges as the most affected process, with downregulation of key ECM components (indicated in blue). A zoomed-in view of the ECM-related pathways is shown in the final panel.

Figure: Gene set enrichment analysis (GSEA) after UA treatment showing GO Biological Process, GO Molecular Function, and KEGG pathways.

Figure: Overview of Reactome pathway enrichment analysis highlighting alterations in response to UA treatment. A zoomed-in view of down-regulated ECM-related pathways is shown at the bottom.

4. Extended Data Figure 4A - This would make much more sense shown as a volcano plot, especially to examine consistency for lines 249-252.

The data presented in Extended Data Figure 11A (former 4A) is a heatmap depicting the 65 proteins found to be significantly differentially abundant following UA treatment. These data are displayed as a volcano plot in Figure 3C. However, in response to the referee’s comment regarding the examination of consistency, we have: revised Extended Data Figure 11A to include a heatmap illustrating the abundance changes of the 65 significant proteins for individual participants; recoloured 3C (volcano plot) to allow the reader to identify the direction of change for significant ‘Matrisome’ proteins; and

updated Figure 3E to include the fold change in abundance of the significantly down-regulated 'Matrisome' proteins for individual participants. We believe these revisions provide the reader with a comprehensive view of the consistency within our dataset.

5. Extended Data Figure 4B - Markers selected how?

Markers used for Imaging Mass Cytometry (IMC) are detailed in the figure legend of Extended Data Figure 11B (former 4B). E-Cadherin (E-Cad, luminal epithelial cell marker), α -smooth muscle actin (SMA, basal/myoepithelial cell marker) and Collagen I (Col-I, stromal marker) are shown. E-Cadherin and SMA are commonly used in immunostaining to distinguish luminal and myoepithelial cells in breast tissue, as demonstrated in studies such as Jackson et al., 2020 (Nature, DOI: 10.1038/s41586-019-1876-x). Collagen I is a well-established component of the breast stroma, as reported by Bodelon et al., 2024 (Breast Cancer Research, DOI: 10.1186/s13058-021-01482-z).

Col-I, Col-VI, and FN1 were selected based on the LCM-proteomics results, where they represent 3 of the 27 significantly downregulated matrisome proteins. Moreover, COL1A1/A2, COL6A1/A2/A3 and FN1 gene expression was also found to be significantly downregulated in fibroblasts (scRNA-seq data available in Supplementary Table 2). Selection of IMC markers also considered antibody availability.

6. Figure 3F and Extended Data Figure 4C - Please do not use green-red.

We have altered the colours used in Figure 3F and Extended Data Figure 11C (former 4C) to avoid green-red.

7. Line 259 - How were cells classified as high vs low? Also, what fraction are Ki67+?

For each image, a single analyst, blinded to slide identity and referencing a common reference image, selected a pixel intensity threshold to captured cells exhibiting comparable staining intensity and cellular distribution for E-Cadherin and Sox9. Thresholding was performed independently for each channel. Cells with Sox9 and E-cadherin intensity above the threshold were categorized as "Sox9^{high}" and "Ecad+" respectively. As E-Cadherin staining produced high signal-to-noise ratio, the pixel intensity threshold remained consistent across all images. All Ecad+ cells in each image were ordered by Sox9 intensity, and the number of Sox9^{high} cells was determined (n). The top and bottom n cells were classified as "Sox9^{high}" and "Sox9^{low}", respectively. To ensure our method was unbiased, a second analyst independently repeated the process. Results were consistent, showing no significant differences between analyst.

To further assess the robustness of our analysis method and to ensure manual thresholding did not introduce analytical bias, we repeated our analysis using a fixed percentage threshold per image (see figure below). Specifically, Ecad+ cells were ranked by Sox9 staining intensity, and the top and bottom 20% of cells were classified as "Sox9^{high}" and "Sox9^{low}", respectively. Comparison of the results between manual (top graphs) and fixed percentage (bottom graphs) thresholding revealed no significant differences.

Figure: Comparison of manual versus fixed percentage thresholding (top and bottom graphs, respectively) for Sox9 classification.

Analysis of Ki67 staining in the IMC data revealed a significant reduction in Ki67+ in the E-cad+ cell populations following UA treatment (see figure below), consistent with our IHC results in Figure 1B.

T1/T2 Ki67% in epithelial region

Figure: Percentage of Ki67+ cells in the IMC data at baseline (T1) and post-treatment (T2).

In both baseline and post-treatment conditions, Ki67+ cells were significantly more prevalent in Sox9^{high} populations compared to Sox9^{low} populations [Extended Data Figure 11D], confirming that Sox9^{high} cells exhibit higher proliferative activity, as anticipated. After UA treatment, a reduction in the frequency of proliferative (Ki67+) cells was observed in both Sox9^{high} and Sox9^{low} populations [Extended Data Figure 11D]. Notably, participants with the highest percentage of Ki67+ Sox9^{high} cells at T1 (e.g. >6%) exhibited a marked reduction in this percentage at T2.

We have included the following statement in the results section “Anti-progestin treatment remodels the breast matrix”:

“In both baseline and post-treatment conditions, Ki67⁺ cells were significantly more prevalent in the SOX9^{high} compared to SOX9^{low} populations, confirming their higher proliferative activity, although UA treatment reduced proliferation in both populations (Extended Data Figure 11D).”(Lines 307-310)

8. The strong spatial association between the Collagens and SOX9hi LP cells is interesting. Does colocalization shift as a result of treatment? We’d like to better understand how treatment folds into the results shown in Figure 4G and Extended Data Figure 4C&D.

Figure 4H (former 4G) explores differentially expressed genes between high MD and low MD via whole tissue gene expression analysis (RNA-seq).

Figure 3G & Extended data figure 11C (former 4C&D) examine the spatial relationship between Sox9 expression in epithelial cells and the abundance of extracellular Collagen VI, Fibronectin and Collagen I in baseline (T1) and post-treatment samples (T2). The data presented in these figures demonstrate that, in both baseline and post-treatment conditions, Sox9^{high} cells are spatially associated with regions of higher Collagen VI and Fibronectin abundance compared to Sox9^{low} cells. In contrast, the association with Collagen I is reversed, with Sox9^{low} cells showing a stronger spatial association with Collagen I. In Extended Data Figure 11B we show a decrease in the abundance of Collagen VI, Collagen I, and Fibronectin in the epithelia and peri-epithelial stroma following UA treatment. This reduction in ECM abundance is also evident when comparing the proximal regions of Sox9^{high} and Sox9^{low} cells at T1 and T2 [Figure 3G, Extended Data Figure 11 C].

Together, these data indicate that while the colocalization of Sox9^{high} cells with regions of high Collagen VI and Fibronectin abundance does not change in response to UA treatment, the overall abundance of the ECM proteins decreases globally across both the epithelia and peri-epithelial stroma. Thus, both Sox9^{high} and Sox9^{low} cells experience a similar decrease in ECM protein abundance in their proximal microenvironment.

To clarify this point for the reader, we have reformatted plots of local ECM intensity for IMC experiments in Figure 3G and Extended Data Figure 11C into box and whisker plots.

We have also incorporated the changes highlighted below in bold in the results section “Anti-progestin treatment remodels the breast matrix”:

*“Single-cell neighbourhood analysis of SOX9^{high} and SOX9^{low} cells at baseline identified the SOX9^{high} cells to be in close proximity to regions of high Col-VI and FN1 but not Col-I expression **compared with SOX9^{low} cells, a finding that persisted following UA treatment** (Figure 3F-G and Extended Data Figure 11C).”* (Lines 303-307)

*“These data identify stromal remodelling as an early event in breast tissue perturbed by anti-progestin treatment, although the **persistent spatial association of SOX9^{high} cells with Col-VI/FN1 post-treatment** suggests some continued co-localisation despite short term UA therapy (Extended Data Figure 11C).”* (Lines 312-315)

Figure 4 and Extended Data Figure 5

Figure 4A&B - How was expression measured? There are several details required to understand your analyses that are missing. Also, please do a tied analysis (Ditto for 4D & 4F).

We agree that more detail was needed and have now updated the Figure 4A legend to clarify that gene expression results were obtained by real-time PCR. The methodology used for these analyses is detailed in the "VitroGel Assay" and "Real-time PCR" sections of the Methods. Regarding the statistical analysis, we have detailed above how the statistical test accounts for the presence of ties in the datasets throughout the manuscript.

2. Figure 4C - Please quantify and show replicates.

We have now included replicates for an additional 2 breast samples that were treated in vitro with Onapristone under soft versus stiff conditions (Figure 4B, Extended Data Figure 12B). We have also repeated these experiments using UA in 3 normal breast samples (Figure 4B, Extended Data Figure 12A). These new data show the same trends; an increase in SOX9 and CKIT expression when cells are grown in stiff conditions and a decrease in both proteins in the presence of anti-progestin treatment. Additionally, densitometry quantification normalized to beta-actin has now been included and is shown at the top of each western blot image.

3. Lines 284-286 - There are logic gaps here.

We agree with the reviewer's comment. The effects of anti-progestin treatment are evident not only under stiff conditions but also under soft conditions. Both SOX9 and CKIT protein levels were reduced in the presence of either anti-progestin across both matrix stiffness conditions. These results suggest that there is basal PR activation within the 3D hydrogel culture system, even in softer matrices. Consistent with our previous findings, increased stiffness led to upregulation of SOX9 and CKIT expression and this effect was attenuated by anti-progestin treatment. Together, these results support the conclusion that stiffness-induced enhancement of LP marker expression is mitigated by PR blockade, and also point to a basal level of PR activity in this in vitro system.

To clarify this point, we have revised the sentence in the Results section ("Anti-progestin treatment remodels the breast matrix") as follows, with the changes highlighted in bold:

*"Overall, these results **establish** that anti-progestin treatment **attenuates** stiffness-induced up-regulation of progesterone signalling and progenitor cell activity, **and also reduces the basal level of PR activity seen in softer gels in this in vitro system.**"(Lines 331-334)*

4. Figure 4G & Extended Data Figure 5 - We would suggest replacing this with standard methods for finding biomarkers (ROC), filtering for expression change (since P19 would be missed). Also, these markers need to be validated.

We thank the reviewer for this suggestion. While ROC/AUC analyses can be useful, particularly when effect sizes are modest yet consistent, their statistical power is highly dependent on adequate sample sizes. Given the small sample size in our study (n=9 with available RNAseq and VBD), we believe ROC/AUC is not the most appropriate method in this context. Prior studies have shown that for non-parametric tests related to AUC, statistical power is extremely limited when sample sizes are below 12

per group. Additional power simulations suggest that at least 30 samples may be required to estimate AUC values with sufficient confidence. Therefore, due to limited power and the increased risk of false positives/negatives, we believe that differential expression analysis remains the most appropriate approach for our dataset. Nevertheless, we have included a list below with the AUC values for all differentially expressed genes between high MD and low MD, shown in Figure 4H and Extended Data Figure 12C. The two genes previously most related to progesterone signalling, CXCL13 and TNFSF11, have AUC values of 1 and 0.9, respectively, which are considered outstanding AUC values.

We agree with the reviewer that validation of the identified markers in a larger, independent cohort would be valuable. However, this is currently not feasible due to the unique nature of our dataset, which includes controlled sampling during the luteal phase of the menstrual cycle. To our knowledge, no publicly available datasets offer comparable control over menstrual cycle timing. Nonetheless, we identified a study that supports our findings: Toriola et al. (2017, Oncotarget, DOI: 10.18632/oncotarget.17909) assessed TNFSF11/RANKL gene expression in breast tissue from 48 premenopausal women (without controlling for menstrual cycle phase) and found a positive association between TNFSF11 expression and higher mammographic density. This provides independent support for our observation linking TNFSF11 to increased mammographic density.

AUC values for all the differentially expressed genes between high MD and low MD:

AKR7A3: 0.9

ALOX15B: 0.76

C2CD4C: 0.95

CACHD1: 0.9

CLIC6: 1

CRHBP: 0.81

CUX2: 1

CXCL13: 1

GJB5: 1

ITGA10: 1

KLF14: 0.86

LALBA: 0.86

NINJ2: 0.9

PCDH9: 0.81

PRG4: 0.57

RAB3B: 1

RRAGD: 0.95

SLC26A3: 1

TNFSF11: 0.9

TPPP3: 0.9

TPSD1: 0.95

Other points:

1. How were potential confounders (like birth control) accounted for in the analyses?

The BC-APPS1 trial was designed with eligibility criteria to minimise potential confounding from hormonal status variables (study protocol in Supplementary Appendix 1). Potential participants had to have ceased using hormonal contraceptives at least 6 months prior to study entry, with the resumption of regular menstrual cycles.

As mentioned in response to reviewer 3 above, we examined several baseline characteristics - age, breast cancer risk, parity and BMI - to assess whether there was any evidence of differential effects among participants in response to UA, specifically in terms of reduction in cell proliferation (Ki67) or the proportion/activity of LP/LASP cell population. We did not observe any significant correlation between these parameters and treatment response. However, it is important to note that our dataset is limited to 24 women which will restrict the power to detect associations.

We have added the following sentence to the Discussion:

“The response to UA did not show any correlation with age, breast cancer risk, parity or BMI.” (Lines 433-434)

2. The resolutions among figures are very different and some are very hard to visualize.

We have updated the figures in the manuscript to ensure they are all presented at the highest possible resolution for improved clarity and consistency.

Rebuttal Letter for BC-APPS1 Manuscript (#2024-05-10498B)

We thank the Referees for their constructive assessment of our revised manuscript. We are pleased that Referees #1, #2 and #3 found the revisions satisfactory and are very supportive of the manuscript's scientific contribution and impact. We appreciate the detailed feedback from Referee #4, which has helped to further clarify and strengthen several aspects of our data presentation and interpretation. We have carefully addressed each remaining comment and made additional revisions, with the most important changes to the manuscript summarised below:

- 1) We have added connecting lines between paired data points across multiple figures to display individual participants' responses. These updates have been made in Figure 1B-G, Extended Data Figure 1A-C and F-H, Extended Data Figure 2B, and Figure 4D, F and G.
- 2) To better link differential gene expression to downstream cell-cell interactions, we have replaced the original Figure 2F (now moved to Extended Data Figure 7B) with an UpSet plot showing downregulated ligands across the seven major cell populations.
- 3) We have included a dot plot of the 13 genes upregulated in LHS cells after treatment, as well as individual expression graphs for the four MMP genes across participants at baseline and post-treatment, demonstrating their significant upregulation (Extended Data Figure 9).
- 4) We have removed the dot plot in Figure 2J and added a new Extended Data Figure 11 presenting individual gene expression graphs for collagens differentially expressed in BMYO, FB1, FB2 and FB3 cells.
- 5) We have reanalysed the SOX9^{high} versus SOX9^{low} populations using a fixed-intensity thresholding approach confirming that the results remain consistent with our original adaptive thresholding method.
- 6) Additional clarifications and figure enhancements:
 - We revised sentences in the main text and Methods section in response to feedback from Referees 1 and 4.
 - We clarified in the text that the list of collagen ligand-receptor interactions shown in Figure 2J (former 2K) is provided in Supplementary Table 4.
 - We standardised the colour ranges between related panels in Figure 3E and Extended Data Figure 13A (former 11A).
 - We employed consistent nomenclature throughout the manuscript using "baseline" and "post-treatment".
 - We updated figure legends to define abbreviations and clarify cell type labels.

We believe these updates have clarified key findings and strengthened the mechanistic connections within the study.

A detailed point-by-point response to each of Referees' comments follows below.

Referees' comments:

Referee #1 (Remarks to the Author):

The authors addressed all my major concerns.

For the sake of not propagating incorrect (albeit widely used) nomenclature, please correct the following sentence in the introduction:

In both mouse and human mammary glands, lobular-alveolar development results from progesterone induced proliferation of the stem/progenitor cell population.

Should read: "increased branching or increased ductal complexity" instead of "lobular-alveolar development"

We thank Referee #1 for this correction and agree with the suggestion. We have revised the sentence in the introduction as follows:

"In both mouse and human mammary glands, progesterone-induced proliferation of the stem/progenitor cell population results in increased branching and ductal complexity."

Referee #2 (Remarks to the Author):

The authors have substantially revised the manuscript and addressed the prior concerns of this reviewer. The manuscript is now acceptable for publication.

We thank Referee #2 for the positive feedback and for confirming that the revised manuscript is acceptable for publication.

Referee #3 (Remarks to the Author):

The authors have addressed all my comments very well in their revised manuscript and letter of response. Furthermore, the changes made and considerable new data added as a result of the feedback from the other referees have led to an even stronger manuscript.

This study remains extremely important to the field; representing an excellent example of an early phase prevention trial, underpinned by high quality translational research. It supports the important concept of precision prevention whereby anti-progestin therapy could be specifically targeted to women with high mammographic breast density and provides a strong mechanistic rationale for using measures of breast density as surrogate biomarkers. I look forward to following the undoubted impact that this manuscript will have on breast cancer

prevention and more widely on other tumour types.

We thank Professor Brown for her supportive comments. We are grateful for the recognition of our work's significance in the wider cancer prevention field.

Referee #4 (Remarks to the Author):

Overall, the authors' revisions have addressed many of our previous concerns. The narrative has become clearer, the data presentation largely improved, and the methods section is now more thorough. However, a few issues of differing significance remain (see below; organized according to our original points).

We thank Referee #4 for their constructive feedback and agree that their original comments resulted in significant improvements to the manuscript. We appreciate the additional comments and address each point in detail below.

Figure 1 and Extended Data Figure 1:

1. The data in Figures 1 and S1 are tied. Please fix your statistics to consider this. This is true for several other analyses as well.

We appreciate authors' clarifications around the use of paired statistics. Adding lines connecting data points from the same patient, pre and post, would aid interpretation (like current extended figure 10C). Please do this consistently.

We thank the Referees for this helpful suggestion. As requested, we have added connecting lines between paired data points to aid interpretation in the following updated figures: Figure 1B, C, D, E, F, G; Extended Data Figure 1A, B, C, F, G, H; Extended Data Figure 2B; and Figure 4D, F, G.

2. Figure 1E - how were colonies classified?

This point has been well addressed but one clarification is needed. If the cells are seeded at two different densities, which one is used in the results section? This should be specified.

Thank you for your comment. Although we initially plated the cells at two different densities (200 and 400 cells/cm²), we only used the data from the plates seeded at 400 cells/cm² for our analysis. We have updated the Methods section accordingly to clarify this, as follows:

“For each participant sample, cells were plated at 400 cells/cm² in 60-mm culture dishes.”

Figure 2

The authors have done a better job examining their single-cell data but the paper would benefit substantially from better linkage between the findings described in Fig 2A-F and Fig 2G-K (i.e., how does the downregulation observed connect to the interactions presented?).

We thank the Referees for this comment. Since paracrine signalling plays a crucial role in normal breast development we examined differentially expressed ligands to better link the observed downregulation in gene expression to the cell-cell interactions presented in the bottom part of Figure 2. Through this analysis, we highlight that BMYO, FB, and LHS cells are the top source populations for downregulated ligands following UA treatment. This suggests that BMYO and FB populations may exert substantial paracrine effects on other cell types, consistent with their importance in the interaction focused panels of the figure.

To strengthen this connection, we have replaced the original Figure 2F (now moved to Extended Data Figure 7B) with a new UpSet plot showing the numbers of downregulated ligands across the seven major cell populations. For completeness, an UpSet plot of upregulated ligands has also been included (Extended Data Figure 7D). The lists of differentially expressed ligands for each of the seven cell populations is provided in Supplementary Table 2.

We have inserted the following text into the manuscript to clarify this point:

“Given that paracrine signalling is known to play a critical role in normal mammary gland development, we next investigated differentially expressed ligands following UA treatment. LHS cells, but also FB and BMYO cells, showed a high number of down-regulated ligands (Figure 2F). The number of upregulated ligands was lower overall, but higher in LHS cells compared to the other cell types (Extended Data Figure 7D). The list of differentially expressed ligands for each of the seven cell populations is provided in Supplementary Table 2.”

6. Line 214 - Why did you restrict the target genes to the Reactome ECM gene set? This is not justified and trivializes the results in lines 212 - 233. Also, the comment in line 227 - 229 comes out of nowhere.

Current narrative is clearer. Two remaining points: 1. Legend in 2D refers to a star for significance but gives a p-value – please proofread;

We thank the reviewer for pointing this out. We have now updated the figure legend to read as follows:

“Significance is noted with adjusted p-value using Benjamini–Hochberg correction.”

2. For Figure 2H, is there a statistical test to confirm the decrease of strength of these pathways?

The ‘interaction strength’ function plots the inferred communication probability. Prior to comparative analysis, each condition (baseline and post-treatment) is downsampled and processed separately, and ligand–receptor interactions are filtered using a significance threshold of $p < 0.05$ based on the interaction strength (i.e. normalised expression of the ligand in the sender cell and the receptor in the receiver cell). This threshold is also applied at the pathway level using the `computeCommunProbPathway` function.

The pathway plots in Figure 2H compare pathway ‘interaction strength’ for significantly inferred connections in both conditions, with pathways annotated based on their assigned ligand–receptor pairs. At present, a reliable statistical test to compare overall pathway strength between conditions is not available. However, changes in the collagen pathway are supported by multiple complementary analyses, including pathway enrichment analysis and Memento, as well as independent validation from proteomics and Hyperion IMC data presented in Figure 3.

8. Figure 2H - For the ligand-receptor interaction of collagen, are the receptors also down-regulated after UA treatment? This panel is hard to interpretate. Please show single-cell expression data to demonstrate that collagen is downregulated post-UA.

Figure 2J does not make clear how much collagen decreases post treatment. Plots like those in Extended Data 10D would be more convincing. In addition, Figure 2K does not help the statement in line 248-251 regarding the downregulation of collagen production.

We agree that the dot plot in Figure 2J did not clearly show the extent of collagen decrease post-treatment. In response, we have removed the dot plot and, as suggested, added a new Extended Data Figure 11 presenting individual expression graphs for the collagens that are differentially expressed in BMYO, FB1, FB2 and FB3 cell populations. The graphs display expression data across participants at baseline and post-treatment for all collagens included in the chord diagrams, providing a clearer view of collagen downregulation after treatment.

We’ve updated the text in the manuscript as follows:

“Collagen gene expression was down-regulated after UA treatment in FB1 (specifically COL1A1, COL1A2, COL4A1, COL4A2, COL6A1, COL6A3, and COL6A6), FB2 (specifically COL1A1, COL1A2, COL4A1, COL4A2, and COL6A1), FB3 (COL6A3) and BMYO1 cells (specifically COL4A1, COL4A2, and COL6A1) - Extended Data Figure 11; Extended Data Figure 12D.”

To highlight that the collagen is mostly acting on epithelial clusters, you should either use the shade/size of the arrows to indicate strength or at least highlight the epithelial portion of the circos plot.

We have now highlighted the epithelial portion of the chord diagram plots to clearly show that collagens interact predominantly with epithelial subclusters expressing their receptors.

Another approach to check the interaction strength is to look at the expression of collagen receptors on the receptor cell types, rather than only looking at ligand expression.

We have included a table showing the percentage of cells within each target population expressing collagen receptors, both at baseline and following UA treatment. From the perspective of the receptor-expressing cells, the data show that receptor expression remains largely stable with some examples of slight downregulation following treatment.

We have updated the text in the manuscript to make a reference to this table, as follows:

*“A list of the collagen ligand–receptor interactions shown in **Figure 2J**, along with the percentage of cells within each target population expressing collagen receptors, is provided in **Supplementary Table 4**.”*

One additional comment on Figure 2: the authors mention briefly that LHS upregulates MMP genes after treatment. This helps support the idea of an overall stiffness decrease, but is only reported in the text and Supplemental Table 2. It might be worth to bring into a main figure or supplemental figure.

We thank the Referees for this suggestion. We agree that highlighting the upregulation of MMP genes in LHS cells adds support to the proposed reduction in tissue stiffness. In response, we have now included a dot plot showing the 13 genes upregulated in LHS cells, as well as individual expression graphs for the 4 MMPs across participants at baseline and post-treatment. These additions included in Extended Data Figure 9 demonstrate the significant upregulation of MMPs in LHS cells following treatment.

We’ve updated the text in the manuscript as follows:

*“LHS cells displayed a greater number of up-regulated ECM regulatory genes, with 4 out of the 13 genes being matrix metalloproteinases (MMP1, MMP3, MMP10, MMP12), known to play key roles in ECM degradation (**Extended Data Figure 9**).”*

Figure 3

1. Line 241-242. Unexpected why? Was the direction of change consistent?

To be clear, the overlap of detected proteins does not indicate the reliability of the data. We suggest removing this statement.

We agree with this point and have deleted the statement mentioning data reliability.

2. Figure 3C - Why was patient treated as a random effect rather than explicitly considered when it was modelled in the bulk RNA work?

As pre and post treatment samples are paired from the same patient, your statistical tests should use fixed effect. It would be best to build a GLM that models patient.

As pre- and post-treatment samples are paired from the same patient, we agree that the statistical model must account for this pairing. In our analysis, we model treatment as a fixed effect and patient as a random effect, serving as a blocking factor (i.e. baseline/post-treatment samples from individual patients are considered paired in the analysis). This approach allows us to estimate the specific impact of treatment while accounting for the variation between patients (i.e., across baseline/post-treatment pairs). By modelling patient as a random effect, we assume that the individuals in our study are drawn from a larger population, and that individual-level differences are random, not of direct interest, but should be controlled for. This structure appropriately captures the inter-patient variation while focusing inference on the fixed effect of treatment. Our decision to model patient as a random effect is further supported by the peer-reviewed studies, such as the tutorial for MSqRob, in which the authors treated biological replicates as random effects, stating “The effect of each single biological repeat will differ each time one would repeat the experiment” [DOI: 10.1016/j.jprot.2017.04.004].

In response to the referee’s suggestion, we have explored the use of a fixed-effect model to account for patient identity in paired analyses within the MSqRob framework. This model produced results comparable to our original analysis, as shown in the figures below. Using the fixed-effect model, we identified 28 differentially abundant proteins, with 17 of the 28 downregulated proteins being ECM proteins, mirroring the random-effect model and further validating our findings. Two Collagen VI chains (COL6A3 and COL6A6) remained among the most significantly downregulated proteins after UA treatment, regardless of whether patient was modelled as a fixed or random effect. Two others showed model-specific significance: COL6A2 (random) and COL6A1 (fixed). COL1A1 was also consistently downregulated, while COL1A2’s q-value increased to 0.058, and FN1 lost significance in the fixed-effect model. Overall, differential abundance patterns, including q-values (<0.05) and fold changes, were very consistent confirming that Collagen VI and other matrisome protein downregulation is robust to model specification.

We chose to analyse our mass spectrometry data using a robust ridge regression model within the MSqRob framework, for its capacity to address analytical limitations inherent in mass spectrometry-based proteomic analysis. Label-free data-dependent acquisition (DDA) mass spectrometry data differs significantly from discrete, count-based data (e.g., RNA-seq) in terms of missingness, stochastic sampling, and high variability, making generalised linear models (GLMs) less suitable. MSqRob, by contrast, applies robust estimation to down-weight outliers, incorporates implicit normalisation, and uses mixed-effects modelling to account for variability across peptides, samples, and timepoints. These features make it well-suited for mass spectrometry-based proteomics where noise and missing values are common challenges. Furthermore, MSqRob performs normalisation, handles missing values adequately, and conducts statistical modelling at the peptide level, before inferring protein-level quantification. This peptide-level approach provides a more reliable framework for analysing mass spectrometry-based proteomics data compared to common GLM-based tools (e.g., DESeq2 or limma), which typically begin by aggregating a limited number of peptides into protein-level summaries before applying normalisation and modelling. By modelling at the peptide level first, MSqRob retains more of the raw data's structure and variability, improving the robustness and accuracy of differential expression analysis in proteomics.

Importantly, the biological conclusions drawn from our random-effects model are supported by robust follow-up data, reinforcing that this approach is statistically valid and biologically appropriate for our study. For example, Imaging Mass Cytometry confirmed the downregulation of COL1, COL6 and FN1 proteins (Extended Data Figure 13B), while scRNA-seq data confirmed the downregulation of COL1A1, COL1A2, COL6A1, COL6A2, COL6A3, and FN1 in fibroblasts (data available in Supplementary Table 3).

Figure: The same panel figures as presented in Figure 3C, D and E of the manuscript are shown here, displaying results obtained using the fixed-effect model.

Extended Data Figure 11

Figure: The same panel figure as presented in Extended Data Figure 13 of the manuscript is shown here, displaying results obtained using the fixed-effect model.

7. Line 259 - How were cells classified as high vs low? Also, what fraction are Ki67+?

This requires better quantitation, and the threshold testing should be done based on intensity.

We thank the reviewer for their comment regarding the classification of cells as SOX9^{high} vs SOX9^{low} and agree that ensuring robust and reproducible thresholding is essential. In our analysis, we used an **adaptive thresholding** approach (see Figure below), where a SOX9 intensity cut-off was defined individually for each image to best capture SOX9-positive cells while accounting for slide-to-slide variability in staining and signal intensity. This method is based on expert-guided interpretation of image-specific background levels and is supported by published Imaging Mass Cytometry studies (Ijsselsteijn et al., *Cytometry A*, 2021; DOI: 10.1002/cyto.a.24480), which highlight the limitations of applying a single global threshold across heterogeneous tissue sections.

To further assess the robustness of our analysis and address concerns regarding subjectivity, we re-analysed the data using two alternative approaches:

- **Fixed-intensity thresholding** (see Figure below), in which a single SOX9 intensity cut-off, defined as the median of the adaptive thresholds, was applied uniformly across all images. While straightforward, this approach does not account for inter-slide variation and may fail to capture image-specific background differences caused by variation in staining, tissue quality or acquisition conditions.

- Fixed-percentage thresholding** (please see page 38 of previous rebuttal letter submitted on 29th May 2025), in which E-cadherin-positive cells were ranked by SOX9 intensity within each image, and the top and bottom 20% were classified as SOX9^{high} and SOX9^{low}, respectively. This method avoids the need for an intensity cut-off altogether and provides a relative comparison across samples.

Both alternative approaches generated results highly consistent with our original adaptive thresholding method, with only minor differences in statistical significance and no change in the biological interpretation. This consistency across three thresholding strategies indicates that the underlying signal is robust and not an artifact of the classification method.

For the reasons outlined above, we opted to retain our original adaptive thresholding method, as it best accounts for image-specific variability. However, we have added a sentence to the Methods section noting that all three approaches (adaptive, fixed and fixed-percentage threshold) produced similar results, with differences only in statistical significance rather than in biological interpretation, as follows:

“Changing the SOX9^{high} and SOX9^{low} threshold analysis to either a fixed top/bottom 20% expression or a fixed-intensity value produced similar results to those reported (data not shown).”

Figure: Results obtained using the adaptive thresholding approach for SOX9 classification in baseline (T1, top panel) and post-treatment (T2, bottom panel) samples.

Figure: Results obtained using the fixed thresholding approach for SOX9 classification in baseline (T1, top panel) and post-treatment (T2, bottom panel) samples.

Figure 4

Figure 4A&B - How was expression measured? There are several details required to understand your analyses that are missing. Also, please do a tied analysis (Ditto for 4D & 4F).

See comment in Figure 1 point #1 about tie lines.

As detailed above, we have now added connecting lines between paired data points in all figures, including Figure 4D, F, G.

3. Lines 284-286 - There are logic gaps here.

Mostly addressed but the authors should be clear in line 329-331 that onapristone does not decrease MFE in the soft condition.

We have revised the text to specify that onapristone does not decrease MFE under soft conditions, as follows:

“Anti-progestin treatment of breast microstructures using UA or onapristone blocked stiffness-induced increases in SOX9 and C-KIT, as well as MFE; however, onapristone did not reduce MFE under soft conditions (Figure 4B-C, Extended Data Figure 14A-B).”

Small comments:

1. Figure quality and consistency could be improved. For instance, related main and supplemental figures should have the same color range (q values in Figure 3E and Suppl Figure 11A) and consistent nomenclature (T1/T2 versus baseline/post-treatment)

Thank you for this feedback. We have now standardised the colour ranges for related main Figure 3E and Supplementary Figure 13A (former 11A), and have employed consistent nomenclature throughout the manuscript by using “baseline” and “post-treatment”.

2. Some abbreviations are not explained in the figure legends. For example, VST in extended figure 2A and cell types in extended figure 5. Please fix.

We thank the Referees for pointing this out. We have updated the figure legends to define abbreviations, including “VST” (variance stabilised transformation) in Extended Data Figure 2A, Figure 4H, and Extended Data Figure 14C, and have clarified the cell type labels in Extended Data Figure 5.